# TDP-43 and PINK1 mediate *CHCHD10*<sup>S59L</sup> mutation–induced defects in *Drosophila* and in vitro

Minwoo Baek[1,5], Yun-Jeong Choe[1,5], Sylvie Bannwarth[2], JiHye Kim[1], Swati Maitra[1], Gerald W. Dorn II[3], J. Paul Taylor[4], Veronique Paquis-Flucklinger[2] & Nam Chul Kim [1✉]

Mutations in coiled-coil-helix-coiled-coil-helix domain containing 10 (*CHCHD10*) can cause amyotrophic lateral sclerosis and frontotemporal dementia (ALS-FTD). However, the underlying mechanisms are unclear. Here, we generate *CHCH10*<sup>S59L</sup>-mutant *Drosophila melanogaster* and HeLa cell lines to model *CHCHD10*-associated ALS-FTD. The *CHCHD10*<sup>S59L</sup> mutation results in cell toxicity in several tissues and mitochondrial defects. *CHCHD10*<sup>S59L</sup> independently affects the TDP-43 and PINK1 pathways. *CHCHD10*<sup>S59L</sup> expression increases TDP-43 insolubility and mitochondrial translocation. Blocking TDP-43 mitochondrial translocation with a peptide inhibitor reduced *CHCHD10*<sup>S59L</sup>-mediated toxicity. While genetic and pharmacological modulation of *PINK1* expression and activity of its substrates rescues and mitigates the *CHCHD10*<sup>S59L</sup>-induced phenotypes and mitochondrial defects, respectively, in both *Drosophila* and HeLa cells. Our findings suggest that *CHCHD10*<sup>S59L</sup>-induced TDP-43 mitochondrial translocation and chronic activation of PINK1-mediated pathways result in dominant toxicity, providing a mechanistic insight into the *CHCHD10* mutations associated with ALS-FTD.

[1] Department of Pharmacy Practice and Pharmaceutical Sciences, College of Pharmacy, University of Minnesota, Duluth, MN, USA. [2] Inserm U1081, CNRS UMR7284, IRCAN, Université Côte d'Azur, CHU de Nice, Nice, France. [3] Center for Pharmacogenomics, Washington University School of Medicine, St. Louis, MO, USA. [4] Howard Hughes Medical Institute and Department of Cell and Molecular Biology, St. Jude Children's Research Hospital, Memphis, TN, USA. [5] These authors contributed equally: Minwoo Baek, Yun-Jeong Choe. ✉email: kimn@umn.edu

In 2014, Bannwarth et al.[1] identified an S59L substitution in coiled-coil-helix-coiled-coil-helix domain containing 10 (CHCHD10) as the causes of a familial disease characterized by motor function defects, declined cognitive function, and myopathy. Subsequent analyses revealed that CHCHD10[S59L] is a genetic cause of amyotrophic lateral sclerosis with frontotemporal dementia (ALS-FTD)[1]. CHCHD10 encodes a functionally unknown, small protein comprising a putative N-terminal mitochondrial-targeting sequence and a C-terminal CHCHD domain[1]. Identification of mutant CHCHD10 in ALS-FTD and related diseases and its predominant localization in mitochondria provide an opportunity to investigate the mitochondrial origin of neurodegenerative diseases.

Many additional CHCHD10 variants are now known to cause ALS, FTD, and other related degenerative diseases[2,3]. However, the pathogenicity and penetrance of some variants are debatable. Although CHCHD10[R15L] was identified in familial and sporadic cases of ALS, the existence of unaffected carriers in familial cases suggests incomplete penetrance[4–7]. The CHCHD10[P34S] variant occurs in sporadic ALS[8,9], ALS-FTD[10], Parkinson disease[6], and Alzheimer's disease[6], and its overexpression causes mitochondrial defects[11]. However, its pathogenicity is not well supported by genetic evidence[12,13]. CHCHD10[G58R] and CHCHD10[G66V] were identified in mitochondrial myopathy and late-onset spinal muscular atrophy, Jokela type (SMAJ), or Charcot-Marie-Tooth disease type 2 (CMT2), respectively[5,14–16]. Indeed, other ALS-FTD-associated genes, such as valosin-containing protein (VCP) and Matrin 3 (MATR3), also exhibit clinical pleiotropy, including myopathy[17,18]. Thus, although neurodegenerative diseases may be generally associated with myopathy, the basis of CHCHD10-mediated clinical pleiotropy is not understood.

The CHCHD10 mutations identified in familial diseases are dominantly inherited[1,4,5,14]. However, experimental evidence does not support that all disease-causing variants have the same mode of action. Bannwarth et al.[1] showed that CHCHD10 expression in patient tissues is not affected and that overexpression of CHCHD10[S59L] causes mitochondrial defects similar to that in affected patients. This suggests that CHCHD10[S59L] is a dominant gain-of-function mutant. Woo et al.[19] also confirmed such overexpression-mediated cell toxicity of CHCHD10[S59L]. However, CHCHD10[S59L] does not retain its wild-type (WT)-like activity, indicative of a dominant-negative mechanism. Furthermore, patient fibroblasts with either CHCHD10[R15L] or CHCHD10[G66V] exhibit reduced expression and protein instability, supporting a haploinsufficiency mechanism[20,21]. These data indicate that more detailed investigation is necessary to understand the disease-causing mechanism(s) of mutant CHCHD10 and suggest that CHCHD10 mutations have multiple modes of action.

Although some molecular mechanisms for CHCHD10-mediated cell toxicity are known, it is unclear how these mechanisms drive the disease phenotype and whether they can be controlled therapeutically. CHCHD10 interacts with components of the mitochondrial contact site and cristae organizing system (MICOS), and the MICOS complex is decreased in patients with CHCHD10 mutations[11]. However, Straub et al.[21] reported that CHCHD10 is not well localized with the MICOS complex and that CHCHD10–CHCHD2 hetero-complex formation decreases in patient fibroblasts carrying CHCHD10[R15L]. Although Woo et al.[19] reported the physical interaction of CHCHD10 and TDP-43, Genin et al.[22] demonstrated that phosphorylated TDP-43 levels are not associated with the phenotypic severity of CHCHD10[S59L] or CHCHD10[G66V]. Indeed, severity was more closely associated with MICOS complex formation[22]. Because of a lack of a genetically tractable model, most studies rely on the analysis of protein–protein interactions. Therefore, systematic

evaluation of which upstream effectors or signaling pathways are essential for disease pathogenesis is lacking.

Identification of CHCHD10 mutations and pathogenic mitochondrial pathways in ALS-FTD suggest that mitochondrial defects are a primary cause of ALS, FTD, or other related diseases[23–28]. However, this also raises several intriguing questions. Are mitochondrial defects the primary driver of disease in specific subtypes of ALS-FTD? Do all of these subtypes share a common mechanism? Can mitochondrial defects in different subtypes be rescued by activating protective pathways or targeting a common pathogenic mechanism? To address these questions, we used Drosophila and mammalian cell culture models of CHCHD10[S59L]-mediated cell toxicity. In these models, we found that CHCHD10[S59L] expression imparts a toxic gain of function and that this dominant toxicity is mediated through two distinct axes: TDP-43 and PINK1. CHCHD10[S59L] expression increased insolubility and mitochondrial association of TDP-43 and also activated PINK1-mediated pathways. Pharmacologic treatments with peptide inhibitors of TDP-43 translocation to mitochondria or PINK1 kinase activity successfully mitigated degenerative phenotypes in HeLa cells. Small-molecule agonists of mitofusin (MFN), a downstream substrate of PINK1, rescued mutant CHCHD10-induced mitochondrial defects in Drosophila and HeLa cells. Moreover, the MFN agonists enhanced mitochondrial ATP production in a Drosophila ALS-FTD model expressing C9orf72 with expanded GGGGCC repeats. These findings suggest that CHCHD10[S59L]-induced ALS-FTD shares a disease-causing mechanism through mitochondrial translocation of TDP-43 with other subtypes of ALS-FTD and that modulating mitochondrial function by targeting PINK1-mediated pathways may provide a therapeutic strategy for CHCHD10-mediated disease.

## Results

### CHCHD2 and CHCHD10 share a common *Drosophila melanogaster* ortholog.

To elucidate whether an ortholog of CHCHD10 exists in Drosophila, we used the Drosophila RNAi (RNA interference) Screening Center's Integrative Ortholog Prediction Tool[29]. We found that Drosophila and humans (Homo sapiens) have three and two homologous genes for CHCHD10, respectively. Further phylogenetic analysis with a neighbor-joining tree of these genes revealed that CG5010 is the Drosophila gene sharing the highest homology with both human CHCHD2 and CHCHD10 (Supplementary Fig. 1a). Two additional Drosophila homologs, Dmel\CG31007 and Dmel\CG31008 appeared independently after their speciation. The substituted amino acids in human patients are conserved in Drosophila CG5010 (Fig. 1a). In addition, CG5010 is highly expressed in all Drosophila tissues, whereas Dmel\CG31007 and Dmel\CG31008 are expressed only in the testis and weakly in the imaginal discs. A comparison between H. sapiens and Mus musculus suggested that CHCHD10 and CHCHD2 were duplicated before their speciation and that they may be involved in common processes, but in a distinct manner[30]. According to its phylogenetic and expression profile, we deemed CG5010 as a common Drosophila ortholog for both CHCHD10 and CHCHD2. Therefore, we hereafter refer to CG5010 as C2C10H (i.e., CHCHD2 and CHCHD10 homolog) and Dmel\CG31007 and Dmel\CG31008 as C2C10L1 (i.e., CHCHD2–CHCHD10-like 1) and C2C10L2 (i.e., CHCHD2 and CHDHD10-like 2), respectively.

### CHCHD10[S59L] causes dominant degeneration in fly eyes.

To develop Drosophila models for mutant CHCHD10-induced human disease, we generated both Drosophila and codon-optimized human versions of transgenic animals carrying C2C10H[WT] and S81L (CHCHD10[WT] and S59L in human), with

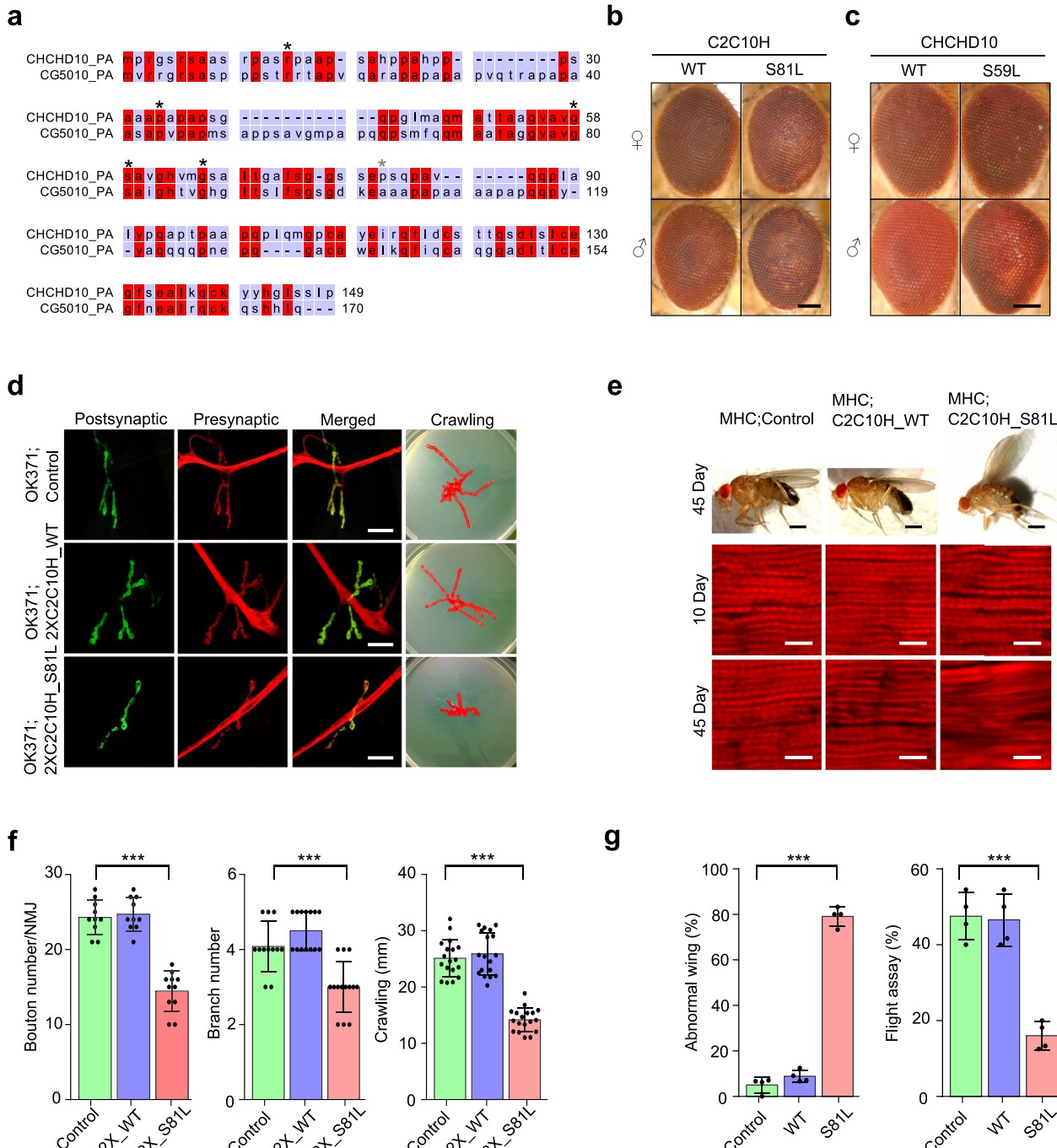

**Fig. 1 C2C10H^S81L is toxic in *Drosophila* eyes, neurons, and muscles. a** Protein sequence alignment of human CHCHD10 and *Drosophila* C2C10H (CG5010). Disease-causing sites (asterisk) are conserved between human CHCHD10 and *Drosophila* C2C10H. **b** *C2C10H^S81L* (representative images from two independent experiments) and **c** human *CHCHD10^S59L* cause age-dependent mild rough eye phenotypes in 40-day-old flies (representative images from two independent experiments). Scale bar = 200 μm. **d** Representative images of neuromuscular junctions and crawling traces from the genotypes indicated (see below for statistical analysis). Scale bar = 20 μm. **e** Adult thoraxes dissected to expose longitudinal indirect flight muscles and stained with phalloidin–Alexa Fluor 594. Flies expressing *C2C10H^S81L* in muscles under control of *MHC*-GAL4 exhibit disrupted sarcomere structures. Scale bar = 0.5 mm (fly) and 10 μm (muscle). **f** Expression of *C2C10H^S81L* in motor neurons results in small synapses, with reduced bouton and branch numbers and defective locomotive activity assessed by the crawling behavior of third-instar larvae. Data are mean ± SD (one-way ANOVA and post hoc Dunnett test, two-sided, ***$p = 5.3e − 10$, 0.0002 and 3.8e − 14 for bouton number ($n = 10$–11), branch number ($n = 12$–14), and crawling distance ($n = 18$) from three independent experiments, respectively). **g** Expression of *C2C10H^S81L* in muscle tissues causes abnormal wing postures and locomotor defects assessed by flight ability. Data are mean ± SD (one-way ANOVA and post hoc Dunnett test, two-sided, ***$p = 4.3e − 10$, 5.5e − 05 for abnormal wing posture, flight assay, respectively; $n = 4$ with >40 flies).

or without a C-terminal FLAG tag by ΦC31 integrase-mediated site-specific integration into the *attp2* landing site on the third chromosome.

When $C2C10H^{WT}$ and $CHCHD10^{WT}$ were expressed in *Drosophila* eyes by the *glass* multimer reporter (GMR)-GAL4 driver, they did not cause any abnormal phenotypes at eclosion. Although $C2C10H^{WT}$ and $CHCHD10^{WT}$ expression did not cause any apparent defects, expression of $C2C10H^{S81L}$ and $CHCHD10^{S59L}$ caused mild but mutation-dependent degeneration with depigmentation as the flies aged, regardless of FLAG tagging (Fig. 1b, c and Supplementary Fig. 1d). The severity of rough eye phenotypes was determined by Flynotyper scoring of the disorderliness of ommatidia[31]. There were statistically significant differences in eye phenotypes (Supplementary Fig. 1b, c, d) without substantial differences in the expression level of WT and mutant proteins (Supplementary Fig. 1e). When we generated fly lines carrying two copies of $C2C10H^{S81L}$ at the *attp2* locus with GMR-GAL4, the rough eye phenotype of $C2C10H^{S81L}$ was enhanced and obvious at eclosion due to high levels of expression by transvection[32] (Supplementary Fig. 1f, g). However, the expression of two copies of $C2C10H^{WT}$ did not induce any eye defects (Supplementary Fig. 1f). These findings indicate that overexpression of $C2C10H^{S81L}$ ($CHCHD10^{S59L}$) generates mutant-dependent degeneration in *Drosophila* eyes.

**$C2C10H^{S81L}$ recapitulates morphologic and functional defects in motor neurons and muscles.** Patients carrying $CHCHD10^{S59L}$ experience motor neuron defects and myopathy. Thus, we examined whether $C2C10H^{S81L}$ and $CHCHD10^{S59L}$ expression cause degenerative phenotypes in *Drosophila* motor neurons and muscles. Due to the weaker cell toxicity of CHCHD10S59L compared to C2C10HS81L in the *Drosophila* system, $C2C10H^{S81L}$ was mainly used in most experiments. Expressing $C2C10H^{WT}$ or $C2C10H^{S81L}$ in motor neurons with OK371-GAL4 did not cause noticeable abnormalities in larvae or adult flies, including viability and fertility. However, homozygous animals for OK371-GAL4 and $C2C10H^{S81L}$ exhibited robust degenerative phenotypes, whereas homozygous $C2C10H^{WT}$ flies with OK371-GAL4 did not show any abnormalities. Third-instar $C2C10H^{S81L}$ homozygous larvae showed striking locomotor dysfunction (Fig. 1d, f), with marked morphologic defects in their neuromuscular junctions (NMJs), including decreased synaptic bouton and branch numbers, and the absence of such NMJ defects in driver-only or $C2C10H^{WT}$-expressing flies (Fig. 1d, f). Consistently, a locomotive defect was also observed with third-instar homozygous larvae expressing $CHCHD10^{S59L}$ in motor neurons (Supplementary Fig. 1h). Expression of $C2C10H^{S81L}$ but not $C2C10H^{WT}$ in indirect flight muscles with MHC-GAL4 led to mutant-dependent muscle degeneration, which was characterized by abnormal wing posture and loss of sarcomere architecture in aged flies (Fig. 1e, g). In addition, $C2C10H^{S81L}$ expression in muscles resulted in functional locomotor defects, as evident in flight assays (Fig. 1g). These findings reveal that the *Drosophila* model recapitulates the dominant cell toxicity of $CHCHD10^{S59L}$ in vivo.

**$CHCHD10^{S59L}$ induces mitochondrial defects.** CHCHD10 is primarily localized at cristae junctions in the mitochondrial intermembrane space[1,33]. To determine whether $CHCHD10^{S59L}$ induces mitochondrial defects, we transiently expressed $CHCHD10^{WT}$ or $CHCHD10^{S59L}$ tagged with C-terminal FLAG in HeLa cells. Immunocytochemistry using an anti-FLAG antibody revealed CHCHD10 localization in mitochondria but not in other cellular sites (Fig. 2a). Consistent with the previous reports[1,19,34], expression of $CHCHD10^{S59L}$ caused notable mitochondrial fragmentation and functional respiratory defects, as measured by

Seahorse XF Cell Mito Stress tests. In contrast, $CHCHD10^{WT}$ expressed at a similar level to $CHCHD10^{S59L}$, induced mitochondrial elongation and improved respiratory function over that of empty vector-transfected cells (Fig. 2a–c and Supplementary Fig. 2a). CHCHD10S59L displayed a punctate staining pattern in mitochondria, whereas CHCHD10WT staining was evenly distributed (Fig. 2a, b). It is unclear whether this punctate pattern is due to CHCHD10 aggregation or mitochondrial fragmentation because the staining patterns of the mitochondrial marker and CHCHD10S59L mostly overlapped (Fig. 2a, b).

To examine mitochondrial morphology and function in response to $C2C10H^{S81L}$ expression in *Drosophila*, we expressed two copies of $C2C10H^{S81L}$ in muscle tissues with the MHC-GAL4 driver. Visualization of mitochondria with a green fluorophore-conjugated streptavidin revealed that expression of $C2C10H^{S81L}$ in indirect flight muscles resulted in muscular degeneration, along with fragmented mitochondria. This finding was in contrast with our observations in the indirect flight muscles of MHC-GAL4 only or $C2C10H^{WT}$-expressing flies (Fig. 2d). We measured ATP levels as an indicator of mitochondrial dysfunction in the flies' thoraxes, which contain primarily muscle tissues. We observed reduced ATP levels in the muscle tissues of flies expressing $C2C10H^{S81L}$, as compared with those of MHC-GAL4/+ control flies (Fig. 2e). Consistent with the effect of $CHCHD10^{WT}$ overexpression in HeLa cells, $C2C10H^{WT}$ expression also increased ATP levels in *Drosophila* muscle tissues (Fig. 2e). Consistently, expression of CHCHD10S59L protein in *Drosophila* muscles caused similar defects (Supplementary Fig. 2d). Therefore, $C2C10H^{S81L}$ and $CHCHD10^{S59L}$ induced consistent mitochondrial fragmentation and functional respiratory defects in *Drosophila* and mammalian cells.

**$CHCHD10^{S59L}$ is not a hypermorphic gain-of-function mutant.** The S59L substitution in CHCHD10 is dominantly inherited[1]. The dominant cell toxicity imparted by $CHCHD10^{S59L}$ overexpression suggests two possible modes of action: dominant negative or dominant gain of function. There are also two possible modes for gain of function: gain of a WT function (hypermorph) or gain of an abnormal function (neomorph)[35]. First, we tested if $CHCHD10^{S59L}$ is a hypermorph. If $CHCHD10^{S59L}$ enhances its own functions to generate cell toxicity, $CHCHD10^{WT}$ co-expression with $CHCHD10^{S59L}$ should enhance the degenerative phenotype of CHCHD10S59L. For this genetic interaction study using $C2C10H^{S81L}$, we generated a fly model carrying two copies of $C2C10H^{S81L}$ in the second and the third chromosomes. When the two $C2C10H^{S81L}$ copies were expressed via GMR-GAL4, it induced relatively mild rough eye phenotypes at eclosion (Fig. 3a) compared with that of third chromosome homozygotes (Supplementary Fig. 1f). We tested several different control *Drosophila* lines with this model fly and did not observe the significant modification of phenotypes by the control lines (Supplementary Fig. 3a). Significantly, $C2C10H^{WT}$ co-expression did not enhance but suppressed the $C2C10H^{S81L}$-induced rough eye phenotype (Fig. 3a), in addition to the morphologic and functional mitochondrial defects in both *Drosophila* (Fig. 3b, c) and HeLa cells (Fig. 3d, e and Supplementary Fig. 3c). $C2C10H^{S81L}$-induced rough eye phenotypes were also not enhanced but rescued by human $CHCHD10^{WT}$ co-expression (Supplementary Fig. 3e). This suggests that $CHCHD10^{S59L}$ is not a hypermorph enhancing its own WT activity. To exclude the possibility of unknown positional effects such as transvection, we generated another transgenic fly line with $C2C10H^{WT}$ or $C2C10H^{S81L}$ in an unrelated landing site, VK27. $C2C10H^{WT}_{VK27}$ also rescued the $C2C10H^{S81L}$ phenotypes, whereas $C2C10H^{S81L}_{VK27}$ enhanced the rough eye phenotype

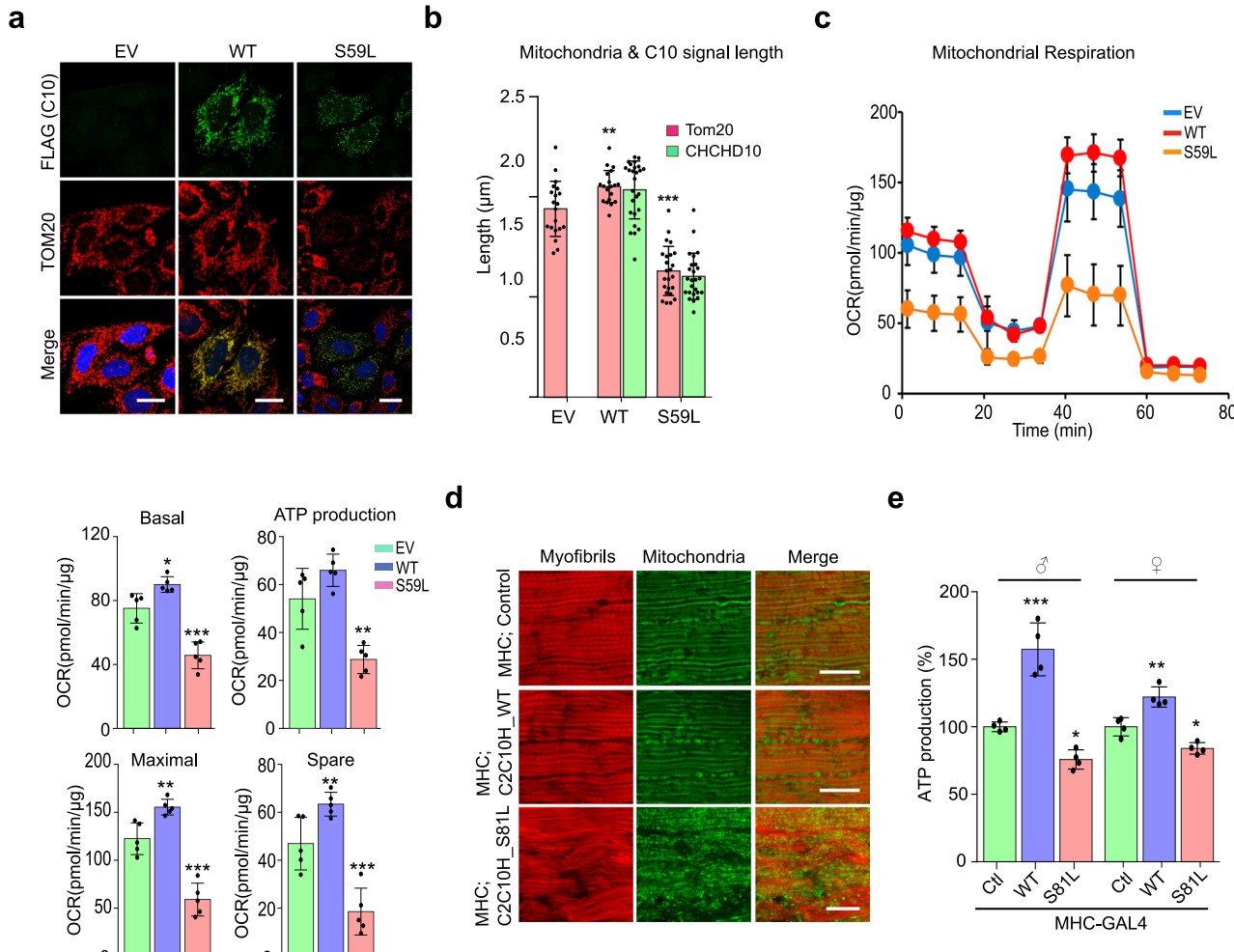

**Fig. 2 CHCHD10$^{S59L}$ mutant induces mitochondrial defects.** HeLa cells were transfected with FLAG-tagged CHCHD10$^{WT}$, CHCHD10$^{S59L}$, and empty vector (EV). **a** Representative images from three independent experiments of HeLa cells immunostained with antibodies against FLAG (green, transfected CHCHD10) and TOM20 (red, mitochondria) 24 h after transfection. Scale bar = 20 µm. **b** Quantification of CHCHD10-FLAG (green) and TOM20 (red) signal length. Data are shown as mean ± SD (one-way ANOVA and post hoc Dunnett test, two-sided, comparison with EV, **$p = 0.0069$, ***$p = 3.7e - 12$; $n = 20$–24 cells from three independent experiments). **c** Oxygen consumption rate (OCR) for each transfectant was measured. After measuring basal respiration rates, oligomycin, FCCP, and antimycin A/rotenone were serially injected to measure ATP production and maximal respiration, respectively. Spare respiratory capacity was then calculated. Graphs of a single representative experiment are shown (mean ± SD). Actual statistical analyses were performed with 11 independent experiments (one-way ANOVA and post hoc Dunnett test, two-sided, comparison with EV, *$p = 0.01867$ and ***$p = 0.00012$ in basal level; **$p = 0.0015$ in ATP level; **$p = 0.0071$ and ***$p = 3.2e - 05$ in maximal level; **$p = 0.002453$ and ***$p = 0.00058$ in spare capacity; detailed information on statistical analyses is available in Supplementary Fig. 9a). **d** Mitochondrial morphology of 45-day-old C2C10H$^{WT}$- and C2C10H$^{S81L}$-expressing flies. Indirect flight muscles were stained with streptavidin–Alexa Fluor 488 (mitochondria, green) and phalloidin–Alexa Fluor 594 (actin filaments, red). Representative images from two independent experiments with 16 flies. Scale bar = 20 µm. **e** ATP levels in thoraxes containing muscle tissues of the indicated genotypes (aged 10 days) were measured. Data are shown as mean ± SD (one-way ANOVA and post hoc Dunnett test, two-sided, ***$p = 0.00018$ and *$p = 0.03662$ in male; **$p = 0.0015$ and *$p = 0.011$ in female; $n = 4$ independent experiments).

(Supplementary Fig. 3d). Consistent with our observations in *Drosophila* eyes, co-expression of *C2C10H$^{WT}$* mitigated the *C2C10H$^{S81L}$*-induced mitochondrial fragmentation and ATP production defects in indirect flight muscles (Fig. 3b, c). An additional copy of *C2C10H$^{S81L}$* exacerbated the defects in both the eyes and muscles (Fig. 3a–c).

In HeLa cells, co-expression of *CHCHD10$^{WT}$* with *CHCHD10$^{S59L}$* clearly mitigated morphologic (Fig. 3d, e) and functional defects (Supplementary Fig. 3c). However, the punctate staining pattern of CHCHD10$^{S59L}$ was not altered by *CHCHD10$^{WT}$* co-expression (Fig. 3d, e), indicating that *CHCHD10$^{WT}$* improves mitochondrial integrity by a mechanism other than restoring mutant protein aggregation. Immunostaining of indirect flight muscles revealed that C2C10H$^{S81L}$ and

CHCHD10$^{S59L}$ proteins also formed aggregate-like structures in vivo (Supplementary Fig. 2b, c). Together, these results suggest that *C2C10H$^{S81L}$* and *CHCHD10$^{S59L}$* clearly form aggregates and do not act as a hypermorph (gain of WT function).

**CHCHD10$^{S59L}$ is a dominant gain-of-function mutant**. The mislocalized punctate pattern of CHCHD10$^{S59L}$ (C2C10H$^{S81L}$) suggests that *CHCHD10$^{S59L}$* (*C2C10H$^{S81L}$*) may be a dominant gain-of-toxic (neomorphic) mutant acquiring abnormal functions. To distinguish a dominant gain-of-toxic mutant from a dominant-negative mutant suppressing the activity of its WT form, we tested whether the toxicity of *CHCHD10$^{S59L}$* and *C2C10H$^{S81L}$* is dependent on the existence of *CHCHD10$^{WT}$* and

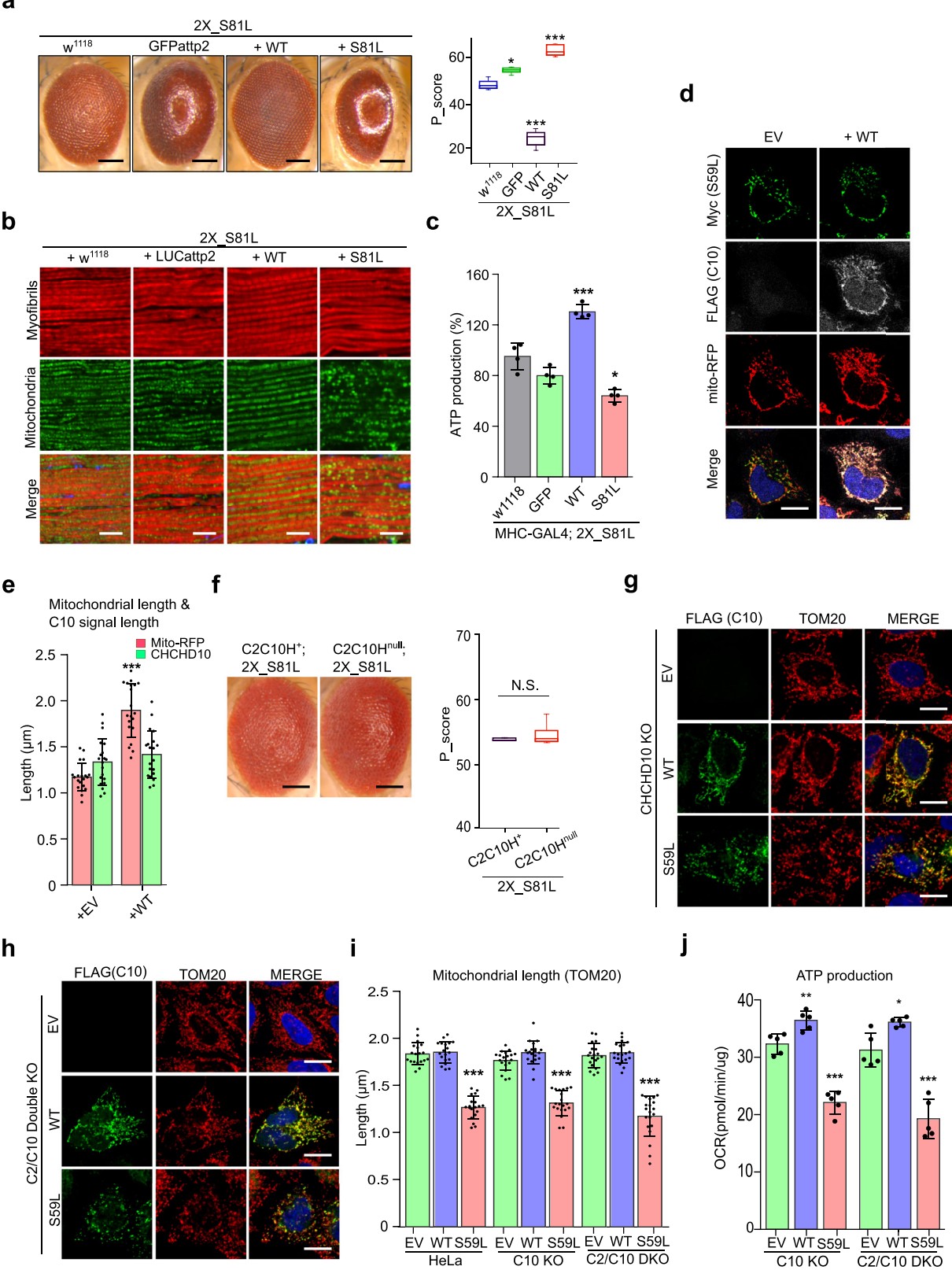

$C2C10H^{WT}$, respectively. If they are true dominant negative, the phenotype of $C2C10H$ deletion mutant ($C2C10H^{null}$) animals and $CHCHD10$ knockout ($CHCHD10^{KO}$) cells will not be enhanced by the expression of $C2C10H^{81L}$ and $CHCHD10^{S59L}$. Because $C2C10H^{null}$ flies do not exhibit an abnormal eye phenotype and are generally healthy[36], we expected that the $C2C10H^{S81L}$-

induced rough eye phenotype would not be present in the $C2C10H^{null}$ background if it is a dominant-negative mutant. However, the rough eye phenotype was robust in the $C2C10H^{null}$ background (Fig. 3f). Although we tested if two paralogs of $C2C10H$ ($Dmel\backslash CG31007$ and $Dmel\backslash CG31008$) can modify the $C2C10H^{S81L}$-induced phenotypes by RNAi-mediated knockdown

**Fig. 3 CHCHD10$^{S59L}$ shows gain-of-toxic function. a** Expression of $C2C10H^{WT}$ with two copies of $C2C10H^{S81L}$ by $GMR$-GAL4 did not exacerbate $C2C10H^{S81L}$-induced rough eye phenotypes. The severity of eye phenotypes was analyzed with Flynotyper[31] after being processed by ilastik[73]. Boxes indicate median and 25th and 75th percentiles (one-way ANOVA and post hoc Dunnett test, two-sided, *$p$ = 0.00132, ***$p$ < 2e − 16 for WT, and ***$p$ = 4e − 06 for S81L; $n$ = 11, 9, 9, and 5 for w$^{1118}$, GFP, WT, and S81L, respectively. Samples were collected from two to three independent experiments). Scale bar = 200 μm. **b** Expression of $C2C10H^{WT}$ with two copies of $C2C10H^{S81L}$ in muscle tissues did not exacerbate abnormal muscle and mitochondrial phenotypes. Scale bar = 10 μm. **c** ATP levels in each of the indicated genotypes were measured. Data are shown as mean ± SD (one-way ANOVA and post hoc Dunnett test, two-sided, ***$p$ = 2.2e − 07 for W and *$p$ = 0.0234 for S81L; $n$ = 4 independent experiments). **d** HeLa cells were co-transfected with Myc-tagged CHCHD10$^{S59L}$ and FLAG-tagged CHCHD10$^{WT}$. Empty vector (EV) was used as a control. $mTagRFP$-$T$-$Mito$-$7$ was co-transfected to visualize mitochondria. Representative images of HeLa cells stained with anti-Myc (green) and anti-FLAG (gray) antibodies 24 h after transfection. Scale bar = 20 μm. **e** Quantification of $mTagRFP$-$T$-$Mito$-$7$ (mitochondria) and CHCHD10$^{S59L}$-Myc signal length. Data are shown are mean ± SD (two-sided $t$ test, ***$p$ = 5.308e − 12; $n$ = 20 cells from three independent experiments). **f** $C2C10H^{S81L}$-induced rough eye phenotypes were robust in the $C2C10H^{null}$ background. Boxes indicate median and 25th and 75th percentiles (two-sided $t$ test, $n$ = 5 flies from two independent experiments). Scale bar = 200 μm. **g** Representative images of $CHCHD10^{KO}$ HeLa cells transfected with $CHCHD10^{WT}$ and $CHCHD10^{S59L}$. Cells were immunostained with antibodies against FLAG (green, CHCHD10) and TOM20 (red, mitochondria). Scale bar = 20 μm. **h** Representative images of $CHCHD2$ and $10^{KO}$ HeLa cells transfected with $CHCHD10^{WT}$ and $CHCHD10^{S59L}$. Cells were immunostained with FLAG (green, CHCHD10) and TOM20 (red, mitochondria) antibodies. Scale bar = 20 μm. **i** Quantification of TOM20 signal length (mitochondria) from $CHCHD10^{WT}$ and $CHCHD10^{S59L}$ transfected HeLa, $CHCHD10^{KO}$, and $CHCHD2/10^{DKO}$ HeLa. Data are shown as mean ± SD (one-way ANOVA and post hoc Dunnett test, two-sided, ***$p$ = 3.3e − 16, <2e − 16, and <2e − 16 for HeLa, C10 KO and C2/C10 KO, respectively; $n$ = 20 cells from three independent experiments). **j** $CHCHD10^{KO}$ and $CHCHD2/10^{DKO}$ HeLa cells were transfected with EV, FLAG-tagged $CHCHD10^{WT,}$ and $CHCHD10^{S59L}$. After 24 h, mitochondrial respiration for each transfectant was measured via Seahorse XF Cell Mito Stress tests. A representative graph of a single experiment is shown (mean ± SD). Actual statistical analyses were performed with three (C10 KO) and five (C2/C10 KO) independent experiments (one-way ANOVA and post hoc Dunnett test, two-sided, comparison with EV, **$p$ = 0.0073 and ***$p$ = 2.2e − 06 for WT and S59L in C10 KO, *$p$ = 0.0238 and ***$p$ = 2.1e − 05 for WT and S59L in C2/C10 KO; detailed information on statistical analyses is available in Supplementary Fig. 9b, c). NS not significant.

(Supplementary Table 1), we did not observe any significant interaction as they are not well expressed in *Drosophila* tissues except testis. Together, these results suggest that $C2C10H^{S81L}$ is not a dominant-negative mutant requiring $C2C10H$ to generate its cell toxicity.

To further validate this result in a mammalian system, we generated $CHCHD10^{KO}$ HeLa cells via the CRISPR/Cas9 system (Supplementary Fig. 3f). Although the CHCHD10 protein was not detected with an anti-CHCHD10 antibody in the $CHCHD10^{KO}$ lines, mitochondrial morphology and respiratory functions were not affected (Fig. 3g, i and Supplementary Fig. 3g). Consistent with results from our *Drosophila* model, the expression of $CHCHD10^{S59L}$ induced $CDCDD10^{WT}$-independent mitochondrial toxicity in $CHCHD10^{KO}$ HeLa cells (Fig. 3g, i, j and Supplementary Fig. 3h). To further investigate the possibility that CHCHD10$^{S59L}$ might suppress a close paralog CHCHD2 activity to generate the mitochondrial toxicity in a dominant-negative manner, we generated $CHCHD2$ and $CHCHD10$ double-knockout ($CHCHD2/10^{DKO}$) cells (Supplementary Fig. 3f). Consistent with a previous report[34], mitochondrial morphology and respiratory functions were not affected in $CHCHD2/10^{DKO}$ (Fig. 3h, i and Supplementary Fig. 3g). The expression of $CHCHD10^{S59L}$ in $CHCHD2/10^{DKO}$ cells also generated significant morphological and functional defects in mitochondria (Fig. 3h–j and Supplementary Fig. 3h). All these data support that $CHCHD10^{S59L}$ mutation is a dominant gain-of-function (i.e., toxic) mutation causing aggregation of the mutated protein, not a dominant-negative mutant suppressing the activity of its WT counterpart.

**CHCHD10$^{S59L}$ proteins form insoluble aggregates.** The punctate staining pattern of CHCHD10$^{S59L}$ was not affected when CHCHD10$^{WT}$ rescued CHCHD10$^{S59L}$-induced mitochondrial fragmentation. The S59L substitution is located in the hydrophobic domain of an intrinsically disordered region in CHCHD10 (Supplementary Fig. 4a). Because the hydrophobic domain and intrinsically disordered region play a role in protein folding and aggregation, we examined whether CHCHD10$^{S59L}$ proteins accumulate in mitochondria as relatively insoluble aggregates. We detected insoluble CHCHD10$^{S59L}$ proteins (i.e.,

RIPA-insoluble and urea-soluble) by immunoblotting (Fig. 4a). Co-expression of $CHCHD10^{WT}$ with $CHCHD10^{S59L}$ did not suppress the accumulation of insoluble CHCHD10$^{S59L}$ (Fig. 4a), which was consistent with the presence of punctate structures of CHCHD10$^{S59L}$ in elongated mitochondria by co-expressing CHCHD10$^{WT}$ (Fig. 3d, e). Therefore, CHCHD10$^{S59L}$ clearly forms insoluble aggregates in mitochondria. Again, CHCHD10$^{WT}$ co-expression did not affect the insolubility and aggregate formation. However, it is still unclear whether the presence of insoluble CHCHD10$^{S59L}$ is more relevant to toxicity than that of other intermediate species, such as oligomers.

**CHCHD10$^{S59L}$ induces TDP-43 insolubility and mitochondrial translocation.** Mutations in *TARDBP* cause ALS and FTD, and cytoplasmic TDP-43 aggregates are a hallmark in most patients with ALS and/or FTD[37,38]. In addition, mitochondrial translocation of TDP-43 and toxicity is reported in cell cultures and patient tissues[25]. Therefore, we hypothesized that CHCHD10 affects TDP-43 aggregation or mitochondrial translocation. We examined insoluble TDP-43 levels after $CHCHD10^{WT}$ and $CHCHD10^{S59L}$ transfection. Strikingly, $CHCHD10^{S59L}$ expression increased insoluble TDP-43, whereas $CHCHD10^{WT}$ expression decreased insoluble TDP-43 (Fig. 4b). Co-expression of $CHCHD10^{WT}$ with $CHCHD10^{S59L}$ suppressed $CHCHD10^{S59L}$-induced insoluble TDP-43 (Fig. 4b).

We next determined whether $CHCHD10^{WT}$ and $CHCHD10^{S59L}$ expression affects mitochondrial translocation of TDP-43. Despite similar expression levels in total lysates, exogenously expressed $TARDBP^{WT}$ and three pathogenic mutants (G298S, A315T, and A382T) showed an elevated amount in the mitochondrial fraction of $CHCHD10^{S59L}$-expressing cells over that of empty vector-transfected cells. We observed decreased mitochondrial distribution of TDP-43$^{WT}$ and its mutants in $CHCHD10^{WT}$-expressing cells (Fig. 4c, d). Furthermore, co-expression of $CHCHD10^{WT}$ with $CHCHD10^{S59L}$ reduced the amount of TDP-43 and its mutants in the mitochondrial fraction, which was increased by $CHCHD10^{S59L}$ expression (Fig. 4e, f). Immunofluorescence staining confirmed the increased mitochondrial localization of TDP-43 in $CHCHD10^{S59L}$-expressing HeLa cells with transiently transfected

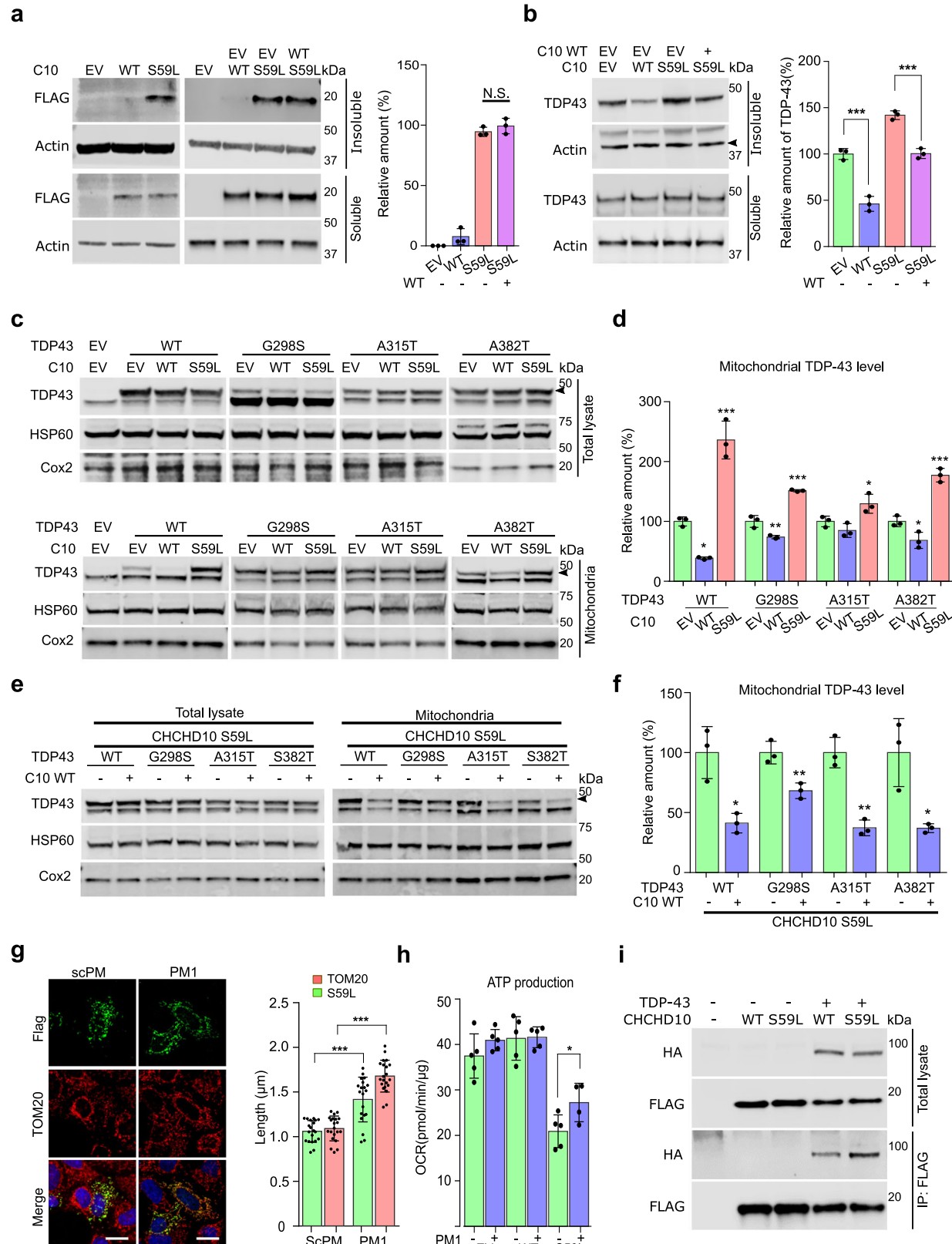

TDP-43 (Supplementary Fig. 4b). In addition, increased cytoplasmic and mitochondrial localization of endogenous TDP-43 was observed in SH-SY5Y neuroblastoma cells expressing *CHCHD10^{S59L}* (Supplementary Fig. 4c, d). An in vivo experiment with a humanized *Drosophila* model replacing the entire TBPH coding region with human *TARDBP*[39] revealed significantly

increased TDP-43 association with mitochondria in *C2C10H^{S81L}*-expressing muscle tissues, as compared to *C2C10H^{WT}*-expressing muscles (Supplementary Fig. 4e). These observations suggest that the prevention of TDP-43 mitochondrial translocation, as demonstrated by Wang et al.[25], may be a potential therapeutic strategy for *CHCHD10*-induced ALS-FTD. To test whether

**Fig. 4 $CHCHD10^{S59L}$ increases TDP-43 insolubility and mitochondrial translocation. a** HeLa cells were transfected with FLAG-tagged $CHCHD10^{WT}$ or $CHCHD10^{S59L}$. After 24 h, cells were subjected to sequential protein extraction with RIPA and urea buffers. Immunoblotting was conducted with anti-FLAG and anti-actin (loading control) antibodies. Data are shown as mean ± SD (one-way ANOVA and post hoc Tukey's test, two-sided, NS, not significant; $n = 3$ independent experiments). **b** HeLa cells transfected with FLAG-tagged $CHCHD10^{WT}$ or $CHCHD10^{S59L}$ with $TARDBP$ were subjected to sequential protein extraction with RIPA and urea buffers. Immunoblotting was performed with anti-TDP-43 or anti-actin (loading control) antibodies. Data are shown as mean ± SD (one-way ANOVA and post hoc Tukey's test, two-sided, ***$p = 0.000738$ for WT, ***$p = 0.000533$ for S59L + WT; $n = 3$ independent experiments). **c, d** HeLa cells were co-transfected with FLAG-tagged $TARDBP^{WT}$ or disease-causing mutants combined with empty vector (EV), $CHCHD10^{WT}$, or $CHCHD10^{S59L}$. After 24 h, mitochondria were fractionated. Immunoblotting was conducted with anti-TDP-43 (arrowhead indicates transfected TDP-43), anti-HSP60, and anti-TOM20 (loading controls) antibodies. Data are shown are mean ± SD (one-way ANOVA and post hoc Dunnett test, two-sided, *$p = 0.01284$ and ***$p = 0.00021$ in TDP-43 WT, **$p = 0.0032$ and ***$p = 7.9e-05$ in G298S, *$p = 0.0475$ in A315T, *$p = 0.02604$ and ***$p = 0.00031$ in A382T). **e, f** HeLa cells were co-transfected with FLAG-tagged $TARDBP^{WT}$ or disease-causing mutants combined with EV or $CHCHD10^{WT}$. After 24 h, mitochondria were fractionated. Immunoblotting was conducted with anti-TDP-43 (arrowhead indicates transfected TDP-43) or anti-HSP60 and anti-TOM20 (loading control) antibodies. Data are shown as mean ± SD (two-sided $t$ test, *$p = 0.011759$, **$p = 0.008552$, *$p = 0.001604$, and **$p = 0.018798$ for TDP-43 WT, G298S, A315T, and A382T, respectively). **g** HeLa cells expressing $CHCHD10^{S59L}$ were treated with a control peptide (scPM, 5 μM) or TDP-43 inhibitor (PM1, 5 μM). Representative images of HeLa cells stained with anti-FLAG (green, $CHCHD10^{S59L}$) and TOM20 (red, mitochondria) antibodies. Data are shown as mean ± SD (two-sided $t$ test, ***$p = 1.26e-06$ and $8.98e-15$ for S59L and Tom20, respectively; $n = 20$ cells from three independent experiments). Scale bar = 20 μm. **h** HeLa cells expressing $CHCHD10^{S59L}$ were treated with a control peptide (scPM, 2 μM) or TDP-43 inhibitor (PM1, 2 μM). Mitochondrial respiration was measured by Seahorse XF Cell Mito Stress tests. A representative graph of a single experiment is shown (mean ± SD). Actual statistical analyses were performed with four independent experiments (one-way ANOVA and post hoc Dunnett test, two-sided, comparison with EV, *$p = 0.024685$; Detailed information on statistical analyses is available in Supplementary Fig. 9f). **i** HEK293 cells were transfected with FLAG-tagged $CHCHD10^{WT}$ or $CHCHD10^{S59L}$ with/without HA-tomato-tagged $TARDBP$. After 24 h, lysates were subjected to co-immunoprecipitation with anti-FLAG (M2) affinity gels. Immunoblotting was conducted with anti-FLAG (CHCHD10) and HA (TDP-43) antibodies. A representative image from three independent experiments is shown.

inhibition of TDP-43 mitochondrial translocation recovers $CHCHD10^{S59L}$-induced mitochondrial morphologic and functional defects, we treated $CHCHD10^{S59L}$-transfected cells with the PM1 peptide inhibitor of TDP-43 mitochondrial translocation[25]. Notably, the morphologic and functional defects caused by $CHCHD10^{S59L}$ were ameliorated by PM1 (Fig. 4g, h and Supplementary Fig. 4f), suggesting that increased mitochondrial translocation are critical to $CHCHD10^{S59L}$-induced cell toxicity. Although it has been demonstrated that TDP-43 can translocate into mitochondria in diseases[25] or stressed conditions[40], it is not clear how TDP-43 can be abnormally retained in mitochondria upon $CHCHD10$ mutation. We hypothesized that $CHCHD10^{S59L}$ increases the binding capacity of CHCHD10 to TDP-43 in mitochondria because CHCHD10 lacks catalytic activity, and physical interactions between CHCHD10 and TDP-43 are reported[19]. To test this hypothesis, HEK293T cells were transfected with $CHCHD10^{WT}$-FLAG or $CHCHD10^{S59L}$-FLAG with/without TDP-43-tomato-HA, followed by immunoprecipitation with anti-FLAG affinity beads. Significantly, $CHCHD10^{S59L}$ showed an increased binding capacity to TDP-43 (Fig. 4i and Supplementary Fig. 4g). This was reaffirmed by using another combination of purification tags: CHCHD10-HA and TDP-43-FLAG (Supplementary Fig. 4h).

**PINK1/parkin mediates dominant degeneration in the $C2C10H^{S81L}$ Drosophila model.** Because $C2C10H^{S81L}$ and $CHCHD10^{S59L}$ expression caused consistent mitochondrial toxicity in Drosophila and human cells, respectively, we have further dissected genetic pathways of $C2C10H^{S81L}$-mediated cell toxicity using our Drosophila model. Because of notable mitochondrial fragmentation and functional respiratory defects, we hypothesized that the genes involved in mitochondrial dynamics or mitochondrial quality control are effectors of $CHCHD10^{S59L}$-driven mitochondrial pathogenesis. We performed genetic interaction studies using the Drosophila $C2C10H^{S81L}$ eye model with various RNAi, classical deficiency, and duplication lines (Supplementary Table 1). The most potent dominant suppressor was PINK1. Downregulation of $PINK1$ by RNAi recovered the rough eye phenotype produced by two copies of $C2C10H^{S81L}$, whereas $PINK1$ overexpression enhanced the rough eye

phenotype (Fig. 5a). Moreover, RNAi-mediated depletion of $park$ (i.e., the Drosophila gene encoding parkin), a downstream partner of PINK1 for the mitochondrial quality control, also marginally rescued the rough eye phenotype (Fig. 5a).

We also tested several other genes that are parallel or downstream of the PINK1/parkin pathway, such as $mul1$, $ari1$, $march5$, $Drp1$, $marf$, and $TER94$ (Supplementary Table 1). Only overexpression of $marf$, a Drosophila ortholog of $MFN$ (i.e., mitofusin), showed mild suppressive effects with consistent RNAi-mediated enhancement (Fig. 5a). This indicates that Marf may be a downstream effector of the PINK1/parkin pathway during $CHCHD10^{S59L}$-mediated pathogenesis. However, this observation does not exclude the other genes tested from playing a role in $C2C10H^{S81L}$-mediated pathogenesis. Consistent with our findings in the eye model, RNAi-mediated depletion of $PINK1$ also recovered $C2C10H^{S81L}$-dependent indirect flight muscle degeneration (Fig. 5b). $PINK1$ knockdown rescued sarcomere structure, extended mitochondrial length, and increased ATP production (Fig. 5b, c), improving $C2C10H^{S81L}$-induced defects, including abnormal wing posture and flight ability (Supplementary Fig. 5a). Human $CHCHD10^{S59L}$-induced rough eye phenotypes were also rescued by $PINK1$ knockdown. (Supplementary Fig. 5b). These results suggest that $C2C10H^{S81L}$ (and human $CHCHD10^{S59L}$) activates the PINK1/parkin pathway to generate dominant cell toxicity in the Drosophila system.

**PINK1/parkin mediates dominant cell toxicity in HeLa cells and patient-derived fibroblasts.** RNAi-mediated knockdown of $PINK1$ remarkably rescued $CHCHD10^{S59L}$-induced mitochondrial morphologic and functional defects in HeLa cells (Fig. 5d–f and Supplementary Fig. 5c). Furthermore, the expression of $CHCHD10^{S59L}$ in $PINK1$-knockout HeLa cells did not cause mitochondrial fragmentation (Supplementary Fig. 5d). Although $MFN2$ knockdown marginally affected the $CHCHD10^{S59L}$-induced mitochondrial defects in HeLa cells (Fig. 5d–f), the simultaneous knockdown of the two paralogs $MFN1$ and $MFN2$ enhanced the mitochondrial defects (Supplementary Fig. 5e, f). However, overexpression of $MFN2$-YFP was uninterpretable because of its strong mitochondrial clustering activity[41], although we detected marginal beneficial effects on respiratory function

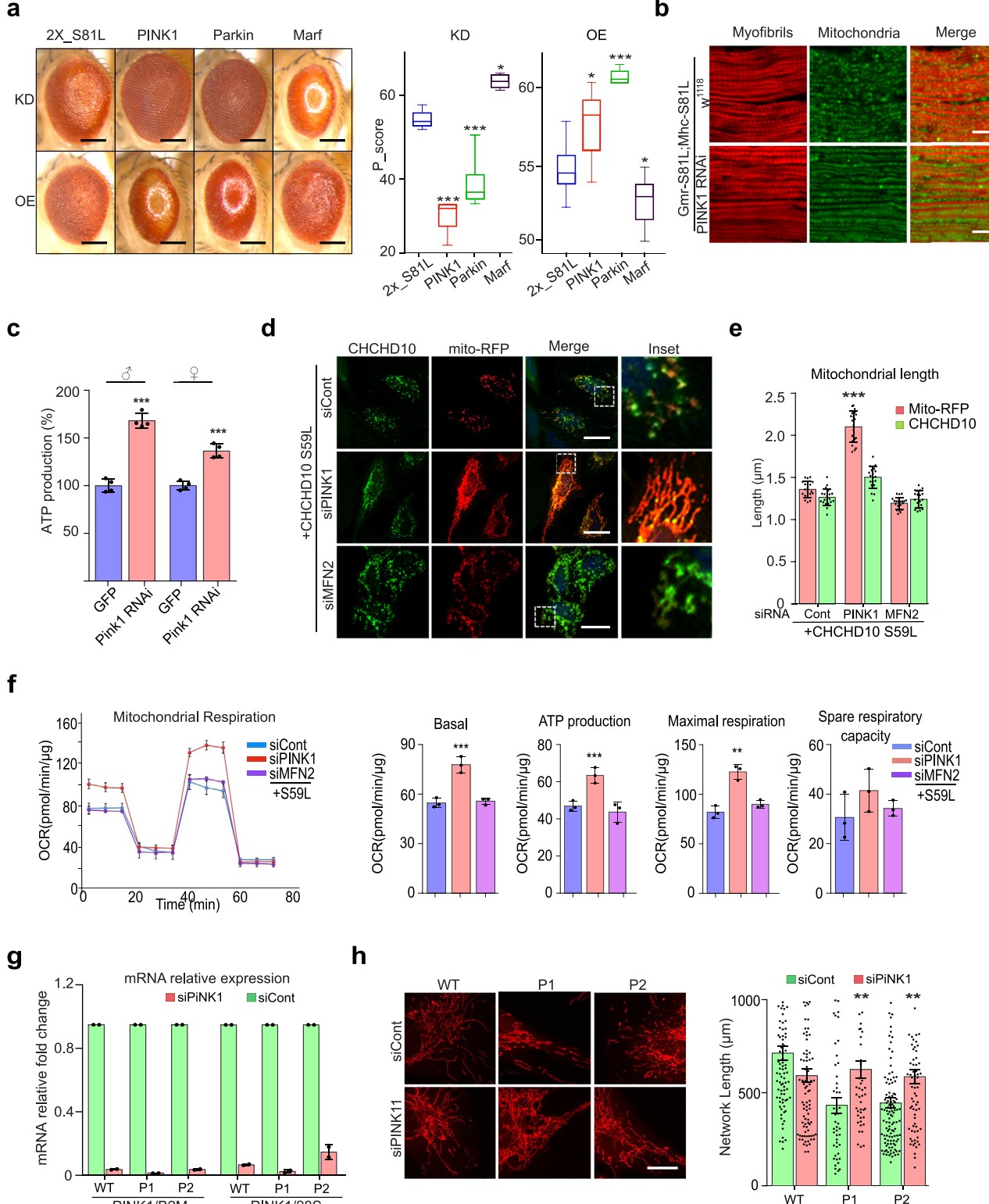

(Supplementary Fig. 5g, h). In our *Drosophila* model, both *Drp1* knockdown and overexpression exhibited detrimental effects (Supplemental Table 1). However, RNAi-mediated depletion of *DRP1* in HeLa cells successfully reversed *CHCHD10^{S59L}*-induced mitochondrial fragmentation and exerted beneficial effects on respiratory function (Supplementary Fig. 5i, j). Consistently, *CHCHD10^{S59L}*-mediated respiratory defects were slightly enhanced by *DRP1-YFP* co-expression, although it was not

statistically significant. (Supplementary Fig. 5h). Indeed, CHCHD10^{S59L} aggregates were nearly absent with *DRP1* knockdown (Supplementary Fig. 5g, i), in contrast with persistent aggregates with *PINK1* knockdown (Fig. 5d, e). We speculated that pre-elongated mitochondria with *DRP1* knockdown reduces the local concentration of CHCHD10^{S59L} proteins and thus prevents aggregate formation. Although it is apparent that modulation of mitochondrial dynamics may be beneficial in reducing

**Fig. 5 $CHCHD10^{S59L}$-induced degeneration is rescued by $Pink1$ downregulation. a** RNAi-mediated knockdown effects of $PINK1$, $Park$, and their downstream target, $Marf$ ($Drosophila$ mitofusin) on C2C10H[S81L]-induced eye phenotypes. Boxes indicate median and 25th and 75th percentiles (one-way ANOVA and post hoc Dunnett test, two-sided, or Fisher's least significant difference, ***$p = 1.3e − 06$, 0.0004 and *$p = 0.0486$ in KD, *$p = 0.04797$, ***$p = 0.00015$, and *$p = 0.0588$ in OE for PINK1, parkin and Marf, respectively; $n = 4$–11 flies from a single experiment; results were verified with multiple fly lines; see Supplementary Table 1). Scale bar = 200 μm. **b** RNAi-mediated knockdown of $PINK1$ in muscles rescued C2C10H[S81L]-induced muscle degeneration and mitochondrial defects. Representative images from three independent experiments. Scale bar = 20 μm. **c** ATP levels in thoraxes from 10-day-old flies were measured. Data are shown as mean ± SD (two-sided $t$ test, ***$p = 1.22e − 05$ for male and 0.000157 for female; $n = 4$ independent experiments). **d**, **e** HeLa cells were transfected with siRNAs targeting $PINK1$ or $MFN2$. After 18 h, the cells were transfected with $CHCHD10^{S59L}$ and $Mito-RFP$. Representative images of transfected HeLa cells immunostained with antibodies against FLAG. Scale bar = 20 μm. Quantification of Mito-RFP and $CHCHD10^{S59L}$-FLAG signal length. Data are shown as mean ± SD (one-way ANOVA and post hoc Dunnett test, two-sided, comparison with EV; ***$p < 2e − 16$; $n = 20$ cells from three independent experiments). **f** Mitochondrial respiration was measured by Seahorse XF Cell Mito Stress tests 24 h after $CHCHD10^{S59L}$ transfection into siRNA-treated cells. Graphs of a single representative experiment are shown (mean ± SD). Actual statistical analyses were performed with 8 independent experiments (one-way ANOVA and post hoc Dunnett test, two-sided, comparison with EV, ***$p = 0.00034$, 0.00031 and **$p = 0.0055$ for basal, ATP and maximal level, respectively; detailed information on statistical analyses is available in Supplementary Fig. 9g). **g** RT-qPCR analysis of $PINK1$ in siRNA-transfected fibroblasts. Results are shown as fold change of $PINK1$ mRNA expression in siPINK1 transfected fibroblasts relative to siControl transfected fibroblasts. Data were normalized to β-2 macroglobulin (B2M) or 28S. Results shown are mean ± SEM from two independent experiments. **h** PINK1 downregulation by siRNA treatment reversed mitochondrial network fragmentation in $CHCHD10^{S59L/+}$ patient fibroblasts. Patients (P1, P2) or wild-type (WT) fibroblasts were transfected with control siRNA (siCont) or PINK1 siRNA (siPINK1). P1 and P2 correspond to patient V-10[1] and to patient IV-3[11], respectively. Representative images of the mitochondrial network with MitoTracker staining. Scale bar = 20 μm. Mitochondrial network length was quantified from two independent experiments with 48–116 randomly selected individual cells. Differences between siCont and siPINK1 were analyzed by two-sided Mann–Whitney test (**$p = 0.0024$ and 0.0013 for patients 1 and 2, respectively).

$CHCHD10^{S59L}$-induced cell toxicity, modulating PINK1, parkin, or MFN is less detrimental than directly modulating DRP1. Finally, we examined whether RNAi-mediated downregulation of PINK1 affects the fragmented mitochondrial network observed in fibroblasts derived from patients carrying a $CHCHD10^{S59L}$ allele[1,11]. Transfection with PINK1 small interfering RNA (siRNA) successfully reduced the amount of $PINK1$ transcripts in both control and patient fibroblasts (Fig. 5g). MitoTracker staining and analysis for control and PINK1 siRNA-transfected cells showed that the loss of PINK1 rescued the fragmented mitochondrial network found in patient-derived fibroblasts (Fig. 5h).

**Parkin-mediated mitophagy induces cell toxicity.** Upon mitochondrial stress or damage, PINK1 accumulates in mitochondria[42] and recruits parkin by phosphorylating ubiquitin and other substrates, including parkin, resulting in MFN1/2 degradation and mitophagy to remove damaged mitochondria[43–46]. The suppressive effects of $PINK1$ and $PRKN$ knockdown suggest that CHCHD10[S59L] induces PINK1 accumulation in mitochondria, activating the PINK1/parkin pathway. Indeed, PINK1-YFP accumulated in the mitochondria of $CHCHD10^{S59L}$-expressing HeLa cells, in contrast with that of $CHCHD10^{WT}$ overexpression (Fig. 6a, c). Because $PRKN$ expression is deficient in HeLa cells, stable cell lines expressing $YFP-Parkin$ (HeLa[YFP-Parkin]) have been established and widely used to study parkin-mediated mitophagy[42]. When $CHCHD10^{S59L}$ was transiently expressed in HeLa[YFP-Parkin] cells, YFP-Parkin was also recruited to mitochondria (Fig. 6b, d), clearly demonstrating that CHCHD10[S59L] induces PINK1 stabilization, accumulation on mitochondria, and subsequent parkin recruitment. This suggests that PINK1/parkin-mediated mitophagy is toxic in this system. Indeed, enhancing autophagy by $Atg1$ expression in the $C2C10H^{S81L}$ eye model was synergistically lethal (Supplementary Table 1), although beneficial effects of $Atg1$ overexpression have been reported in other $Drosophila$ disease models[47,48]. However, strong mitochondrial abnormalities in $PRKN$-deficient HeLa cells also indicate the presence of parkin-independent toxic mechanisms. To explain $CHCHD10^{S59L}$-induced cell toxicity in both $PRKN$-deficient HeLa cells and HeLa[YFP-Parkin], we hypothesized that PINK1-mediated, parkin-independent mitophagy[49] is highly activated in $PRKN$-deficient HeLa cells by $CHCHD10^{S59L}$ and that

dysregulated mitophagy is amplified by Parkin in HeLa[YFP-Parkin]. To test this hypothesis, we first assessed LC3 conversion and accumulation in mitochondria to examine whether autophagosome formation is activated by $CHCHD10^{S59L}$ expression in HeLa cells.

In both HeLa and HeLa[YFP-Parkin] cells, $CHCHD10^{S59L}$ expression increased LC3 accumulation (Fig. 6e, f) and LC3 conversion (Fig. 6g). However, only small portions of LC3 accumulation co-localized with mitochondria in $PRKN$-deficient HeLa cells, as compared with strong mitochondrial co-localization of LC3 accumulation in HeLa[YFP-Parkin] cells (Fig. 6e, h). Co-staining of lysosomes and mitochondria also revealed limited lysosomal marker staining in the mitochondria of $PRKN$-deficient Hela cells (Supplementary Fig. 6a). To confirm this observation, we determined the mitochondrial LC3 levels after fractionating mitochondria. Although LC3-II amount increased in mitochondrial fractions of $CHCHD10^{S59L}$-transfected HeLa and HeLa[YFP-Parkin], significantly more LC3-II were detected in HeLa[YFP-Parkin] as expected (Supplementary Fig. 6b). A mitochondrial fraction from $CHCHD10^{S59L}$- transfected SH-SY5Y expressing parkin endogenously showed clear LC3-II accumulation (Supplementary Fig. 6c). To determine whether the increased LC3 levels from $CHCHD10^{S59L}$-transfected cells correspond with mitophagic turnover, we examined mitolysosomes using the mito-QC system in HeLa, HeLa[YFP-Parkin], and SH-SY5Y cells. Consistent with LC3 conversion and accumulation, mito-QC-positive mitolysosomes (GFP-negative/mCherry-positive) were increased in CHCHD10[S59L]-expressing cells, especially in HeLa[YFP-Parkin] and SH-SY5Y (Fig. 6i, j and Supplementary Fig. 6d, e). Therefore, it is apparent that mitophagy induction is a major phenomenon when CHCHD10[S59L] is expressed. However, it is still unclear whether PINK1-mediated, parkin-independent mitophagy mildly induced by CHCHD10[S59L] contributes to the cell toxicity in $PRKN$-deficient HeLa cells. Thus, to test whether the induction of PINK1-mediated, parkin-independent mitophagy by CHCHD10[S59L] is critical for generating cell toxicity in $PRKN$-deficient HeLa cells, we reduced the expression of two mitophagy receptors involved in PINK1-mediated, parkin-independent mitophagy, $NDP52$ and optineurin ($OPTN$)[49] by RNAi. Knockdown of these two receptors had no effect on $CHCHD10^{S59L}$-mediated cell toxicity in $PRKN$-deficient HeLa cells, but increased respiratory activity significantly in HeLa[YFP-

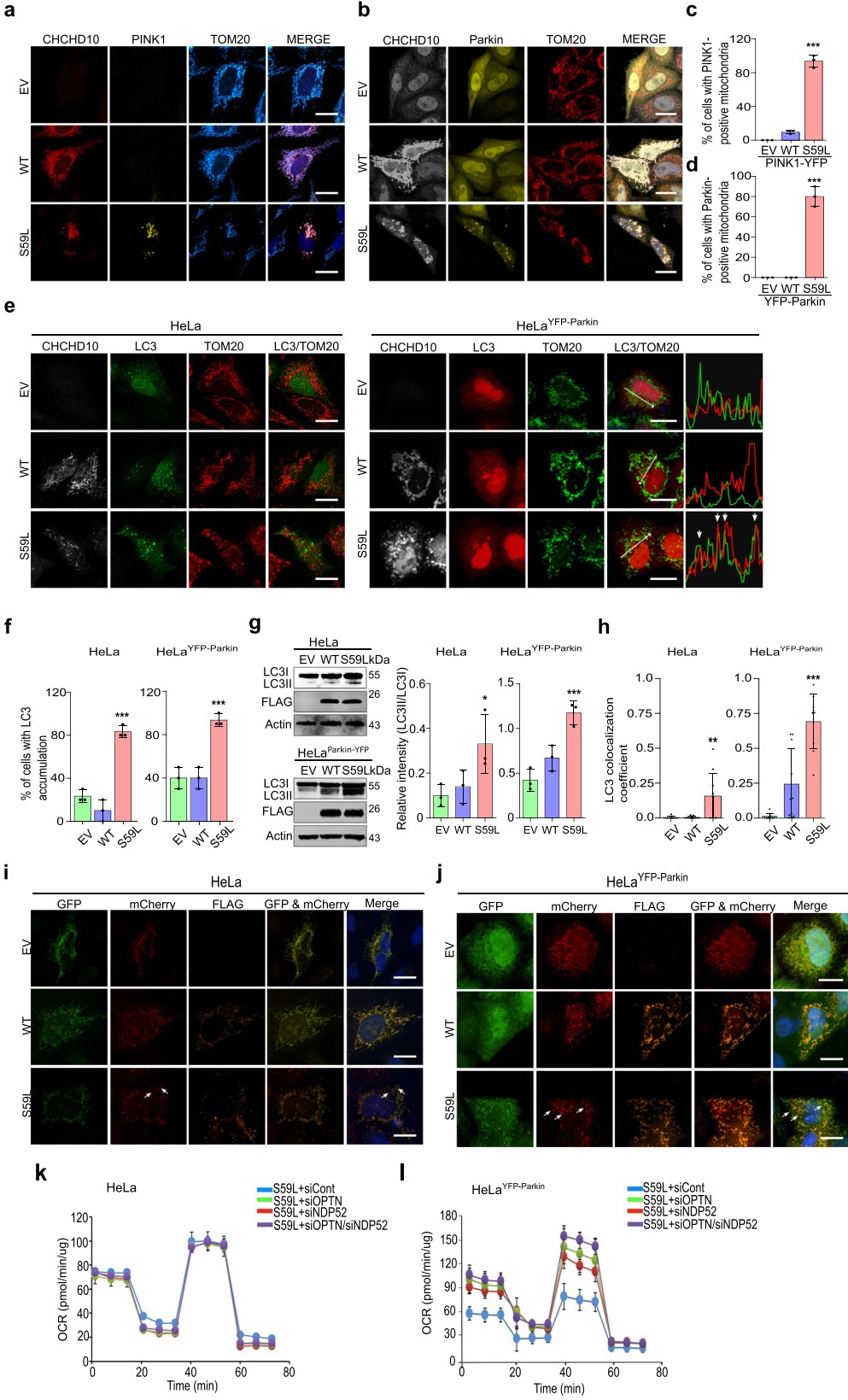

<sup>Parkin</sup> cells (Fig. 6k, l and Supplementary Fig. 6f–h). Therefore, the PINK1/parkin-mediated mitophagy pathway is one of the major toxicogenic pathways when Parkin exists. However, without Parkin, the mitophagic pathway is not essential to generate the substantial cell toxicity observed in *PRKN*-deficient HeLa cells,

indicating that PINK1 downstream factors other than Parkin can also mediate the cell toxicity independent of Parkin.

**Modulating PINK1 downstream pathways mitigates CHCHD10<sup>S59</sup>-induced cell toxicity.** To further define the

**Fig. 6 $CHCHD10^{S59L}$-induced toxicity is mediated by mitophagy when Parkin exists. a** HeLa cells and **b** HeLa$^{YFP-Parkin}$ cells were transfected with FLAG-tagged $CHCHD10^{S59L}$ and $PINK1-YFP$. The transfected HeLa cells were immunostained with antibodies against FLAG and TOM20 24 h after transfection. Scale bar = 20 μm. **c, d** Percentage of HeLa cells showing PINK1-YFP (**c**) and YFP-Parkin (**d**) accumulation in mitochondria; $n = 3$ independent experiments, ≥50 cells for each group. Data are shown as mean ± SD (one-way ANOVA and post hoc Dunnett test, two-sided, comparison with empty vector [EV]; ***$p$ = 3.7e − 07 (**c**) and 4.9e − 06 (**d**). **e** HeLa cells and HeLa$^{YFP-Parkin}$ cells were co-transfected with FLAG-tagged $CHCHD10^{S59L}$ and $GFP-LC3$ or $mCherry-LC3$, respectively. Representative images of transfected cells immunostained with antibodies against FLAG and TOM20. Graphs show the fluorescence intensity profiles of mCherry-LC3 (red), and TOM20 (green) along the region marked by an arrow in HeLa$^{parkin-YFP}$ merged image. Arrowheads indicate highly mCherry-LC3 and Tom20 merged region. Scale bar = 20 μm. **f** Percentage of cells showing GFP- or mCherry-LC3 accumulation. Data are shown as mean ± SD (one-way ANOVA and post hoc Dunnett test, two-sided, ***$p$ = 0.00011 for HeLa and 0.00056 in HeLa$^{YFP-parkin}$; $n = 3$ independent experiments, with ≥20 cells for each group). **g** Immunoblot of LC3 and CHCHD10$^{S59L}$ expression. Data are shown as mean ± SD (one-way ANOVA and post hoc Dunnett test, two-sided, *$p$ = 0.0387, ***$p$ = 0.00084; $n = 3$ independent experiments). **h** GFP-LC3 and mCherry-LC3 co-localization with TOM20 in HeLa cells and HeLa$^{YFP-Parkin}$ cells. Co-localization was measured by Pearson's correlation coefficient. Data are shown as mean ± SD (one-way ANOVA and post hoc Dunnett test, two-sided, **$p$ = 0.0022, ***$p$ = 1.8e − 08; $n = 10$ cells from three independent experiments). **i** HeLa cells and **j** HeLa$^{YFP-Parkin}$ cells were co-transfected with FLAG-tagged $CHCHD10^{WT}$ or $CHCHD10^{S59L}$ and $mito-QC$. Representative images of transfected cells (three independent experiments) immunostained with anti-FLAG antibody. Arrows indicate mitolysosomes (GFP-negative/mCherry-positive). Scale bar = 20 μm. **k** HeLa and **l** HeLa$^{YFP-Parkin}$ cells were transfected with siRNAs targeting $NDP52$ or/and $OPTN$. At 24 h after siRNA transfection, mitochondrial respiration was measured by Seahorse XF Cell Mito Stress tests. Representative graphs of a single experiment are shown (mean ± SD). Statistical analyses were performed with three independent experiments (see Supplementary Fig. 6g, h).

parkin-independent toxic pathway of PINK1, we tested whether the known downstream components ND42, sicily, Miro, and mitofilin modulate the $C2C10H^{S81L}$-induced rough eye phenotype (Supplementary Table 1). Only co-expression of PINK1 phosphorylation-null Mitofilin[50] showed mild but obvious rescue of the $C2C10H^{S81L}$-induced rough eye phenotype (Supplementary Fig. 7a), whereas RNAi-mediated knockdown of Mitofilin enhanced this phenotype (Supplementary Fig. 7b). Similarly, co-expression of PINK1 phosphorylation-null $MFN2^{S378}$ [51] with $CHCHD10^{S59L}$ in HeLa cells reduced the abnormal mitochondrial phenotype (Fig. 7a and Supplementary Fig. 7c). These data suggest that fine modulation of the downstream events caused by PINK1-mediated phosphorylation can be an effective therapeutic strategy. Indeed, recently developed MFN2 agonists reversing mitochondrial defects in a Charcot-Marie-Tooth disease type 2A model[51] also rescued $CHCHD10^{S59L}$-induced morphologic and functional mitochondrial defects (Fig. 7b, c and Supplementary Fig. 7d, e). We also observed the moderately elongated mitochondrial phenotype in the indirect flight muscles of $C2C10H^{S81L}$-expressing flies reared on two MFN2 agonists (Fig. 7d), which also increased ATP production in flies expressing $C2C10H^{S81L}$ (Fig. 7e) and in $C9orf72$-expressing flies with expanded GGGGCC repeats (Supplementary Fig. 7f)[52].

**Inhibition of PINK1 activity mitigates CHCHD10$^{S59}$-induced cell toxicity.** We hypothesized that inhibition of PINK1 kinase activity is a more effective and robust therapeutic strategy than regulating PINK1 expression level or its downstream substrates. Because of a lack of small-molecule inhibitors of PINK1, we used LALIGN to generate peptide sequences that may act as pseudo-substrate inhibitors of PINK1 and ubiquitin (Fig. 7f)[53,54]. When we treated HeLa cells with two putative peptide inhibitors of PINK1 via a protein delivery reagent, we observed that $CHCHD10^{S59L}$-mediated mitochondrial fragmentation was completely abolished in both pretreated and post-treated experiments (Fig. 7g and Supplementary Fig. 7g, h). Immunoblotting and immunostaining with an anti-phospho-ubiquitin antibody confirmed that ubiquitin phosphorylation was reduced by peptide treatment (Fig. 7h, i). Peptide treatment also reduced PINK1 accumulation, along with decreased phospho-ubiquitin staining (Fig. 7i), suggesting that a positive feedback mechanism amplifies $CHCHD10^{S59L}$-mediated cell toxicity via PINK1 accumulation (Fig. 7j).

**PINK1 knockdown does not affect TDP-43, but CHCHD10$^{WT}$ prevents PINK1 accumulation.** To determine whether the two axes of $CHCHD10^{S59L}$-induced cell toxicity exhibit crosstalk, we examined TDP-43 insolubility and mitochondrial localization after RNAi-mediated $PINK1$ knockdown in $CHCHD10^{S59L}$-transfected HeLa cells. Although $PINK1$ knockdown rescued the morphologic and functional defects in Drosophila and HeLa cells, insoluble TDP-43 levels and mitochondrial localization of TDP-43$^{WT}$ and TDP-43$^{A382T}$ were unaffected in HeLa cells (Fig. 8a, b). In addition, insoluble CHCHD10$^{S59L}$ and TDP-43 levels were not changed by $PINK1$ knockdown (Fig. 8c, d), suggesting that PINK1 accumulation in response to $CHCHD10^{S59L}$ expression is parallel or downstream (or both) of the TDP-43 pathway. $CHCHD10^{WT}$ co-expression with $CHCHD10^{S59L}$ reduced PINK1 accumulation (Fig. 8e). Intriguingly, $CHCHD10^{WT}$ expression also reduced PINK1 and parkin accumulation caused by mild CCCP treatment in HeLa$^{PINK1-V5}$ and HeLa$^{YFP-Parkin}$ cells, respectively (Fig. 8f and Supplementary Fig. 8a, b). Therefore, $CHCHD10^{WT}$ protected mitochondria not only through TDP-43 but also by preventing PINK1 accumulation. $CHCHD10^{WT}$ exerted a protective effect through both TDP-43 and PINK1 without modulating CHCHD10$^{S59L}$ insolubility (Fig. 9), suggesting that augmenting $CHCHD10$ expression or activity is also a promising therapeutic strategy, in addition to specifically blocking TDP-43 mitochondrial translocation and/or PINK1 activity.

## Discussion

Since the initial identification of the S59L substitution in CHCHD10, additional variants have been identified and suggested as pathogenic mediators of ALS-FTD, SMAJ, and mitochondrial myopathy. Despite efforts to elucidate their pathogenic mechanisms, many controversial findings suggest that mutations in $CHCHD10$ do not share a common disease-causing mechanism[1,11,12,19–22]. We found that only $CHCHD10^{S59L}$ induced consistent dominant cell toxicity in both Drosophila and HeLa cells[55].

Consistent with recent studies demonstrating a dominant mechanism of $CHCHD10^{S59L}$ in $CHCHD10^{KO}$ mice and $CHCHD10^{S59L}$ knockin mice[56,57], our own findings do not support simple loss-of-function or haploinsufficiency mechanisms. Woo et al. previously proposed a dominant-negative mechanism of $CHCHD10^{S59L}$ in Caenorhabditis elegans and mammalian systems. Interestingly, their observations are similar to our findings. However, our data support a dominant gain-of-toxic function mechanism for $CHCHD10^{S59L}$, whereas Woo et al. proposed

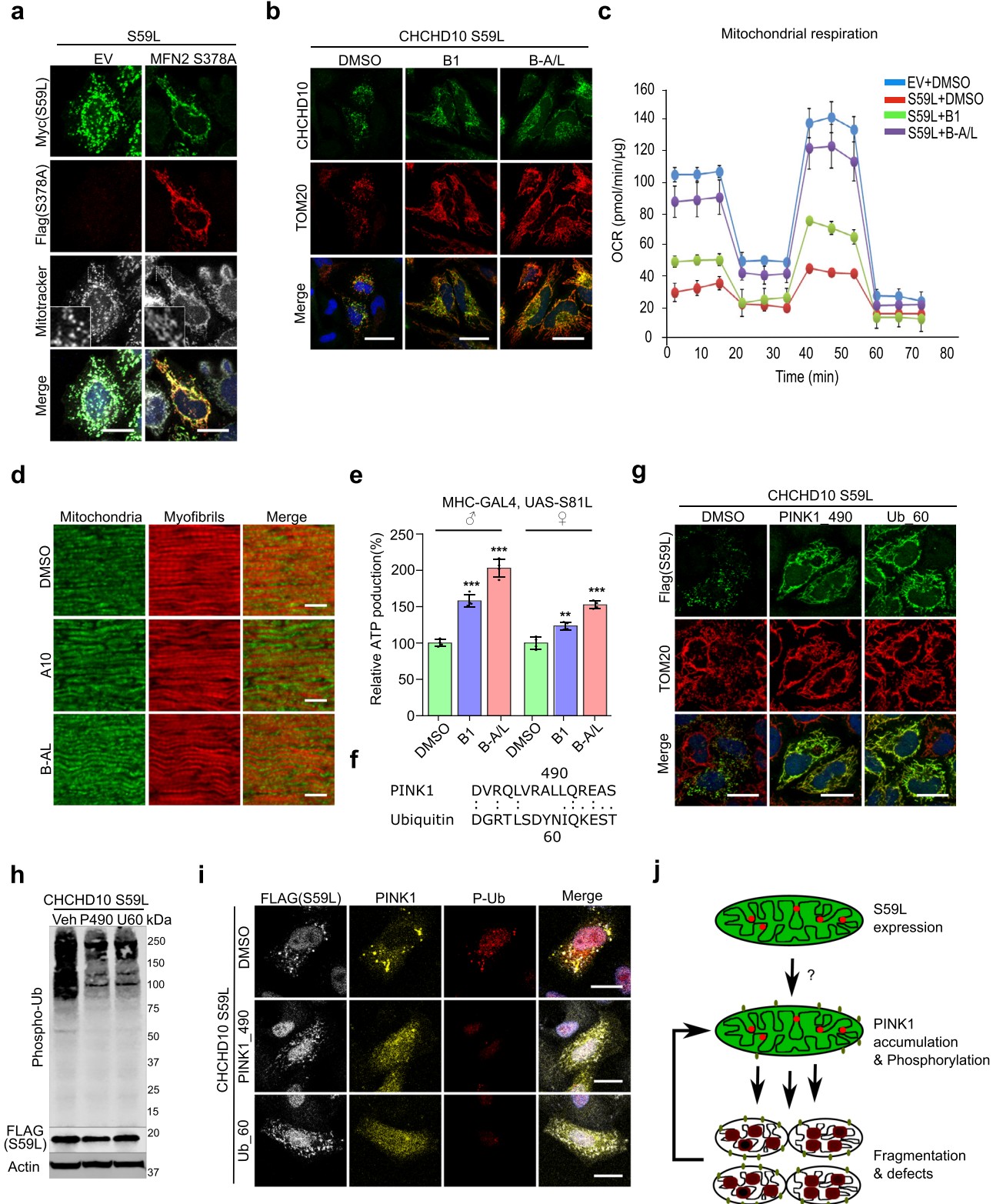

a dominant-negative mechanism, although they also observed results that can support a dominant gain-of-toxic function mechanism. Although these two mechanisms are not necessarily mutually exclusive, several aspects of our findings primarily support a dominant gain-of-toxic function mechanism. First, the *Drosophila* eye phenotypes driven by *C2C10H^{S59L}* did not differ between the *C2C10H^{null}* or *C2C10H^{WT}* background, indicating that the mutations exert a dominant gain-of-toxic function rather

than suppressing WT activity as a dominant negative. Second, *CHCHD10^{S59L}* is toxic regardless of the presence of *CHCHD10^{WT}* (and *CHCHD2^{WT}*). Third, we confirmed the genetic modifiers modulating the dominant gain-of-toxic function in *Drosophila* in mammalian cells, as well. Fourth, parkin accumulation did not occur in *CHCHD10^{KO}* HeLa^{YFP-Parkin} cells, but it did occur with *CHCHD10^{S59L}* expression[55], indicating that the cell toxicity caused by reduced *CHCHD10* differs from that of *CHCHD10^{S59L}*.

**Fig. 7 CHCHD10$^{S59L}$-induced toxicity is mitigated by inhibiting PINK1 activity and modulating its target. a** HeLa cells were transfected with Myc-tagged *CHCHD10$^{S59L}$* and FLAG-tagged *MFN2$^{S378A}$*. Mitochondria were stained with MitoTracker (deep red FM) and antibodies against Myc and FLAG 24 h after transfection. White outlined boxes (MitoTracker) were magnified in the left corner. Representative images from three independent experiments are shown. Scale bar = 20 μm. **b** HeLa cells were transfected with FLAG-tagged *CHCHD10$^{S59L}$* and treated with MFN2 agonists B1 (50 nM) and B-A/L (5 nM) for 24 h. Cells were immunostained with antibodies against FLAG and TOM20. Representative images from three independent experiments are shown. Scale bar = 20 μm. **c** Mitochondrial respiration was measured by Seahorse XF Cell Mito Stress tests after mitofusin agonists treatment. A representative graph of a single experiment is shown (mean ± SD). Actual statistical analyses were performed with four independent experiments (see Supplementary Figs. 7e, 9l). **d** Indirect flight muscles from 10-day-old adult flies fed with B1 and B-A/L (each 10 μM) were stained with streptavidin–Alexa Fluor 488 and phalloidin–Alexa Fluor 594. DMSO was used as a vehicle. Representative images from three independent experiments are shown. Scale bar = 10 μm. **e** ATP levels in thoraxes from the indicated genotypes (aged 10 days) were measured. Data are mean ± SD (one-way ANOVA and *post hoc* Dunnett test, two-sided, ***$p$ = 0.00013 and 1.8e − 05 in male, **$p$ = 0.0012 and ***$p$ = 0.00027 in female for B1 and B-A/L, respectively, compared to DMSO; $n$ = 4 independent experiments). **f** Peptide sequences of two PINK1 inhibitors. **g** HeLa cells were treated with PINK1_490 (0.15 μg/ml) or Ub_60 (0.15 μg/ml) and transfected with FLAG-tagged *CHCHD10$^{S59L}$*. DMSO was used as a vehicle. Cells were immunostained with antibodies against FLAG and TOM20 24 h after *CHCHD10$^{S59L}$* transfection. Representative images from three independent experiments are shown. Scale bar = 20 μm. **h** HeLa cells treated with peptide inhibitors and transfected with FLAG-tagged *CHCHD10$^{S59L}$* were analyzed with an anti-phospho-ubiquitin antibody. Representative images from three independent experiments are shown. **i** HeLa cells were treated with peptide inhibitors followed by transfection of FLAG-tagged *CHCHD10$^{S59L}$* and *YFP-PINK1*. The cells were immunostained with antibodies against FLAG and TOM20. Representative images from three independent experiments are shown. Scale bar = 20 μm. **j** A schematic depicting the positive feedback mechanism of *CHCHD10$^{S59L}$*-mediated toxicity through PINK1 accumulation.

Fifth, CHCHD10$^{WT}$ did not accumulate in CHCHD10$^{S59L}$ aggregate-like structures. Sixth, *CHCHD10$^{WT}$* overexpression did not affect CHCHD10$^{S59L}$ aggregate formation and insolubility. Therefore, both CHCHD10$^{WT}$ and CHCHD10$^{S59L}$ are involved in the same pathway, independently in reverse directions, because CHCHD10 has a protective role in mitochondria that is also protective for *CHCHD10$^{S59L}$*-driven dominant cell toxicity, which occurred independently of disrupting WT activity. However, we cannot completely rule out the contribution of loss-of-function- or dominant-negative-like effects. We anticipate that some effect of reduced WT activity occurs in disease pathogenesis. Our data suggest that the dominant cell toxicity of *CHCHD10$^{S59L}$* can be mitigated by co-expressing similar levels of CHCHD10$^{WT}$. Therefore, it is possible that a heterozygous *CHCHD10$^{S59L}$* mutation does not induce severe degeneration, as long as WT activity blocks mutant cell toxicity. However, an age-dependent reduction of WT function or a change in the ratio of WT to mutant expression may trigger disease symptoms later. Taken together, multiple or possibly mixed mechanisms may exist, and *CHCHD10$^{S59L}$* may cause a strong dominant phenotype that can be successfully modeled in both *Drosophila* and HeLa cells.

Our efforts to define the downstream pathways of dominant *CHCHD10$^{S59L}$*-mediated cell toxicity yielded two axes and multiple molecular targets that can be therapeutically modulated (Fig. 9). The mitochondrial translocation of TDP-43 is a toxicity-generating mechanism in *CHCHD10$^{S59L}$*-expressing cells. Wang et al.[25] demonstrated that excess TDP-43 mitochondrial translocation induces mitochondrial dysfunction, and blocking this translocation abolishes mitochondrial toxicity. *CHCHD10$^{S59L}$* induced TDP-43 mitochondrial translocation and inhibiting this translocation mitigated *CHCHD10$^{S59L}$*-induced mitochondrial abnormalities. The association between TDP-43 and CHCHD10 was previously proposed[19], and Wang et al. raised a question for the importance of the mitochondrial TDP-43. In support of this, we showed that CHCHD10$^{S59L}$ bound to TDP-43 more greatly than did CHCHD10$^{WT}$ and that inhibition of TDP-43 mitochondrial translocation mitigated the *CHCHD10$^{S59L}$*-induced phenotype. In addition, the effects of CHCHD10$^{WT}$ on TDP-43 insolubility and translocation generally support that TDP-43 is a key effector generating mitochondrial toxicity in ALS-FTD.

We identified *PINK1* and *PRKN* as strong genetic modifiers of *C2C10H$^{S81L}$*-mediated cell toxicity. PINK1/parkin-mediated pathways are generally regarded as protective for cells by removing damaged mitochondria. However, *CHCHD10$^{S59L}$* expression induced PINK1 stabilization in mitochondria, and genetic/pharmacologic inhibition of PINK1 clearly mitigated *CHCHD10$^{S59L}$*-induced cell toxicity. Previous reports showed that reducing PINK1 or parkin-mediated pathways are beneficial in in vivo disease models of *SOD1*, *FUS*, and *TARDBP* mutations[58–60]. We demonstrated that MFN2 agonists enhanced ATP production in flies expressing the GGGGCC repeats of *C9ORF72*.

Two downstream phosphorylation substrates of PINK1, MFN, and mitofilin mediated *CHCHD10$^{S59L}$*-induced cell toxicity. Although fusion activity in *CHCHD10$^{S59L}$*-expressing cells is not altered[1], our data indicate that PINK1 accumulation and subsequent inactivation of MFN by phosphorylation is responsible for fragmented mitochondria and the respiratory defects caused by *CHCHD10$^{S59L}$*. An MFN2 agonist developed for CMT2 was also effective for *CHCHD10$^{S59L}$*-mediated cell toxicity in this context and may also be effective for the *CHCHD10$^{G66V}$* mutation causing SMAJ or CMT2. Overexpression of a PINK1 phosphorylation-null *Mitofilin* mutant also rescued *C2C10H$^{S81L}$*-mediated cell toxicity, suggesting that deformation of the MICOS complex was not based on a direct interaction, but through phosphorylation by PINK1. Therefore, the degree of PINK1 stabilization may correspond with the phenotypic severity of mutant *CHCHD10*.

While this manuscript was under review for publication, two independent studies reported CHCHD10$^{S59L}$-mediated OMA1 peptidase activation[61], subsequent degradation of OPA1 resulting in mitochondrial fragmentation[61,62], and a protective effect of CHCHD10$^{WT}$ against TDP-43 mitochondrial accumulation[62]. Because OMA1 and PINK1 can be activated in the same experimental conditions, elucidating the association between the OMA1–OPA1 pathway and the PINK1-mediated pathway in CHCHD10$^{S59L}$-induced pathogenesis will be worthwhile. Although we did not observe any meaningful protective effects of CHCHD10$^{WT}$ against toxic TDP-43$^{M337V}$ in *Drosophila*, this may be due to the strong overexpression of *TDP-43$^{M337V}$*. Therefore, investigating the protective role of CHCHD10 in ALS-FTD and other degenerative diseases with mitochondrial defects is important. In contrast with our findings, two studies reported that their findings were also observed in multiple *CHCHD2* and *CHCHD10* double-knockout models[61] or by *CHCHD10* knockdown in cell culture[62]. These findings combined with our data suggest that *CHCHD10$^{S59L}$*-induced gain-of-toxic function, partial loss of *CHCHD10$^{WT}$*, and dominant-negative-like inhibition of CHCHD2 may coexist during disease pathogenesis or individually contribute to specific aspects of disease pathogenesis.

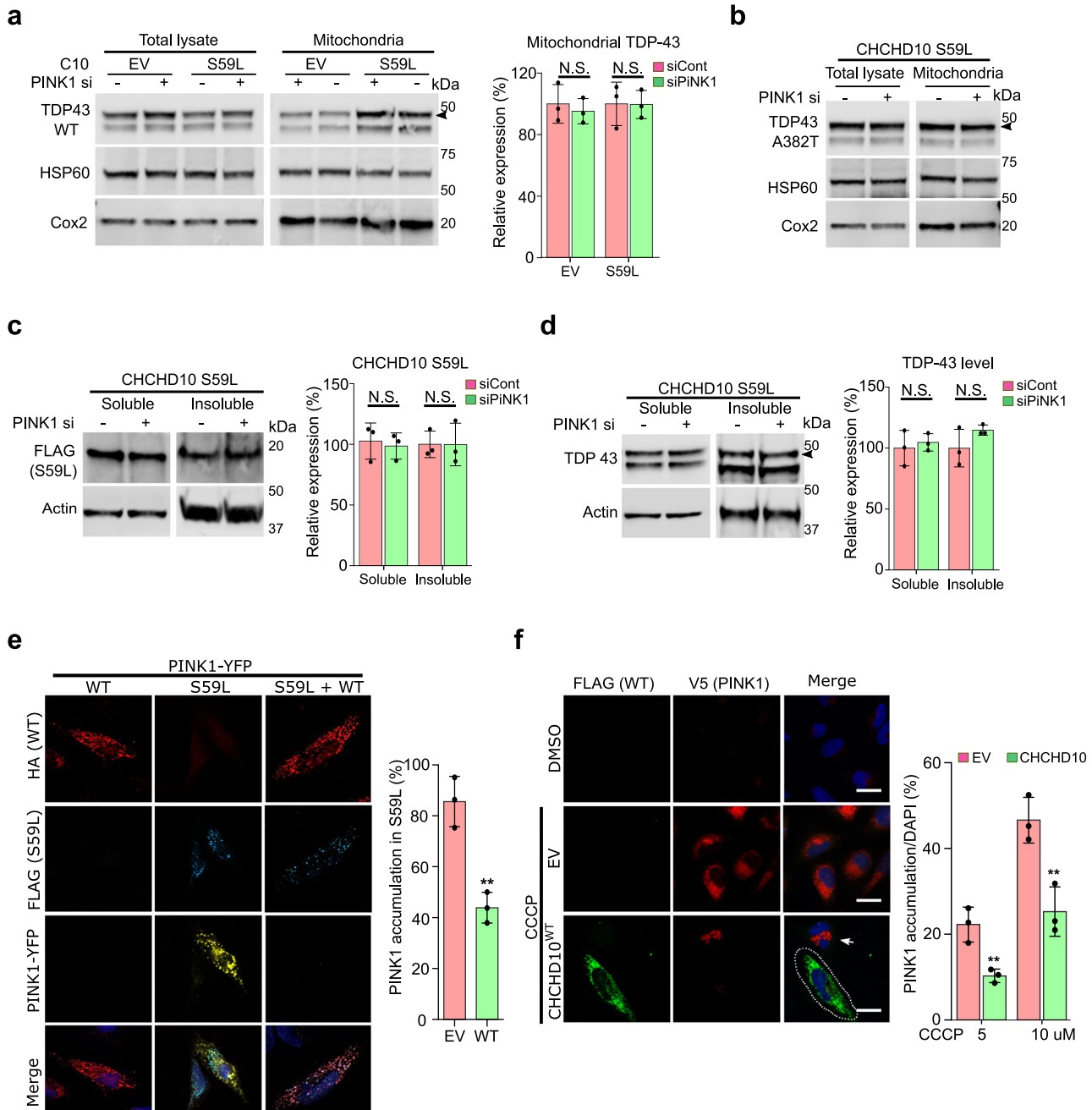

**Fig. 8 Dominant toxicity of *CHCHD10^S59L* is mediated independently by TDP-43 and PINK1 signaling. a** HeLa cells transfected with *PINK1* siRNA were co-transfected with empty vector (EV) or FLAG-tagged *CHCHD10^S59L* and FLAG-tagged *TARDBP^WT*. Fractionated mitochondria were analyzed with anti-TDP-43 (arrowhead indicates transfected TDP-43) or anti-HSP60 and anti-Cox2 (loading controls) antibodies. Data are shown as mean ± SD (two-sided *t* test, NS = not significant; *n* = 4 independent experiments). **b** HeLa cells transfected with *PINK1* siRNA were co-transfected with empty vector (EV) or FLAG-tagged *CHCHD10^S59L* and *TARDBP^A382T*. Fractionated mitochondria were analyzed with anti-TDP-43 (arrowhead indicates transfected TDP-43) or anti-HSP60 and anti-Cox2 (loading controls) antibodies. **c** HeLa cells transfected with *PINK1* siRNA were transfected with FLAG-tagged *CHCHD10^S59L*. RIPA-soluble and insoluble fractions were analyzed with anti-FLAG and anti-actin (loading control) antibodies. Data are shown as mean ± SD (two-sided *t* test, NS = not significant; *n* = 3 independent experiments). **d** HeLa cells transfected with *PINK1* siRNA were co-transfected with FLAG-tagged *CHCHD10^S59L* and *TARDBP*. RIPA-soluble and insoluble fractions were analyzed with anti-TDP-43 and anti-actin (loading control) antibodies. Data are shown as mean ± SD (two-sided *t* test, NS = not significant; *n* = 4 independent experiments). **e** HeLa cells transfected with *PINK1-YFP*, FLAG-tagged *CHCHD10^S59L*, and HA-tagged *CHCHD10^WT* or EV were visualized with anti-FLAG (blue) and anti-HA (red) antibodies and YFP (yellow) after 24 h of transfection. PINK1-positive cells with/without CHCHD10^WT in CHCHD10^S59L-expressing cells were counted (two-sided *t* test, **p = 0.00335; *n* = 3 independent experiments, >50 cells for each group). Scale bar = 20 μm. **f** HeLa^PINK1-V5-His cells transfected with EV or *CHCHD10^WT* were treated with CCCP (10 μM) for 6 h. Cells were immunostained with anti-FLAG (green) and anti-V5 (red) antibodies to visualize CHCHD10 and PINK1, respectively. Arrow indicates PINK1 accumulated in a non-transfected cell neighboring a *CHCHD10*-transfected cell (white dashed line). The percentage of PINK1-positive cells from the empty vector (EV) or *CHCHD10^WT*-transfected cells were calculated after 5 or 10 μM CCCP treatment for 6 h. Data are shown as mean ± SD (two-sided *t* test, **p = 0.009055 for 5 μM and 0.009203 for 10 μM CCCP treatment; *n* = 3 independent experiments, >200 cells for each group). Scale bar = 20 μm.

**a**

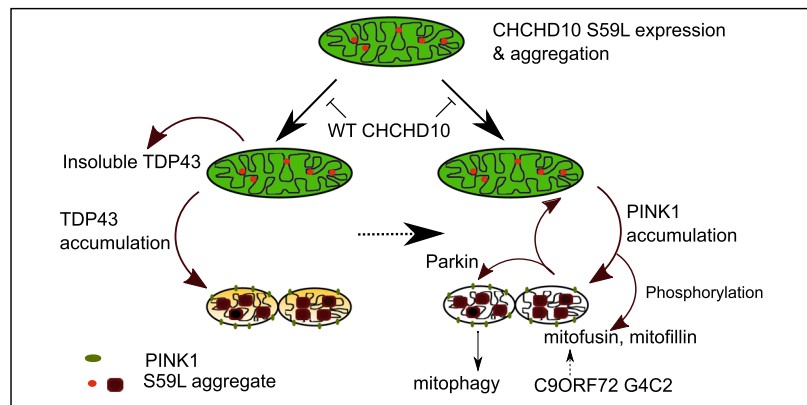

**Fig. 9 Two axes of CHCHD10$^{S59L}$-induced toxicity. a** CHCHD10$^{S59L}$ can generate cellular toxicity with or without WT CHCHD10 and its close paralog CHCHD2. The toxicity of the gain-of-function CHCHD10$^{S59L}$ mutant is mediated by two different pathways. TDP-43 axis: Upon *CHCHD10$^{S59L}$* expression, TDP-43 becomes more insoluble in the cytoplasm, probably due to mitochondria-induced stresses[56,57] and accumulates in mitochondria via increased binding with CHCHD10$^{S59L}$. The toxicity generated by mitochondrial TDP-43 can be mitigated by inhibitors for TDP-43 mitochondrial translocation. PINK1 axis: in response to *CHCHD10$^{S59L}$* expression, PINK1 is stabilized and accumulates on mitochondria in an uncontrolled manner. PINK1 phosphorylates its downstream targets, including mitofusin, mitofilin, and Parkin, which results in significant mitochondrial fragmentation, maladaptive cristae formation, and excessive mitophagy. A positive feedback mechanism via PINK1 accumulation may accelerate these processes. Genetic or chemical modulation of this pathway can mitigate CHCHD10$^{S59L}$-induced cellular toxicity and C9orf72 G4C2 repeat-induced degeneration. Interestingly, WT CHCHD10 expression reduces CHCHD10$^{S59L}$-mediated cellular toxicity through both TDP-43 and PINK1 axes.

Augmenting CHCHD10$^{WT}$ activity may be a promising therapeutic strategy, regardless of specific *CHCHD10* mutations. *CHCHD10$^{WT}$* expression is increased in response to various stresses[63]. We and others observed that *CHCHD10$^{WT}$* expression not only rescued mutant phenotypes but also increased mitochondrial length and respiratory activity when it was expressed alone in both *Drosophila* and HeLa cells. Although the mechanism by which CHCHD10 enhances mitochondrial function and reduced PINK1-mediated cell toxicity is not clear, our findings support that pharmacologic or epigenetic augmentation of *CHCHD10* expression may mitigate such mitochondrial defects.

## Methods

**Chemicals and peptides.** CCCP was purchased from Sigma-Aldrich. The MFN agonists B1 and B/A–L were described previously[51]. ScPM and PM1 peptides for TDP-43 were kindly provided by Xinglong Wang. Pink1_490 and Ub_60 peptides were synthesized by Thermo Scientific with N-terminal TAMRA 5/6 and C-terminal amidation.

**DNA constructs.** All complementary DNAs (cDNAs) for human *CHCHD10$^{WT}$* and variants were synthesized and inserted in the pcDNA3 vector containing a FLAG, HA, or Myc tag by Genescripts. *mTagRFP-T-Mito-7* (#58023)[64], *TDP43tdTOMATO-HA* (#28205)[65], *EGFP-LC3* (#21073)[66], *mCherry-LC3B* (#40827)[67], *pEYFP-C1-DRP1* (#45160)[68], *MFN2-YFP* (#28010)[69], *pEYFP-N1-PINK1* (#101874)[70], and *LAMP1-mGFP* (#34831)[71] plasmids were obtained from Addgene. *FLAG-MFN2$^{WT}$*, *FLAG-MFN2$^{S378A}$*, and *FLAG-MFN2$^{S378D}$* were described previously[51]. *TARDBP$^{WT}$*, *TARDBP$^{G298S}$*, *TARDBP$^{A315T}$*, and *TARDBP$^{A382T}$* plasmids were gifts from Xinglong Wang. Mito-QC (pBabe.hygro-mcherry-GFP fis 101-152) was provided by Ian Ganley.

**RNA extraction, cDNA synthesis, and quantitative reverse transcription-PCR (qRT-PCR).** Total RNAs were extracted from patient fibroblasts using TRIzol reagent (Thermo Fisher Scientific). Prior to reverse transcription, residual genomic DNA was removed with DNase I (Thermo Fisher Scientific). The cDNA was then reverse transcribed using transcription first-strand cDNA Synthesis Kit (Roche Applied Science) with 1 μg total RNA and oligo-dT as primer. All PCRs were performed in triplicate. QRT-PCR was carried out using SYBR Green Master Mix (Roche Applied Science) on a Light Cycler LC480. Results were normalized to β-2 macroglubulin or 28S genes. All primer sequences are included in Supplementary Table 2.

**Cell culture and transfection.** HeLa$^{YFP-Parkin}$, HeLa with stable *PINK1* expression (HeLa$^{Pink1-V5/His}$), and a control host HeLa cell line were kind gifts from Richard

Youle. *PINK1$^{KO}$* HeLa cells and a matched control HeLa cell line were kindly provided by Wade Harper. HEK293T and SH-SY5Y cells were purchased from ATCC. Cells were maintained in culture in Dulbecco's modified Eagle's medium (Gibco) or DMEM/F12 50/50 (Corning) supplemented with 10% fetal bovine serum (Gibco), 1× penicillin/streptomycin (Invitrogen), and GlutaMax-1× (Gibco). Cells were transfected using FuGENE6 transfection reagent (Promega), Lipofectamine 3000 (Invitrogen), or jetPrime (Polyplus). RNAi-mediated knockdown of target genes was performed by transfection of the ON-TARGET plus-SMART pool siRNAs (Dharmacon) by using Lipofectamine RNAiMAX (Invitrogen) for the following genes: non-targeting control, *PINK1*, *MFN1*, *MFN2*, *OPTN*, *NDP52*, and *DRP1*. Pink1_490 and Ub_60 peptides were delivered by using PULSin protein delivery reagent (Polyplus).

**CRISPR/Cas9-mediated gene editing and generation of cell lines.** Each two plasmid vectors [pSpCas9(BB)-2A-Puro (XP459) V2.0] containing a single guide RNA (sgRNA) oligomer for *CHCHD10* and *CHCHD2* targeting near the N-terminal region were purchased from GenScript. Sequences for the sgRNA targeting the N-terminal of *CHCHD10* are as follows: 5′-GCCTCGGGGAAGCCG-CAGCG-3′ and 5′-GCCGCGCTGCGGCTTCCCCG-3′ and for the *CHCHD2* are as follows: 5′-CCGAAGCCGCACCTCCCGCA-3′ and 5′-CTCAGATGA-GAGCTGCACCC-3′. The plasmids were transfected into HeLa and HeLa$^{YFP-Parkin}$ cells with jetPrime reagent. After 24 h, the transfected cells were detached with trypsin and plated individually in 96-well round-bottom plates. Cells were then expanded, and single clones were analyzed by immunofluorescence and immunoblotting to screen protein levels in *CHCHD10* knockout and *CHCHD10/2* double knockout.

**Antibodies and immunoblotting.** The following primary antibodies were used: FLAG (Sigma and Proteintech, 1:1000), HA (Cell Signaling and Proteintech, 1:1000), Myc (Proteintech, 1:1000), CHCHD10 (Proteintech, 1:1000), CHCHD2 (Proteintech, 1:1000), TOM20 (Cell Signaling and Santa Cruz Biotechnology, 1:1000), TDP-43 (Proteintech and Santa Cruz Biotechnology, 1:1000), HSP60 (Cell Signaling Technology, 1:1000), COX2 (Abcam, 1:2000), LC3B (Cell Signaling Technology, 1:1000), phospho-ubiquitin (EMD Millipore, 1:1000), Actin (Santa Cruz Biotechnology and Proteintech, 1:3000), PINK1 (Santa Cruz Biotechnology and Novus, 1:1000), DRP1 (Cell Signaling Technology, 1:1000), MFN1 (Cell Signaling Technology, 1:1000), MFN2 (Cell Signaling Technology, 1:1000), NDP52 (Proteintech, 1:1000), and OPTN (Proteintech, 1:1000). Samples were collected and lysed in RIPA buffer (Cell Signaling Technology) containing a protease inhibitor cocktail (Sigma) and subsequently separated by sodium dodecyl sulfate-polyacrylamide gel electrophoresis (SDS-PAGE) after measuring protein concentration by bicinchoninic acid (BCA) (Pierce). Immunoblots were visualized and analyzed with the Odyssey FC System (LI-COR).

**Co-immunoprecipitation.** HEK293T cells transfected FLAG-tagged CHCHD10 WT or S59L and TDP-43-tomato-HA or HA-tagged CHCHD10 WT or S59L and

TDP-43-FLAG were solubilized with NP-40 lysis buffer (20 mM Tris, 137 mM NaCl, 1% NP-40, 2 mM EDTA with protein inhibitor cocktail). After sonication on ice, lysates were collected by centrifuge. An equal amount of protein lysates were incubated with anti-FLAG M2 or anti-HA affinity gel (Sigma) overnight at 4 °C, washed three times with phosphate-buffered saline (PBS) containing 1% Tween-20, and then resuspended the pellet with 2× LDS sample buffer. The precipitates were subjected SDS-PAGE and analyzed with the Odyssey FC System (LI-COR).

**Solubility and biochemical analyses.** Transfected cells were washed twice with PBS, lysed in cold RIPA buffer [50 mM Tris (pH 7.5), 150 mM NaCl, 1% Triton X-100, 0.5% sodium deoxycholate, 0.1% SDS, and 1 mM EDTA) and sonicated on ice. Cellular lysates cleared by ultracentrifugation at $100,000 \times g$ for 30 min at 4 °C to prepare RIPA-soluble fractions. RIPA-insoluble pellets were washed twice with protease inhibitors in cold PBS, sonicated, and recentrifuged. RIPA-insoluble proteins were extracted with urea buffer [7 M urea, 2 M thiourea, 4% CHAPS, 30 mM Tris (pH 8.5)], sonicated, and centrifuged at $100,000 \times g$ for 30 min at 22 °C. Supernatants from the first centrifugation were analyzed for soluble fractions. Total protein in each sample was measured by the BCA method and resolved on ExpressPlus PAGE 4–20% gels (GenScript).

**Cellular and mitochondrial fractionation.** Mitochondria were isolated from cells with a Mitochondrial Isolation Kit (Thermo Scientific) by using Dounce homogenizers. A mixture of cytosolic and mitochondrial fractions was obtained after low-speed centrifugation at $500 \times g$ for 15 min at 4 °C. Mitochondrial-enriched pellets were collected at $3000 \times g$ for 10 min at 4 °C. Cytosolic supernatants were obtained and cleared by centrifugation at $12,000 \times g$ for 10 min at 4 °C. Cross-contamination between fractions was analyzed with anti-tubulin and anti-HSP60 antibodies for each cytosolic and mitochondrial compartment.

**Mitochondrial respiratory activity assay.** Mitochondrial respiration in HeLa, $CHCHD10^{KO}$ HeLa, $CHCHD2/10^{DKO}$ HeLa, HeLa$^{YFP-Parkin}$, and $CHCHD10^{KO}$ HeLa$^{YFP-Parkin}$ cells were measured by using the Seahorse Extracellular Flux Analyzer XFp (Agilent Technologies) with the XF Cell Mito Stress Test Kit (Agilent Technologies). Transfected cells ($1 \times 10^4$) were counted using ADAM-MC2 (NanoEntek) and plated into V3-PS 96-well plates the day before performing the assay. Assay media were supplemented with 1 mM pyruvate, 2 mM glutamine, and 15 mM glucose. Standard mitochondria stress tests were performed by first measuring basal values followed by measurements after sequential addition of 1 μM oligomycin, 0.5 μM FCCP, and 0.5 μM rotenone/antimycin A. After the assay, protein concentrations of each well were determined via BCA assay and used to normalize oxygen consumption rate values. The Seahorse assay parameters and experimental outcomes are summarized in Supplementary Table 3.

**Immunofluorescence staining and imaging.** The following primary antibodies were used: FLAG (Sigma and Proteintech, 1:200), FLAG-Alexa Fluor 488 (Invitrogen, 1:250), CHCHD10 (Proteintech, 1:200), CHCHD2 (Proteintech, 1:200), V5 (Life Technology, 1:200), TOM20 (Cell Signaling and Santa Cruz Biotechnology, 1:250), TDP-43 (Proteintech, 1:200), LAMP1 (Cell Signaling Technology, 1:200), and phospho-ubiquitin (EMD Millipore, 1:100). Cells were plated on 4-well chamber slides (Lab-Tek), fixed with 4% paraformaldehyde in PBS (EMS Millipore), permeabilized with 0.1% Triton X-100, and blocked with 5% bovine serum albumin (BSA) in PBS. Primary antibodies were diluted in 5% BSA in PBS and incubated overnight at 4 °C. Samples were then rinsed three times with PBS-Tween-20 and incubated with secondary antibodies for 1.5 h at room temperature. Coverslips were mounted onto microscope slides with Prolong Diamond Antifade Reagent with DAPI (4′,6-diamidino-2-phenylindole) (Invitrogen). Samples were observed with an LSM 710 confocal microscope (Carl Zeiss) or Nikon Crest X-light 2 spinning disc confocal microscope (Nikon). Co-localization was analyzed with co-localization tool of ZEN software (Carl Zeiss). For patient primary fibroblast imaging, cells were incubated in a 100 nM solution of MitoTracker red (Invitrogen) for 15 min at 37 °C. The medium was replaced by DMEM supplemented with 10% FBS and penicillin (100 U/ml)/streptomycin (0.1 mg/ml) and cells were incubated for 2 h at 37 °C, washed in PBS, and then fixed. Specimens were analyzed using a Zeiss LSM880 confocal laser-scanning microscope. The images were deconvolved with Huygens Essential Software (Scientific Volume Imaging). The deconvolved images were used for quantitative mitochondrial network analysis with Huygens Essential Software. Mitochondrial network length was quantified for 48–116 randomly selected individual cells.

**Generation of Drosophila lines.** To generate transgenic Drosophila lines carrying C2C10H, codon-optimized CHCHD10, and their variants, all cDNAs were synthesized with or without a C-terminal FLAG tag and cloned into pUASTattB by GenScript. Transgenic Drosophila lines were generated by BestGene with standard ΦC31 integrase-mediated transgenesis into the attP2 site on chromosome 3 (locus 68A4), the attP40 site on chromosome 2 (locus 25C6), or the VK27 site on chromosome 3 (locus 89E11).

**Drosophila genetics.** Fly cultures and crosses were performed on standard fly food (Genesee Scientific) and raised at 25 °C. The GMR-GAL4 and OK371-GAL4 drivers were obtained from the Bloomington Stock Center. MHC-GAL4 was a gift from Guillermo Marques. All tested RNAi, deficiency, duplication, and classical allele lines are listed in Supplementary Table 1.

**Drosophila neuronal phenotype analysis.** To quantify the synaptic bouton numbers of Drosophila larvae[44], ten third-instar larvae for each group were collected and pinned on Sylgard dishes with tungsten pins. After dissecting the dorsal sides of larvae and removing the trachea and internal organs, the larvae were fixed with 4% paraformaldehyde in PBS. Larvae were stained with a presynaptic anti-horseradish peroxidase antibody (1:200, Jackson Immunoresearch) and postsynaptic anti-discs large antibody (1:200, Developmental Studies Hybridoma Bank). The NMJs at muscle 4 were used for all analyses.

**Drosophila behavioral assay.** Six to seven wandering third-instar larvae from each group were collected, washed, and placed onto a 3% agarose gel in a 10-cm dish. Larval crawling was recorded by a digital camera for 30 s. Moving distances of individual larva were tracked and measured using Tracker software (Open Source Physics, 5.0.7). To test flight ability, 20 flies from each group were funneled into a 500-ml glass cylinder. The distribution of flies in the cylinder was recorded by a digital camera, and the average scores from five independent experiments were calculated.

**Drosophila immunoblotting.** Heads of 2-day-old adults were prepared and ground in LDS sample buffer by using a motor-driven plastic pestle homogenizer and centrifuged at $16,000 \times g$ for 10 min. Supernatants were boiled for 10 min and analyzed by immunoblotting with the Odyssey FC system (LI-COR). Proteins were separated with 4–20% ExpressPlus PAGE Gels (GenScript), transferred onto nitrocellulose membranes, and probed with anti-FLAG. Anti-actin antibody was used as a loading control.

**Drosophila adult muscle preparation and immunohistochemistry.** To assess the mitochondrial morphology and sarcomere structure of the indirect flight muscles[44], dissected adult flies were fixed with 4% paraformaldehyde in PBS for 1 h, embedded in OCT compound (Fisher Scientific), and frozen with liquid nitrogen or dry ice. Samples were sectioned by a cryomicrotome (Leica). After fixing with 4% paraformaldehyde in PBS, samples were permeabilized with 0.2% Triton X-100 buffer in PBS and blocked with 5% BSA solution in PBS for 1 h. Samples were incubated with streptavidin–Alexa Fluor 488 (Invitrogen) and phalloidin–Alexa Fluor 594 (Invitrogen) overnight at 4 °C for mitochondrial and muscle staining, respectively. Samples were mounted with Prolong Diamond Antifade Reagent with DAPI and imaged with an LSM 710 confocal microscope (Carl Zeiss) with ×63 magnification.

**Drosophila ATP assay.** Fly thoraxes from each group were collected and homogenized in 20 μl of homogenization buffer [100 mM Tris, 4 mM EDTA, and 6 M guanidine-HCL (pH 7.8)] and centrifuged at $16,000 \times g$ for 10 min. The supernatants were diluted 1:200 and 1:10 with deionized water and subjected to ATP concentration and protein concentration measurements, respectively. ATP concentration was determined by using the CellTiter Glo Luminescent Cell Viability Assay Kit (Promega) and normalized to total protein.

**Image analysis and statistical analysis.** Mitochondrial branch length was measured by using the MiNA toolset combined with the ImageJ software[72]. Statistical analysis was performed with Prism5 (GraphPad) software. For Drosophila eye quantification, we implemented ilastik, which is an interactively supervised machine learning-based tool for various bioimage analysis to classify ommatidia in fly eye images[73]. Then we used Flynotyper[31], which is a computational tool for assessing the Drosophila eye morphological defects, to calculate phenotypic scores (P-scores), which indicate the irregularity of ommatidial arrangement. Eye images for group A (Fig. 1b, Supplementary Fig. 1d, Fig. 3a, Supplementary Fig. 3d, and Fig. 5a) were captured with a Leica M205C stereomicroscope equipped with a ring light and a Leica DFC320 digital camera. Eye images for group B (Fig. 1c, Fig. 3f, Supplementary Fig. 3a, e, and Supplementary Fig. 5b, 7a, b) were captured with a Nikon SMZ1500 stereomicroscope equipped with a ring light and a Nikon DXM1200 digital camera. Although we applied the same criteria to analyze all eye images with ilastik and Flynotyper, the P-scores can be compared only within each experiment because of differences in the equipment, setting, and researchers. For statistical analyses of Seahorse Assays, Z-scores were calculated using OCR values from each plate. Z-score = (OCR value – mean of OCR values)/standard deviation of OCR values. Student's t test (two-tailed), analysis of variance (ANOVA), and post hoc analysis (Tukey's or Dunnett's tests, two-tailed) were used to test statistical significance. All p values ≤0.05 were considered statistically significant.

**Reporting summary**. Further information on research design is available in the Nature Research Reporting Summary linked to this article.

## Data availability

All data supporting the findings of this study are provided within the paper and its Supplementary information. Additional information, *Drosophila*, and cell lines generated in this study are available upon request from N.C.K. Requests for mitofusin agonists should be made to G.W.D. Source data are provided with this paper.

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

## Acknowledgements

We thank the Bloomington Drosophila Stock Center, the VDRC Stock Center for fly lines, the IRCAN's Molecular and Cellular Core Imaging (PICMI) facility, and the Research Instrumentation Lab at the University of Minnesota Duluth for assistance. We also thank undergraduate volunteers, Sierra Skudlarek, Austin Kurtz, Anna Huy, and Maddie Chalmers, helping in *Drosophila* maintenance and plasmids preparation. This work was supported by grants from the Muscular Dystrophy Association, the Wallin neuroscience discovery fund, the Engebretson Drug Design and Discovery Grant, and the NINDS/NIA 1R56NS112296-01 to N.C.K.

## Author contributions

M.B., Y.-J.C, and N.C.K. conceived the project and performed experiments. S.B. performed all experiments using patient-derived fibroblasts. S.M. performed experiments measuring mitochondrial activity. M.B., Y.-J.C, S.B., S.M., J.H.K., V.P.-F., and N.C.K. analyzed data and wrote the manuscript. G.W.D. and J.P.T. provided guidance and helpful insights into design experiments and analyzed the data.

## Competing interests

G.W.D. is the scientific founder of Mitochondria in Motion, Inc., which has license from WUSTL and is commercializing small-molecule mitofusin agonists for the treatment of neurodegenerative diseases. The remaining authors declare no competing interests.
