## [Peer Review File · Nature Communications]

Reviewers' Comments:

Reviewer #1:

Remarks to the Author:

In this study, the authors use both *Drosophila* and HeLa cells to investigate the mechanisms by which mutant CHCHD10 proteins, CHCHD10(S59L) in particular, causes toxicity. These studies, if done well, are an important contribution to the field. However, several major concerns need to be addressed (please see below for more details). (1) The manuscript is quite difficult to read in part because it appears that the authors want to present everything they have done on CHCHD10 and different mutants in one manuscript, even though some results are not very relevant to the focus of the study, CHCHD10(S59L). (2) Some control experiments should be done as described below. (3) The whole study heavily relies on the overexpression approach. Even the effects of mutant CHCHD10 on TDP-43 were examined in cells overexpressing TDP-43. Considering that it has already been published before that overexpression of CHCHD10(S59L) causes mitochondrial defects, more sophisticated approaches should be used to increase the novelty of the study and relevance to real human disease.

1. This reviewer would like to suggest the authors just focus on the pathogenic mechanisms of CHCHD10(S59L) and take out most if not all the data on other mutants, which are quite confusing, often irrelevant and sometimes inconclusive, making the story incoherent and difficult to follow. This reviewer assumes the authors will agree, thus only focuses his/her comments on CHCHD10(S59L).

2. The authors should carefully compare the relative expression levels of C2C10H(WT) and C2C10H(S81L) in fly eyes on western blot and QUANTIFY. Similarly, Fig. S2 is so overexposed that it is difficult to tell without proper quantification (based on multiple independent western blots) whether these proteins are indeed expressed at the same level in HeLa cells as the authors claim.

3. The authors claim that enhanced phenotype in C2C10H(S81L) homozygous flies is due to transvection but without providing any actual data. Is it true that the levels of C2C10H(S81L) in homozygous flies much more than that in heterozygous flies?

4. It is unclear why the authors do not express CHCHD10(S59L) in fly motor neurons and muscles. Do CHCHD10(S59L) and C2C10H(S81L) behave in the same or different way when expressed in fly cells?

5. Overexpression studies in Fig. 1 and Fig. 2 are confirmatory in nature and consistent with what have been published in the literature. To go beyond the published overexpression studies, is it possible to obtain S59L patient fibroblast cells to confirm mitochondrial defects, and more importantly the proposed mechanisms?

6. The interpretations of genetic interaction studies on Pages 10-12 are confusing. The authors first conclude that "CHCHD10(S59L) acts as a dominant-negative mutant", then conclude "CHCHD10(S59L) is a dominant gain-of-function mutation". If the authors want to build a case for the latter, just present all the evidence to support their hypothesis.

7. In Figure 3A, the eye phenotypes of different genotypes must be quantified, and the conclusion must be based on statistical analysis rather than a single image. More importantly, a key control should be included in this experiment: UAS-GFP should be used as the negative control to show whether any suppression of the eye phenotype is due to at least in part dilution of Gal4. Such a critical control is also missing in Figure 3B.

8. The authors used the word "degeneration" in the HeLa experiments in Figure 3D/E. What does that mean?

9. It is not convincing to conclude that many mutants “maintained WT activity” just based on genetic interaction study. The eye phenotype is not a precise readout. Co-expression of different mutants is such an artificial setup and should be deleted to improve clarity of the manuscript. It is also an artificial setup to overexpress Parkin in CHCHD10 KO cells. The physiological relevance of this set of experiments is unclear.

10. The authors already generated CHCHD10 KO HeLa cells using CRISPR. If they could introduce the CHCHD10(S59L) heterozygous mutation into mammalian neurons or even HeLa cells, that would be very useful to confirm the phenotypes and mechanisms revealed by mutant protein overexpression.

11. In Figure 4B, co-expression of UAS-GFP should be used as a control for co-expression of UAS-CHCHD10-WT.

12. Overexpression of both TDP-43 and CHCHD10(S81L) is problematic. (BTW, what is CHCHD10(S81L)? I assume the authors mean to say CHC10H(S81L) here). Does expression of CHC10H(S81L) induce mislocalization of endogenous *Drosophila* TDP-43 (TBPH)? Does endogenous TBPH translocate into mitochondria? To go beyond what has been published in the literature, it would be helpful if the authors could examine whether TDP-43 is mislocalized in mitochondria in mammalian neurons cells with CHCHD10(S59L) knock-in, without artificial TDP-43 and CHCHD10(S59L) overexpression.

13. The identification of PINK1 as a strong suppressor of CHC10H(S81L) toxicity is interesting. However, the section entitled “Parkin-mediated mitophagy induces toxicity” is very confusing, which is in part because the authors describe many experiments using HeLa(YFP-Parkin) cells. If the authors think these cells are an artificial system, then they should delete these problematic experiments that do not provide much relevant insight into mutant CHCHD10 toxicity.

Reviewer #2:

Remarks to the Author:

Summary of main findings and impressions:

In 2014, CHCHD10 mutations were identified by Bannwarth and colleagues as causes of familial ALS, frontotemporal dementia, and myopathy. In this manuscript, Beak and colleagues use *Drosophila* animal models and mammalian HeLa cells to study the role of CHCHD10 in mitochondrial toxicity and associated phenotypes. In *Drosophila*, the CHCHD10 S59L mutation (S81L in C2C10H in *Drosophila*) was found to cause gain of function toxicity in eyes, motor neurons, and muscles. For simplicity sake, I will use human S59L and *Drosophila* S81L mutations interchangeably. Corresponding mitochondrial defects were associated with the S59L mutation in HeLa cells. Two divergent pathways were found to mediate CHCHD10 S59L mutation: TDP-43 & PINK1. First, CHCHD10 S59L mutation promoted TDP-43 insolubility and increased mitochondrial TDP-43, while blocking mitochondrial translocation of TDP-43 reduced CHCHD10 S59L-induced mitochondrial toxicity. At the same time, genetic reduction of PINK1 rescued CHCHD10 S59L-mediated phenotypes in *Drosophila* and HeLa cells. Two downstream substrates of PINK1 (mitofusion & mitofilin) were identified as genetic modifiers of CHCHD10 S59L. In addition, 2 peptide inhibitors of PINK1 mitigated mitochondrial defects induced by CHCHD10 S59L. Authors conclude that mitochondrial TDP-43 and chronic PINK1 activation underlie the dominant toxicity associated with CHCHD10 S59L.

Findings from this paper are interesting and potentially important. The *Drosophila* models are unique and first of kind, albeit concerns regarding the validity and pertinence of the mutations in *Drosophila* C2C10H to human disease. Findings from HeLa cells, for the most part, are not new, except for wild type CHCHD10 effects on suppressing S59L and some findings related to PINK1.

There are elements of the paper (few from drosophila & many from Hela cells) that appear to have been hastily put together without quantification, raising concerns about reproducibility. Some interpretations and conclusions are not justified by the findings. Specifically, experiments related to mitophagy/autophagy are either not quantified, unconvincing, or uninterpretable. On balance, this is a potentially interesting paper that makes some new contributions but requires some additional evaluation, rigorous assessment, and re-interpretation of findings. Below are this reviewer's general and specific critiques that need to be addressed.

General Critique:

1. This is an exhaustive study using drosophila models of CHCHD10 mutations, which is laudable and point to some interesting and potentially important findings. However, the authors used drosophila versions of CHCHD10/CHCHD10 (C2C10H) to introduce human mutations (i.e. S59L, R15L, G58R, G66V, etc), many of which are at distant/discordant positions within the drosophila C2C10H sequence. Although C2C10H has a fair degree of homology to mammalian CHCHD10 and CHCHD2, it is nevertheless questionable whether such C2C10H mutations confer identical or even similar modes of action as in mammalian CHCHD10. Hence, are the effects of these mutations in drosophila directly comparable to the modes of action of mammalian variants? In other words, does human CHCHD10-S59L also induce the same phenotype in drosophila?
2. In Hela cells, it is interesting that wild type CHCHD10 counteracts the effects of S59L mutation on various mitochondrial defects, including mitochondrial TDP-43 and its insolubility. Despite these effects, the authors insist that the S59L mutant is solely a dominant gain of function mutation. The data do not bear this conclusion, as many phenotypes appear to be dominant negative in action. The dominant gain of function associated with S59L only appears to be related to its ability to form insoluble aggregates, which seems to be additive to other effects of S59L.
3. In drosophila, the authors mention the presence of 3 CHCHD10-related proteins (C2C10H & 2 others). Other than phenotypes associated with S59L aggregation, can the authors truly conclude that S59L does not reduce the functions of the other 2 CHCHD10-related proteins? In other words, can they exclude the possibility that S59L does not act in a dominant negative manner? Same logic applies to Hela cells, as there are CHCHD10, CHCHD2, and 2 other CHCHD10/2-related proteins in humans.
4. In Hela cells, the CHCHD10 R15L mutation clearly fared worse than wild type CHCHD10. Despite R15L being able to partially rescue some of the effects of S59L, it was clearly worse than wild type CHCHD10. Hence, such effects should be clearly interpreted.
5. It seems clear that PINK1 RNAi rescues the rough eye phenotype associated with the C2C10H-S81L mutation. However, it would be highly instructive to know if PINK1 RNAi also rescues the effects of human CHCHD10-S59L mutant.
6. Although it is interesting that PINK1 knockdown rescues S59L toxicity, the mechanism by which such rescue occurs is not conceptually clear. Are there examples of PINK1 knockdown-mediated rescue of toxic phenotypes in other systems? It seems that mitofilin and mitofusin are related to the PINK1 knockdown-mediated rescue, but it is not clear how.

Specific critiques:

1. On page 9, the authors describe mitochondrial fragmentation induced by CHCHD10 mutations. Several other papers have seen similar effects and should be cited.
2. Figure 2c shows a lot of respiration data without statistical analyses. This should be done. Same with Fig. 3H, Fig. 4J, Fig. 5G, etc. If not statistically significant, this may be due to insufficient numbers of repeats.
3. On page 14, authors use the wording 'mitochondrial TDP-43 translocation'. This is not true. Translocation was never studied, only the amount associated with mitochondria. All subsequent references to translocation, except for those blocking translocation, should refer to translocation.
4. Fig. 5E needs magnification of mitochondrial images to see differences in mitochondrial length or fragmentation.
5. On page 19, please remove statements in reference to 'data not shown' that are not backed up

quantified data – i.e. 'drp1 knockdown prevents S59L aggregation'

6. In Fig. 6a & 6b, it does not make sense that there is no observable YFP-parkin or YFP-PINK1 in empty vector or wild type CHCHD10 transfected cells.

7. In Fig 6e, also does not make sense that there's no GFP-LC3 or mCherry-LC3 puncta or those colocalize with Tom20+ mitochondria.

8. In Fig. 6G, there's no quantification of LC3I and LC3II.

9. On page 18, authors refer to 'general nonspecific autophagy' in reference to autophagy cargo receptors (i.e. OPTN & NDP52). These are certainly not general nonspecific autophagy or basal autophagy receptors. These autophagy cargo receptors are part of the selective autophagy machinery and are recruited selectively to cargo and are activated under certain conditions. This part of the paper is extremely confusing and is not clear how it fits with the rest of the story.

10. Author claim that S59L does not affect mitochondrial membrane potential in Fig. S7. This claim is not believable and is not quantified. Many claims are made based on supplemental data that are not quantified, some of which are used as rationale for subsequent experiments in the main text. Such statements should not be made and/or such data should be removed.

11. On page 12, RNAi-mediated knockdown of mitofilin eye phenotype should be shown, since this is one of the effectors of the PINK1 pathway claimed to mediate S59L toxicity.

12. In Figure 7a, it is not clear what phenotype is being referred to by looking at the images. No quantification is made.

13. Likewise, quantifications in Fig. 8a, 8c, and 8d are needed.

Reviewer #3:

Remarks to the Author:

In the current manuscript, the authors employed *Drosophila* and mammalian cell culture systems to dissect the molecular events associated with the CHCHD10S59L dominant toxicity in the onset of ALS-FTD. CHCHD10 is known to be associated with mitochondrial contact site and cristae organizing system (MICOS) and involved in the regulation of cristae structure and mitochondrial functions. The loss of function mutations in the CHCHD10 are identified as significant drivers of the ALS and associated neuropathologies. In the current manuscript authors showing that CHCHD10S59L has a toxic gain of function, and this dominant toxicity is mediated through two distinct axes: TDP-43 and PINK1. The expression of CHCHD10S59L increased insolubility and mitochondrial translocation of TDP-43 and also activated PINK1-mediated pathways. The article is clearly written, and overall the study is well performed. However, there are some major fundamental concerns which should be clarified. Primarily the lack of molecular mechanism needs to be addressed.

Specific Comments (Major concerns):

1. In Figure 3, the authors show that when two C2C10HS81L copies induced a relatively mild rough eye phenotypes, which was rescued by WT expression as well as with the expression of R16L, P59S, G80R, and G88V mutants. However, in Figure 2, authors showing that the expression of these mutants has mitochondrial structural abnormalities, mitochondrial fragmentation, and defects in the mitochondrial respiration. What could be explanation of this phenotype? Are most of these mutations essentially the same as WT or is it that the phenotype is so much less severe than S81L that they essentially act as dominant negatives? Weird that you can rescue the mito phenotype of S81L with another mutant that has a similar mito phenotype, please elaborate.

2. The S59L mutants form insoluble protein aggregates, however, formation of S59L aggregates is not recovered by the WT expression which recovered its toxic effects on the mitochondria. This might be suggesting that S59L aggregates are not toxic or do not have major effect on the mitochondrial functions. Can authors provide any explanation for the relevance of S59L aggregates in terms of pathology? What is the size of these aggregates?

3. The authors show that S59L mutants have increased TDP-43 mitochondrial translocation and aggregate formation, which can be recovered by the expression of CHCHD10 WT. How does S59L mediate TDP-43 mitochondrial translocation? Any data to support this?

4. The blocking of TDP-43 mitochondrial localization recovered the S59L phenotype which suggests

that mitochondrial aggregates of the TDP-43 are major downstream targets of the S59L toxicity. Any data that there is a functional difference in PM1 rescued S59L phenotype? The OCR data looks interesting but not significant.

5. The S59L or S81L mutants display severe mitochondrial dysfunction and fragmentation. PINK/Parkin is a major stress-responsive pathway (mitophagy) to clear damaged mitochondria from the cellular system. In Fig 5 the authors show that the knockdown of PINK/Parkin is actually preventing fragmentation and mitochondrial dysfunction. How is blocking mitophagy (i.e. the clearance pathway) protective? These data are counter intuitive.

6. It's strange that the mitophagy is mediating the phenotype when the primary insult is protein aggregation and mito dysfunction. Wouldn't this still be occurring, likely even increased, with loss of mitophagic clearance. What happens to the dysfunctional mitos? If this is all mediated by mitophagy how is the S59L mutant of CHCHD10 directly linked to mitophagic signaling? While there is some evidence in the literature for interaction with mitophagic proteins, for publication in Nat Comms it would be nice to have some clear mechanistic data.

7. What is the molecular link between TDP-43 and PINK1?

We thank the three reviewers for their thoughtful comments on our manuscript. We have taken care to fully address each comment, as detailed below. We have added some key data to the manuscript that corroborates our findings. In brief, this new data include:

- We extended the *in vitro* studies to additional cell types for many experiments, including HEK293T, neuroblastoma-derived SH-SY5Y cells, and patient-derived fibroblasts.
- We show that PINK1 knockdown reduces mitochondrial fragmentation observed in patient-derived fibroblasts.
- We show that the mitochondrial association of endogenous TDP-43 is also increased by CHCHD10^{S59L} expression in neuroblastoma-derived SH-SY5Y cells and a humanized TDP-43 *Drosophila in vivo*.
- We added experiments using CHCHD2/CHCHD10 double knockout cells to test the independency of CHCHD10^{S59L} toxicity from wild-type CHCHD10 and CHCHD2.
- We found out that CHCHD10^{S59L} binds TDP-43 more strongly than wild-type CHCHD10.
- We document CHCHD10^{S59L}-induced mitophagy in multiple cell lines.
- We show that human CHCHD10^{S59L} expression in *Drosophila* also induces abnormal phenotypes in multiple tissues and Pink1 knockdown ameliorates rough eye phenotypes caused by CHCHD10^{S59L}.

Specific responses to each reviewer comment:

Reviewer #1 (Remarks to the Author):

In this study, the authors use both Drosophila and HeLa cells to investigate the mechanisms by which mutant CHCHD10 proteins, CHCHD10(S59L) in particular, causes toxicity. These studies, if done well, are an important contribution to the field. However, several major concerns need to be addressed (please see below for more details). (1) The manuscript is quite difficult to read in part because it appears that the authors want to present everything they have done on CHCHD10 and different mutants in one manuscript, even though some results are not very relevant to the focus of the study, CHCHD10(S59L). (2) Some control experiments should be done as described below. (3) The whole study heavily relies on the overexpression approach. Even the effects of mutant CHCHD10 on TDP-43 were examined in cells overexpressing TDP-43. Considering that it has already been published before that overexpression of CHCHD10(S59L) causes mitochondrial defects, more sophisticated approaches should be used to increase the novelty of the study and relevance to real human disease.

We appreciate this assessment. This reviewer made three suggestions for improvement, which we have completely addressed, as described below.

In response to the reviewer's suggestion, (1) we rewrote our manuscript with the results for CHCHD10^{S59L} and excluded the data for the other variants from the manuscript. All excluded data are still available as a preprint [1] and we are preparing a small manuscript to publish those data. (2) We have included some omitted control experiment data and also performed

additional control experiments. Those data have been incorporated into our revised manuscript. (3) To reduce the reviewer's concerns, we performed many additional experiments using different cell lines, knockout, double knockout cell lines, and patient-derived fibroblasts. Especially, we have documented the behavior of endogenous TDP-43 in both human cells and *Drosophila*. As the reviewer mentioned, mitochondrial defects caused by overexpressing CHCHD10^{S59L} have been previously reported several times. This manuscript describes completely new findings on how CHCHD10^{S59L} overexpression generates mitochondrial toxicity. Furthermore, we have demonstrated possible therapeutic strategies using chemical compounds and novel peptide inhibitors based on our novel findings.

1. This reviewer would like to suggest the authors just focus on the pathogenic mechanisms of CHCHD10(S59L) and take out most if not all the data on other mutants, which are quite confusing, often irrelevant and sometimes inconclusive, making the story incoherent and difficult to follow. This reviewer assumes the authors will agree, thus only focuses his/her comments on CHCHD10(S59L).

Following the reviewer's suggestion, we reorganized our manuscript with a focus on the CHCHD10^{S59L}, and the results for all other mutant variants were excluded from the revised manuscript.

2. The authors should carefully compare the relative expression levels of C2C10H(WT) and C2C10H(S81L) in fly eyes on western blot and QUANTIFY. Similarly, Fig. S2 is so overexposed that it is difficult to tell without proper quantification (based on multiple independent western blots) whether these proteins are indeed expressed at the same level in HeLa cells as the authors claim.

We performed three sets of Western blots from flies expressing C2C10H^{WT}-FLAG and C2C10H^{S81L}-FLAG and quantified expression levels (Supplementary Fig. 1 e and e'). Also, we quantified expression levels of CHCHD10 WT and S59L proteins from HeLa and Parkin-expressing HeLa cells with three independent replicates (Supplementary Fig. 2 a and a'). We observed slightly less S59L expression in multiple experiments. However, not much difference was observed in the expression level of wild-type and mutant proteins after normalization with a loading control in both *Drosophila* and human cells.

3. The authors claim that enhanced phenotype in C2C10H(S81L) homozygous flies is due to transvection but without providing any actual data. Is it true that the levels of C2C10H(S81L) in homozygous flies much more than that in heterozygous flies?

We conducted multiple independent western blot experiments to determine the C2C10H^{S81L} expression level of heterozygous UAS-C2C10H^{S81L} (attp2), heterozygous double copies having UAS-C2C10H^{S81L} (attp40) and UAS-C2C10H^{S81L} (attp2), homozygous UAS-C2C10H^{S81L} (attp2) with a GMR-GAL4. If there is no transvection, we expect two-fold increase of CHCHD10 in GMR-GAL4/+; UAS-C2C10H^{S81L}/UAS-C2C10H^{S81L} compared to GMR-GAL4/+; UAS-C2C10H^{S81L}.

However, as evident in Supplementary Figure 1g and g', homozygous UAS-C2C10H^{S81L} (attp2) with a GMR-GAL4 (GMR-GAL4/+; UAS-C2C10H^{S81L}/UAS-C2C10H^{S81L}) expressed more than four times higher CHCHD10 compared to heterozygous UAS-C2C10H^{S81L} (attp2) (GMR-GAL4/+; UAS-C2C10H^{S81L}/+), and about two-fold higher amount of CHCHD10 compared to the heterozygous double copies having UAS-C2C10H^{S81L} (attp40) and UAS-C2C10H^{S81L} (attp2) (GMR-GAL4/UAS-C2C10H^{S81L}; UAS-C2C10H^{S81L}/+). As expected, the heterozygous double copies expressed about twice of CHCHD10 compared to heterozygous UAS-C2C10H^{S81L} (attp2).

4. It is unclear why the authors did not express CHCHD10(S59L) in fly motor neurons and muscles. Do CHCHD10(S59L) and C2C10H(S81L) behave in the same or different way when expressed in fly cells?

Although some orthologs are evolutionarily conserved and may have the same physiological functions in their own system, expressing human proteins in the *Drosophila* system may not produce expected results for numerous reasons [2]. Therefore, our basic strategy is using a species-matched gene to develop a Disease model (i.e., a *Drosophila* gene in *Drosophila*, a human gene in human cells). Then we examine if they induce the same or similar phenotypes and affect the same pathways and cellular functions. In addition, we observed relatively weak rough eye phenotypes when we used human CHCHD10^{S59L} with GMR-GAL4 compared to when we used C2C10H^{S81L}. Thus, we did not use human CHCHD10 in *Drosophila* after we saw initial abnormal eye phenotypes. However, in response to the reviewer's suggestion, we generated flies expressing human CHCHD10 WT and S59L in motor neurons and muscle tissues with OK371 and MHC-GAL4, respectively. Similar to the results from flies expressing C2C10H^{S81L}, expression of human CHCHD10^{S59L} in motor neurons caused crawling defects (Supplementary Fig. 1 h and h'), and expression in muscles showed punctate staining pattern of human CHCHD10^{S59L} in *Drosophila* muscles, degenerated muscles, abnormal mitochondria and decreased ATP production (Supplementary Fig. 2c, d and d') compared to control animals. Consistently, we got relatively weak abnormal phenotypes compared to when we used C2C10H^{S81L}. The rough eye phenotypes with GMR-GAL4;CHCHD10^{S59L} is available in Supplementary Fig. 5b.

5. Overexpression studies in Fig. 1 and Fig. 2 are confirmatory in nature and consistent with what have been published in the literature. To go beyond the published overexpression studies, is it possible to obtain S59L patient fibroblast cells to confirm mitochondrial defects, and more importantly the proposed mechanisms?

As the reviewer suggested, to access patient-derived fibroblasts, we made a collaboration with Dr. Paquis-Flucklinger who originally identified CHCHD10^{S59L} mutation in patients. We confirmed that mitochondrial defects were observed in patient-derived fibroblast cells, and the fragmented mitochondrial network was rescued by RNAi-mediated knockdown of PINK1 in two different patient-derived fibroblasts. We added this result as a Figure 5g and g' (and supplementary figure 5k). Unfortunately, due to COVID19 pandemic, other long-term

experiments using iPSC-derived neurons could not be performed. Currently, we cannot anticipate when we can re-initiate the collaborative experiments. They are enlisted as coauthors, and we deeply appreciate their effort to get this data during the lockdown of their lab in France.

6. The interpretations of genetic interaction studies on Pages 10-12 are confusing. The authors first conclude that “CHCHD10(S59L) acts as a dominant-negative mutant”, then conclude “CHCHD10(S59L) is a dominant gain-of-function mutation”. If the authors want to build a case for the latter, just present all the evidence to support their hypothesis.

We reorganized and rewrote this section with new data using CHCHD10 knockout cells (HeLa^{C10KO}) and CHCHD2/CHCHD10 double knockout cells (HeLa^{C2C10DKO}). Briefly, because we use an overexpression system to generate the toxicity, there can be three possible modes of action, dominant-negative (antimorph), gain-of-function (hypermorph), and gain-of-function (gain-of-toxicity or neomorph). We first tested whether CHCHD10^{S59L} is a gain-of-function mutant enhancing its normal functions (hypermorph) by co-expressing CHCHD10^{S59L} (or C2C10H^{S81L}) with CHCHD10^{WT} (or C2C10H^{WT}). Because CHCHD10^{WT} (or C2C10H^{WT}) did not enhance the abnormal phenotypes caused by CHCHD10^{S59L} (or C2C10H^{S81L}) (Figure 3a-e), we rejected the possibility of hypermorph. Then, we examined the dependency of CHCHD10^{S59L} on the presence of CHCHD10^{WT} to test whether CHCHD10^{S59L} is a dominant-negative mutant suppressing the activity of CHCHD10^{WT}. Previously, we used CHCHD10 knockout HeLa^{YFP-Parkin} but we generated new knockout cell lines to reduce the reviewer’s concerns. With C2C10H null Drosophila, HeLa^{C10KO}, and HeLa^{C2C10DKO}, we showed that the toxicity of CHCHD10^{S59L} (or C2C10H^{S81L}) was not dependent on the presence of CHCHD10^{WT} (Figure 3f-j). Thus, we rejected the possibility of dominant-negative. Taken all these together, especially the ability of CHCHD10^{S59L} inducing toxicity without CHCHD2 and CHCHD10 (C2C10H) led us to conclude that the toxicity of CHCHD10^{S59L} (C2C10H^{S81L}) is generated by newly acquired abnormal functions of CHCHD10^{S59L} (C2C10H^{S81L}).

7. In Figure 3A, the eye phenotypes of different genotypes must be quantified, and the conclusion must be based on statistical analysis rather than a single image. More importantly, a key control should be included in this experiment: UAS-GFP should be used as the negative control to show whether any suppression of the eye phenotype is due to at least in part dilution of Gal4. Such a critical control is also missing in Figure 3B.

Before we decided which control lines are appropriate for our experiments, we tested several different control lines with our 2X_S81L model flies including w¹¹¹⁸ that was used as a control in many experiments in the previous manuscript. As seen in supplementary figure 3a and a', we did not observe any significant difference in eye phenotypes except two control lines inducing transvection. Two control lines inserted in attp2 (GFPattp2 and B35785, mCherry RNAi on attp2)

enhanced abnormal eye phenotypes due to transvection (Supplementary Fig. 1g-g' and 3b and b'). Although it did not affect scientific interpretation, however, we repeated our experiments for several important experiments using UAS-GFPattp2 or UAS-Luciferaseattp2 to reduce the reviewer's concerns. We added the repeated new data in Figure 3a-c. As shown in Figure 3a and Supplementary Figure 3a, it was observed that UAS-GFPattp2 enhanced degenerative phenotype compared to w¹¹¹⁸. All eye phenotypes were quantified by measuring the disorderliness of ommatidial arrangement with flynotyper [3] after processed with ilastik [4]. Please note that with the control experiment data presented in supplementary figure 3a, we still use w¹¹¹⁸ as a control for Figure 5a to make the figure as simple as possible.

8. *The authors used the word “degeneration” in the HeLa experiments in Figure 3D/E. What does that mean?*

We corrected the term used by mistake. Thank you for pointing out.

9. *It is not convincing to conclude that many mutants “maintained WT activity” just based on genetic interaction study. The eye phenotype is not a precise readout. Co-expression of different mutants is such an artificial setup and should be deleted to improve clarity of the manuscript. It is also an artificial setup to overexpress Parkin in CHCHD10 KO cells. The physiological relevance of this set of experiments is unclear.*

In response to the reviewer's suggestion, we only focus on the CHCHD10^{S59L} mutant to improve the clarity of our manuscript.

10. *The authors already generated CHCHD10 KO HeLa cells using CRISPR. If they could introduce the CHCHD10(S59L) heterozygous mutation into mammalian neurons or even HeLa cells, that would be very useful to confirm the phenotypes and mechanisms revealed by mutant protein overexpression.*

We have plenty of experience to generate knockout cells and *Drosophila* using the CRISPR/Cas9 system. However, the efficiency of introducing a heterozygous mutation is totally different from that of knocking-out in both human cells and *Drosophila*. Thus, we have not pursued this experiment while revising this manuscript. However, we will generate knock-in models (iPSC or neuroblastoma-derived SH-SY5Y) to address various questions raised from this manuscript in the near future (we expect 3-4 years to generate and characterize iPSC-derived neurons bearing heterozygous mutant variants).

11. *In Figure 4B, co-expression of UAS-GFP should be used as a control for co-expression of UAS-CHCHD10-WT.*

Figure 4B is a western blot result from HeLa cells, not from *Drosophila*. We are sorry for the confusion.

12. *Overexpression of both TDP-43 and CHCHD10(S81L) is problematic. (BTW, what is*

CHCHD10(S81L)? I assume the authors mean to say CHC10H(S81L) here). Does expression of CHC10H(S81L) induce mislocalization of endogenous Drosophila TDP-43 (TBPH)? Does endogenous TBPH translocate into mitochondria? To go beyond what has been published in the literature, it would be helpful if the authors could examine whether TDP-43 is mislocalized in mitochondria in mammalian neurons cells with CHCHD10(S59L) knock-in, without artificial TDP-43 and CHCHD10(S59L) overexpression.

We corrected *CHCHD10^{S81L}* to *C2C10H^{S81L}*. Thank you for pointing out. As this reviewer suggested, we have investigated whether the *C2C10H^{S81L}* expression induces mislocalization from the nucleus or colocalization with mitochondria using endogenous *Drosophila TDP-43 (TBPH)*. First, we tested a few polyclonal anti-TBPH antibodies to detect endogenous TBPH. However, the quality of images produced by anti-TBPH antibodies were not enough to clearly detect TBPH in muscle tissues. While we were waiting to get one more anti-TBPH antibodies from Dr. Morton (delayed due to the COVID19 pandemic), we tested a humanized TDP-43 *Drosophila* generated by replacing the entire coding region of TBPH with human TDP-43 cDNA. In this animal, TDP-43 is expressed under the control of the endogenous TBPH transcription unit [5]. Briefly, we generated flies expressing C2C10H WT and S81L in muscle tissues with the replaced human TDP-43 and then performed immunofluorescence experiments with anti-TDP43 antibodies. Most of TDP-43 was observed in the nucleus in the C2C10H WT-expressing muscles, whereas in the C2C10H S81L-expressing muscles, TDP-43 was observed not only in the nucleus but also in other cytosolic locations including mitochondria. We added this result in Supplementary Fig 4e and e'.

We also examined the distribution of endogenous TDP-43 in neuroblastoma-derived SH-SY5Y cells after EV, *CHCHD10^{WT}*, and *CHCHD10^{S59L}* transfection. Similar to the results in *Drosophila*, most of TDP-43 was observed in the nucleus in the EV and WT transfected cells, whereas TDP-43 was observed not only in the nucleus but also in the cytoplasm including mitochondria. To analyze mitochondria-associated TDP-43 levels, mitochondria were isolated from EV, WT, and S59L-transfected SH-SY5Y cells and subjected to Western blotting. As seen in Supplementary Fig. 4d and d', mitochondria-associated TDP-43 was increased more than two-fold in *CHCHD10^{S59L}*-transfected cells.

As explained in #10, the efficiency of introducing a heterozygous mutation is totally different from that of knocking-out in both human cells and *Drosophila*. Thus, we have not pursued this experiment while revising this manuscript.

13. The identification of PINK1 as a strong suppressor of C2C10H(S81L) toxicity is interesting. However, the section entitled "Parkin-mediated mitophagy induces toxicity" is very confusing, which is in part because the authors describe many experiments using HeLa(YFP-Parkin) cells. If the authors think these cells are an artificial system, then they should delete these problematic experiments that do not provide much relevant insight into mutant CHCHD10 toxicity.

As we mentioned in our manuscript, *parkin* is not expressed in HeLa cells (original HeLa cells

distributed by ATCC). To study the relationship between PINK1 and parkin, and their effects on mitophagy in HeLa cells, Richard Youle's group generated HeLa cells stably expressing YFP-Parkin about 12 years ago [6]. Since then, HeLa^{YFP-Parkin} has been used as a standard model to study mitophagy and a lot of scientific knowledge about mitophagy has been generated with this cell line. The role of the PINK1/Parkin pathway identified in HeLa^{YFP-Parkin} was confirmed in other systems [7][8][9]. Thus, we used this established and proven HeLa^{YFP-Parkin} to test whether CHCHD10^{S59L} affects the PINK1/Parkin-mediated mitophagy. In Parkin-deficient HeLa cells (original HeLa cells), mitophagy cannot be dramatically induced like HeLa^{YFP-Parkin} due to the lack of Parkin. However, it has been reported that the PINK1-dependent, Parkin-independent mitophagy still occur in original HeLa cells with mitophagy receptors, optineurin and NDP52[10].

In our *Drosophila* experiments, the degenerated eye phenotypes of C2C10H^{S81L} were reduced by RNAi-mediated knockdown of PINK1 and Parkin (Fig. 5 a and a'). This suggests that a cellular pathway commonly controlled by PINK1 and Parkin, mitophagy, is the main toxicity-generating pathway. To test this possibility in a human system, we used HeLa^{YFP-Parkin} that is an established and proven model system in the mitophagy field. As expected and described in our manuscript, CHCHD10^{S59L} induced PINK1 stabilization and Parkin accumulation resulting in increased mitophagy supported by various experimental methods (Fig. 6). These results clearly support that abnormally increased mitophagy by CHCHD10^{S59L} is toxic. However, there is still an unanswered question. What happens in original HeLa Cells? Many of ours and other labs' data showing mitochondrial toxicity have been generated with original HeLa deficient Parkin expression and the toxicity without Parkin is significantly strong [11]–[13]. Thus, we tested whether the PINK1-dependent, Parkin-independent mitophagy in HeLa without Parkin is a toxicity generating pathway in original HeLa cells. However, experimental data suggest that other cellular pathways rather than mitophagy should be involved in generating PINK1-dependent, parkin-independent toxicity in HeLa deficient Parkin expression. In the current experimental setting with HeLa and HeLa^{YFP-Parkin}, we do not know which mechanism, Parkin-dependent or -independent, is more dominant in human patients. However, the knockdown of PINK1 showed more strong suppressive effects than that of Parkin in *Drosophila* eyes, which might suggest that Parkin-independent toxicity also takes significant parts to generated toxicity *in vivo*. Definitely, it is clear that modulating PINK1 would affect both toxicity-generating pathways and have the strongest suppressive ability (Fig. 5 and 7).

To verify our results from HeLa and HeLa^{YFP-Parkin}, we used a neuroblastoma-derived cell line, SH-SY5Y expressing both endogenous PINK1 and Parkin. New data were added in the revised manuscript (Supplementary Fig. 6 c, c', e, and e').

Reviewer #2 (Remarks to the Author):

Summary of main findings and impressions:

In 2014, CHCHD10 mutations were identified by Bannwarth and colleagues as causes of familial ALS, frontotemporal dementia, and myopathy. In this manuscript, Beak and colleagues use drosophila animal models and mammalian HeLa cells to study the role of CHCHD10 in mitochondrial toxicity and associated phenotypes. In drosophila, the CHCHD10 S59L mutation (S81L in C2C10H in drosophila) was found to cause gain of function toxicity in eyes, motor neurons, and muscles. For simplicity sake, I will use human S59L and drosophila S81L mutations interchangeably. Corresponding mitochondrial defects were associated with the S59L mutation in HeLa cells. Two divergent pathways were found to mediate CHCHD10 S59L mutation: TDP-43 & PINK1. First, CHCHD10 S59L mutation promoted TDP-43 insolubility and increased mitochondrial TDP-43, while blocking mitochondrial translocation of TDP-43 reduced CHCHD10 S59L-induced mitochondrial toxicity. At the same time, genetic reduction of PINK1 rescued CHCHD10

S59L-mediated phenotypes in drosophila and HeLa cells. Two downstream substrates of PINK1 (mitofusion & mitofilin) were identified as genetic modifiers of CHCHD10 S59L. In addition, 2 peptide inhibitors of PINK1 mitigated mitochondrial defects induced by CHCHD10 S59L. Authors conclude that mitochondrial TDP-43 and chronic PINK1 activation underlie the dominant toxicity associated with CHCHD10 S59L.

Findings from this paper are interesting and potentially important. The drosophila models are unique and first of kind, albeit concerns regarding the validity and pertinence of the mutations in drosophila C2C10H to human disease. Findings from HeLa cells, for the most part, are not new, except for wild type CHCHD10 effects on suppressing S59L and some findings related to PINK1. There are elements of the paper (few from drosophila & many from HeLa cells) that appear to have been hastily put together without quantification, raising concerns about reproducibility. Some interpretations and conclusions are not justified by the findings. Specifically, experiments related to mitophagy/autophagy are either not quantified, unconvincing, or uninterpretable. On balance, this is a potentially interesting paper that makes some new contributions but requires some additional evaluation, rigorous assessment, and re-interpretation of findings. Below are this reviewer's general and specific critiques that need to be addressed.

General Critique:

1. This is an exhaustive study using drosophila models of CHCHD10 mutations, which is laudable and point to some interesting and potentially important findings. However, the authors used drosophila versions of CHCHD10/CHCHD10 (C2C10H) to introduce human mutations (i.e. S59L, R15L, G58R, G66V, etc), many of which are at distant/discordant positions within the drosophila C2C10H sequence. Although C2C10H has a fair degree of homology to mammalian CHCHD10 and CHCHD2, it is nevertheless questionable whether such C2C10H mutations confer identical or even similar modes of action as in mammalian CHCHD10. Hence, are the effects of these mutations in Drosophila directly comparable to the modes of action of mammalian variants? In other words, does human CHCHD10-S59L also induce the same phenotype in drosophila?

As explained in the response to Reviewer1's 4th question, although some orthologs are evolutionarily well conserved and may have the same physiological functions in their own

system, expressing human proteins in the *Drosophila* system may not produce expected results for numerous reasons[2]. Therefore, our basic strategy is using a species-matched gene to develop a Disease model (i.e., a *Drosophila* gene in *Drosophila*, a human gene in human cells). We then examine if they induce the same or similar phenotypes, and affect the same pathways and cellular functions. When we expressed human CHCHD10^{S59L} in eyes, muscles, and neurons, we consistently observed relatively weak abnormal phenotypes compared to when we used *Drosophila* C2C10H^{S81L}. We showed that expression of codon-optimized human CHCHD10^{S59L} induced degenerated eye phenotypes in aged flies (Fig. 1C). We also observed that hCHCHD10 WT co-expression with C2C10H^{S81L} in fly eyes ameliorated degenerated eye phenotypes like C2C10H^{WT} (Supplementary Fig. 3d). To further address the reviewer's concerns, we performed additional experiments and observed that expression of human CHCHD10^{S59L} in motor neurons or muscles by OK371 or MHC-Gal4 respectively resulted in behavioral defects and decreased ATP production. We presented these results in the revised manuscript (Supplementary fig. 1h and 2d').

2. In HeLa cells, it is interesting that wild type CHCHD10 counteracts the effects of S59L mutation on various mitochondrial defects, including mitochondrial TDP-43 and its insolubility. Despite these effects, the authors insist that the S59L mutant is solely a dominant gain of function mutation. The data do not bear this conclusion, as many phenotypes appear to be dominant negative in action. The dominant gain of function associated with S59L only appears to be related to its ability to form insoluble aggregates, which seems to be additive to other effects of S59L.

As described in the response to Reviewer1's 6th question, we tested three possible modes of action: gain-of-function (hypermorph), dominant-negative (antimorph), and gain-of-toxicity (neomorph). When examining the possibility of dominant-negative using C2C10H null *Drosophila* and CHCHD10 Knockout HeLa, we observed that CHCHD10^{S59L} and C2C10H^{S81L} generated cellular defects regardless of the presence of wild-type CHCHD10 and C2C10H, which strongly suggests that CHCHD10^{S59L} (C2C10H^{S81L}) does not act in a dominant-negative manner suppressing the function of wild-type CHCHD10 (C2C10H). We included a close paralog CHCHD2 (related to the next reviewer's question) due to possible compensation. However, CHCHD10^{S59L} induced mitochondrial toxicity without both CHCHD10 and CHCHD2. These led us to conclude that CHCHD10^{S59L} does not act in a dominant-negative manner against CHCHD10 and CHCHD2. Please note that knockout of CHCHD10 and double knockout of CHCHD2/10 did not show significant defects in our experimental conditions that are observed with CHCHD10^{S59L} overexpression. This also indicates that those abnormal phenotypes caused by CHCHD10^{S59L} overexpression are not likely generated by the lack of WT activity. Therefore, we conclude that what we are observing in our systems (both *Drosophila* and human) is caused by a gain-of-toxicity, not a dominant-negative. Thus, all downstream pathways that we identified and tested in our systems such as PINK1, Parkin, Mitofusin, and mitofilin are also likely involved in the gain-of-toxicity of CHCHD10^{S59L}.

3. In drosophila, the authors mention the presence of 3 CHCHD10-related proteins (C2C10H & 2 others). Other than phenotypes associated with S59L aggregation, can the authors truly conclude that S59L does not reduce the functions of the other 2 CHCHD10-related proteins? In other words, can they exclude the possibility that S59L does not act in a dominant negative manner? Same logic applies to Hela cells, as there are CHCHD10, CHCHD2, and 2 other CHCHD10/2-related proteins in humans.

As described in our manuscript, two other paralogs of C2C10H expressed strongly in the testis and weakly in early-stage imaginal discs while C2C10H is strongly expressed in all *Drosophila* tissues including eyes. Thus, their activity is unlikely affected by C2C10H^{S81L} and generates toxicity in the eyes. Indeed, when we tested the genetic interaction between C2C10H^{S81L} and two other paralogs using RNAi-mediated knockdown in *Drosophila* eyes (Supplementary Table 1), we did not observe significant phenotype modification.

For human cells, we generated CHCHD10 and CHCHD2 double knockout cell lines and tested the independency of CHCHD10^{S59L} as discussed above (and Reviewer1 #6). As far as we understand, there are no additional CHCHD10/2-related proteins in the human genome other than pseudogenes.

4. In Hela cells, the CHCHD10 R15L mutation clearly fared worse than wild type CHCHD10. Despite R15L being able to partially rescue some of the effects of S59L, it was clearly worse than wild type CHCHD10. Hence, such effects should be clearly interpreted.

We reorganized our manuscript and data to focus on CHCHD10^{S59L} as other reviewers suggested. Thus, the data of R15L was also removed from our new manuscript and we will not discuss any further. However, we believe that this is another data supporting that mutant variants might have multiple different acting mechanisms. Some of them are common in all variants but some of them are unique in each variant. Thus, they may show partial rescue or partial toxicity dependent on the combination or the context. We will discuss this further if we can still have an opportunity to do it after this COVID19 pandemic.

5. It seems clear that PINK1 RNAi rescues the rough eye phenotype associated with the C2C10H-S81L mutation. However, it would be highly instructive to know if PINK1 RNAi also rescues the effects of human CHCHD10-S59L mutant.

In response to the reviewer's suggestion, we tested the effect of PINK1 RNAi on human CHCHD10^{S59L} in fly eyes. As we explained previously, we consistently observed relatively weaker abnormal phenotypes with human CHCHD10^{S59L}. However, the rough eye phenotypes induced by human CHCHD10^{S59L} expression were also reduced by PINK1 RNAi. Results are presented in Supplementary Fig.5b and b'.

6. Although it is interesting that PINK1 knockdown rescues S59L toxicity, the mechanism by which such rescue occurs is not conceptually clear. Are there examples of PINK1 knockdown-

mediated rescue of toxic phenotypes in other systems? It seems that mitofilin and mitofusin are related to the PINK1 knockdown-mediated rescue, but it is not clear how.

Several reports are showing that the reduction of the PINK1/Parkin pathway mitigates abnormal phenotypes in *in vivo* disease models including *Drosophila* and mice. We already briefly described these pieces of literature in the discussion section and cited the literature with a focus on the ALS models. For example, down-regulation of either Pink1 or parkin ameliorates Fus-induced neurodegeneration in *Drosophila* [14]. PINK1 is accumulated in the TDP-43^{Q331K}-induced mouse model and down-regulation of PINK1 extended lifespan in the *Drosophila* model of TDP-43 proteinopathy [15]. Genetic ablation of Parkin delayed disease progression and prolonged survival in the SOD^{G93A} mouse ALS model [16]. There are also much literature showing abnormal or excessive mitophagy-generated toxicity. For example, an ALS-FTD-linked gene, VCP also induces abnormal excessive mitophagy to generate toxicity in Huntington's disease models including mice and patient-derived cells. [17]. This toxicity can be mitigated by an inhibitor for VCP and huntingtin. Drp1-induced PINK1/Parkin-mediated excessive mitophagy was reported to cause neurodegeneration in a mouse model of multiple sclerosis and inhibiting this process with a small molecule reduced MS progression [18]. It can be extended to the outside of neurological diseases. Hyperactivation of PINK1 and excessive mitophagy is a whole mark of Fuchs Endothelial Corneal Dystrophy [19]. The PINK1-dependent mitophagy is a critical mechanism to induce COPD by cigarette smoke. Pink1(-/-) mice were protected against mitochondrial dysfunction induced by cigarette smoke [20]. Indeed, autophagy or mitophagy-mediated cell death has been well established and all these examples simply suggest that mitophagy is not always protective, finding the right balance in mitophagy is more important than just activating mitophagy.

Mitofusin and mitofilin are downstream substrates of PINK1 that can be phosphorylated by PINK1. Mitofusin is a common substrate of PINK1 kinase and Parkin ubiquitination. When mitofusin is phosphorylated by activated PINK1, mitofusin is ubiquitinated by Parkin recruited by activated PINK1, and ubiquitinated mitofusin is degraded by proteasomes. Thus, this process removing mitochondria fusion factor, mitofusin, induce mitochondria fragmentation. Fragmented mitochondria are subjected to mitophagy. CHCHD10^{S59L} induce PINK1 stabilization and accumulation in mitochondria (Figure 6) and the accumulated PINK1 phosphorylates mitofusin for its removal. Thus, overexpression of mitofusin or overexpression of non-phosphorylatable mitofusin should block this process and mitigate PINK1-mediated toxicity. When we overexpressed mitofusin or non-phosphorylatable mitofusin mutant in *Drosophila* or HeLa respectively. We observed partial rescue of degenerated eyes (Figure 5 a and a'). and mitochondrial morphology (Figure 7a and Supplementary Fig. 7 c) as expected. These results showed that the PINK1/Parkin pathway including its downstream mitofusin mediates the toxicity of CHCHD10^{S59L}. In Parkin-deficient original HeLa cells, phosphorylated mitofusin is not ubiquitinated and degraded but phosphorylation itself inactivates mitofusin [21]. As shown in Figure 7a and Supplementary Fig. 7 c, expression non-phosphorylatable mitofusin rescued

CHCHD10^{S59L}-induced mitochondrial fragmentation whereas wild-type mitofusin overexpression was uninterpretable (Supplementary Fig. 5 g). To further validate this result and test the possibility of therapeutic treatment, we treated recently developed mitofusin2 agonists to enhance mitofusin activity. As expected, mitofusin2 agonists ameliorated CHCHD10^{S59L} (C2C10H^{S81L})-induced toxicity in HeLa (Figure 7 b and c) and *Drosophila* (Figure 7 d and e), even in the *Drosophila* model for C9ORF72 (Supplementary Fig. 7f).

Mitofilin is also a downstream substrate of PINK1 kinase [22]. As explained above, if mitofilin is a downstream toxicity mediator phosphorylated by PINK1, non-phosphorylatable mutant mitofilin overexpression should mitigate CHCHD10^{S59L}-induced toxicity. When we express non-phosphorylatable mutant mitofilin in *Drosophila* eyes expressing C2C10H^{S81L}, we observed partially rescued eyes as expected (Supplementary Fig. 7a). Like mitophagy, the PINK1-mitofilin pathway is also reported as a beneficial signaling pathway generating more cristae structures. Our data indicate that excessive activation of the PINK1-mitofilin pathway can be also toxic in some situations.

Specific critiques:

1. On page 9, the authors describe mitochondrial fragmentation induced by CHCHD10 mutations. Several other papers have seen similar effects and should be cited.

We cited other papers as suggested in the revised manuscript.

2. Figure 2c shows a lot of respiration data without statistical analyses. This should be done. Same with Fig. 3H, Fig. 4J, Fig. 5G, etc. If not statistically significant, this may be due to insufficient numbers of repeats.

We invested substantial effort to repeat experiments and perform statistical analyses due to unexpected events such as malfunction of equipment and COVID19. Thus, all of our Seahorse experiments were performed by three different researchers at three different periods. To compare completely independent multiple experiments statistically, we employed a normalization method called Z-score that is calculated using raw data values from each plate (Please see the method section for the detailed information). We showed representative graphs in each figure and all actual statistical analyses results with calculated Z-score are in Supplementary Fig. 9.

3. On page 14, authors use the wording 'mitochondrial TDP-43 translocation'. This is not true. Translocation was never studied, only the amount associated with mitochondria. All subsequent references to translocation, except for those blocking translocation, should refer to association.

As suggested, we revised our manuscript with appropriate terms such as co-localized or associated, etc.

4. Fig. 5E needs magnification of mitochondrial images to see differences in mitochondrial length or fragmentation.

We have inserted magnified images in Figure 5e and Figure 7a.

5. On page 19, please remove statements in reference to 'data not shown' that are not backed up quantified data – i.e. 'drp1 knockdown prevents S59L aggregation'

We omitted the figure number by mistake. Supplementary Figure 5i showed immunofluorescence images after treating drp1 siRNA and quantification of the signals. CHCHD10^{S59L} aggregates were nearly absent with *DRP1* knockdown (Supplementary Fig. 5 i and i'), in contrast with persistent aggregates with *PINK1* knockdown (Figure 5 d and e).

6. In Fig. 6a & 6b, it does not make sense that there is no observable YFP-parkin or PINK1-YFP in empty vector or wild type CHCHD10 transfected cells.

Once PINK1 is synthesized, it is translocated into mitochondria, processed, transported back to the cytosol, and degraded by proteasomes. Thus, its protein level is always very low. Even overexpressed PINK1 by transfection is also degraded and not well detected in normal conditions [7]. However, in very rare cases, we observed cells showing Pink1-YFP signals in empty vector and CHCHD10 WT transfected cells. However, YFP signals were not co-localized with mitochondria, as shown below.

In contrast, YFP-Parkin spreads in the cytosol and nucleus. Figure 6b clearly showed basal YFP-Parkin signals in empty vector and CHCHD10 WT transfected cells.

7. In Fig 6e, also does not make sense that there's no GFP-LC3 or mCherry-LC3 puncta or those colocalize with Tom20+ mitochondria.

When we took confocal microscope images, we set parameters for taking LC-3-positive structures without overexposure in CHCHD10^{S59L} transfected cells. Due to the parameter setting for bright structures, we got dimmed images from EV and CHCHD10 WT transfected cells. However, as you see our quantification data, definitely we observed LC-3 signals in EV and CHCHD10 WT transfected cells. The weak or no signal of GFP-LC3 has been commonly reported in many publications [23][24][25][26]. Simply, we used representative pictures among the many pictures we took. Also, as explained in reviewer 1's 13th question, there is PINK1-dependent mitophagy without Parkin in original HeLa cells but it is hard to detect because only 1% of cells display mitophagy signals in normal conditions. Upon artificial PINK1 recruitment to mitochondria, it increases to 7% [10]. Thus, it is normal to see no colocalization of LC3 with Tom20 in the original HeLa with normal conditions. However, in Parkin-expressing HeLa cells, mitophagy is robust so the colocalization can be seen as you see in our quantification data (Figure 6 f and h). However, as we mentioned, those are minor and cannot be a representative picture. However, to prevent misunderstanding and overemphasizing, we took confocal microscope images again with a modified setting and replaced it with new images in revised manuscript Fig. 6e with quantification and colocalization (Figure 6 f and h).

8. In Fig. 6G, there's no quantification of LC3I and LC3II.

We have quantified the results and presented them in the revised manuscript. Besides, to clarify Fig. 6g results, we examined mitochondrial LC3-II after fractionating mitochondria from HeLa, HeLa^{YFP-Parkin}, and SH-SY5Y (Supplementary Fig 6b-c'). We also confirmed the mito-lysosome formation by mito-QC from each cell lines (Fig 6i, j, and Supplementary Fig 6d-e').

9. On page 18, authors refer to 'general nonspecific autophagy' in reference to autophagy cargo receptors (i.e. OPTN & NDP52). These are certainly not general nonspecific autophagy or basal autophagy receptors. These autophagy cargo receptors are part of the selective autophagy machinery and are recruited selectively to cargo and are activated under certain conditions. This part of the paper is extremely confusing and is not clear how it fits with the rest of the story.

We are sorry for the confusion. We rewrote the section to clearly explain experimental designs, rationales, and interpretations as explained above for reviewer1's 13th question. As the reviewer mentioned, OPTN & NDP52 have been linked to selective autophagy (xenophagy). However, OPTN & NDP52 also have been identified as the primary receptors for the PINK1/Parkin-mediated mitophagy and PINK1-dependent, Parkin-independent mitophagy in HeLa cells without parkin [10][27].

In our Drosophila experiments, the degenerated eye phenotypes of C2C10H^{S81L} were reduced by RNAi-mediated knockdown of PINK1 and Parkin (Fig. 5 a and a'). This suggests that a cellular pathway commonly controlled by PINK1 and Parkin, mitophagy, is a main toxicity-generating pathway. To test this possibility in a human system, we used HeLa^{YFP-Parkin} that is an established and proven model system in the mitophagy field. As expected and described in our manuscript,

CHCHD10^{S59L} induced PINK1 stabilization and Parkin accumulation resulting in increased mitophagy supported by various experimental methods (Fig. 6). These results clearly support that abnormally increased mitophagy by CHCHD10^{S59L} is toxic. However, there is still an unanswered question. What happens in original HeLa Cells that do not express Parkin? Many of ours and other labs' data showing mitochondrial toxicity have been generated with original HeLa deficient Parkin expression and the toxicity without Parkin is significantly strong [11]–[13]. Thus, we tested whether the PINK1-dependent, Parkin-independent mitophagy in HeLa without parkin is a toxicity generating pathway in original HeLa cells. However, experimental data with RNAi-mediated knockdown of OPTN & NDP52 suggest that other cellular pathways rather than mitophagy should be involved in generating PINK1-dependent, parkin-independent toxicity in HeLa deficient Parkin expression. Therefore, we searched which downstream targets of PINK1 can mediate toxicity without Parkin (Figure 7 and Supplementary Fig. 7, mitofusin and mitofilin)

In the current experimental setting with HeLa and HeLa^{YFP-Parkin}, we do not know which mechanism, Parkin-dependent or -independent, is more dominant in human patients. However, the knockdown of PINK1 showed more strong suppressive effects than that of Parkin in *Drosophila* eyes, which might suggest that Parkin-independent toxicity also takes significant parts to generated toxicity *in vivo*. Definitely, it is clear that modulating PINK1 would affect both toxicity-generating pathways and have the strongest suppressive ability (Fig. 5 and 7).

10. Author claim that S59L does not affect mitochondrial membrane potential in Fig. S7. This claim is not believable and is not quantified. Yes, quantify. Many claims are made based on supplemental data that are not quantified, some of which are used as rationale for subsequent experiments in the main text. Such statements should not be made and/or such data should be removed.

To address the reviewer's concern, we repeated TMRM-based membrane potential measurements multiple times with different combinations of experimental conditions. We also have used both live-imaging and a microplate reader. The results are the same. CHCHD10^{S59L} expression with the split GFP technique did not induce membrane potential change compared to CHCHD10^{WT} expression with the split GFP technique. Experiments using a microplate reader with FLAG-tagged CHCHD10 also showed no significant difference between CHCHD10^{S59L} and CHCHD10^{WT} transfected cells. Surprisingly, however, we found out that a TMRM release rate after measuring the initial membrane potential is significantly different between CHCHD10^{S59L} and CHCHD10^{WT} cells, as shown below. We are testing several possibilities to understand this intriguing phenomenon caused by CHCHD10^{S59L}. Thus, we decided to remove this part in the revised manuscript though this section was originally designed to understand how CHCHD10^{S59L} initiates PINK1 stabilization and accumulation.

(a) Representative images of CHCHD10^{WT}- or CHCHD10^{S59L}-GFP11 with MTS-GFP1-10 transfected HeLa cells. Images were taken after staining with 100nM TMRM. Note that TMRM signals of CHCHD10^{WT}- and CHCHD10^{S59L}-expressing cells were similar at the initial measurement (after washing out TMRM), but after an hour TMRM signals were reduced dramatically in CHCHD10^{S59L} expressing cell. **(b)** HeLa cells were plated on the 96 well assay plate and transfected with EV, FLAG-tagged CHCHD10^{WT}, and CHCHD10^{S59L}. After 24 hours, cells were stained with TMRM (100 nM), and the TMRM intensity was measured by a plate reader (BioTek). Ex; 530/25, Em; 575/15. Data shown are mean \pm SD (one-way ANOVA and *posthoc* Dunnett's test, * $p < 0.05$, $n = 3$ independent experiments).

11. On page 12, RNAi-mediated knockdown of mitofilin eye phenotype should be shown, since this is one of the effectors of the PINK1 pathway claimed to mediate S59L toxicity.

In response to the reviewer's suggestion, we added eye phenotypes showing the effect of mitofilin RNAi on C2C10H^{S81L}-induced rough eyes (Supplementary Fig. 7 b and b'). RNAi-mediated mitofilin knockdown enhanced the rough eye phenotypes of C2C10H^{S81L}. Two different RNAi lines (v47615 and v47616) were used. As seen in Supplementary Fig. 7 a and a', non-phosphorylatable mutant mitofilin (mitofilin PR) by PINK1 partially rescued C2C10H^{S81L}-induced phenotypes, suggesting phosphorylated mitofilin by PINK1 is a mediator of the PINK1-generated toxicity.

12. In Figure 7a, it is not clear what phenotype is being referred to by looking at the images. No quantification is made.

We have inserted magnified mitochondrial images marked by the dotted line in the figure and quantified result was added as a supplementary Fig 7c.

13. Likewise, quantifications in Fig. 8a, 8c, and 8d are needed.

The quantified results were added to the revised manuscript.

Reviewer #3 (Remarks to the Author):

In the current manuscript, the authors employed *Drosophila* and mammalian cell culture

systems to dissect the molecular events associated with the CHCHD10S59L dominant toxicity in the onset of ALS–FTD CHCHD10 is known to be associated with mitochondrial contact site and cristae organizing system (MICOS) and involved in the regulation of cristae structure and mitochondrial functions. The loss of function mutations in the CHCHD10 are identified as significant drivers of the ALS and associated neuropathologies. In the current manuscript authors showing that CHCHD10S59L has a toxic gain of function, and this dominant toxicity is mediated through two distinct axes: TDP-43 and PINK1. The expression of CHCHD10S59L increased insolubility and mitochondrial translocation of TDP-43 and also activated PINK1-mediated pathways. The article is clearly written, and overall the study is well performed. However, there are some major fundamental concerns which should be clarified. Primarily the lack of molecular mechanism needs to be addressed.

Specific Comments (Major concerns):

1. In Figure 3, the authors show that when two C2C10H^{S81L} copies induced a relatively mild rough eye phenotypes, which was rescued by WT expression as well as with the expression of R16L, P59S, G80R, and G88V mutants. However, in Figure 2, authors showing that the expression of these mutants has mitochondrial structural abnormalities, mitochondrial fragmentation, and defects in the mitochondrial respiration. What could be explanation of this phenotype? Are most of these mutations essentially the same as WT or is it that the phenotype is so much less severe than S81L that they essentially act as dominant negatives? Weird that you can rescue the mito phenotype of S81L with another mutant that has a similar mito phenotype, please elaborate.

As other reviewers have suggested, we rewrote our manuscript focusing on the results for CHCHD10^{S59L} and excluded the data for the other variants from the manuscript. We are preparing a short manuscript to publish data for other variants. Thus, we hope that we can discuss this soon in another paper.

2. The S59L mutants form insoluble protein aggregates, however, formation of S59L aggregates is not recovered by the WT expression which recovered its toxic effects on the mitochondria. This might be suggesting that S59L aggregates are not toxic or do not have major effect on the mitochondrial functions. Can authors provide any explanation for the relevance of S59L aggregates in terms of pathology? What is the size of these aggregates?

In our systems, CHCHD10^{WT} or C2C10H^{WT} showed promoting effects on mitochondrial length, respiration, and ATP production when CHCHD10^{WT} or C2C10H^{WT} were expressed in normal HeLa cells or Drosophila respectively. CHCHD10^{WT} or C2C10H^{WT} also rescued S59L- or S81L-dependent phenotypes, especially without changing the aggregated pattern or the insolubility of CHCHD10^{S59L} in HeLa cells. CHCHD10^{S59L} does not need to suppress the function of CHCHD10^{WT} to produce cellular toxicity. However, CHCHD10^{WT} has the ability to promote mitochondrial health and reduce CHCHD10^{S59L}-induced toxicity. Our interpretation of all these data is that CHCHD10^{WT} improves mitochondrial integrity by modulating the downstream events of CHCHD10^{S59L} aggregation, not directly restoring mutant protein aggregation. Thus, we

examined whether CHCHD10^{WT} can reduce TDP-43 insolubility and PINK1 accumulation. As seen in Figure 4 b and b', CHCHD10^{WT} reduced TDP-43 insolubility not only in CHCHD10^{S59L}-expressing cells but also in normal HeLa cells. We also examined whether CHCHD10^{WT} reduces PINK1 accumulation caused by CCCP, not caused by CHCHD10^{S59L} expression. As shown in Figure f and f', CHCHD10^{WT} reduced PINK1 accumulation caused by CCCP treatment (membrane potential disruption). (Yes, if we can show membrane potential disruption by the expression of CHCHD10^{S59L}, it may make this story perfect.).

However, this does not give a clear answer for whether CHCHD10^{S59L} aggregate itself is toxic or other intermediates such as soluble oligomers are toxic or if it is just a byproduct. This is actually a fundamental question that has been repeatedly asked in many neurodegenerative diseases, whether aggregates are toxic or not such as tau in Alzheimer's, alpha-synuclein in Parkinson's disease, and TDP-43 in ALS-FTD. Unfortunately, we think addressing this question is beyond a scope of our current manuscript. We are actively preparing a series of experiments to probe the toxicity of CHCHD10^{S59L} aggregate itself using various methods such as optogenetic modulation of CHCHD10^{WT} or CHCHD10^{S59L} aggregation demonstrated in opto-granules or opto-TDP43 papers [28][29]. Direct treatment of pre-formed aggregates used in other literature may not be appropriated for CHCHD10 because CHCHD10 is inside of mitochondria. We hope that we could share this result soon.

We have not measured the size of CHCHD10 aggregates directly. We only see the aggregate under the confocal microscopy. In this case, the length of the fluorescence signal of CHCHD10 aggregates and mitochondria was measured and presented in Figure 2b, Figure 3e, and Figure 5e. In all cases, the length of CHCHD10^{S59L} aggregates was around 1.25-1.5 μm .

3. The authors show that S59L mutants have increased TDP-43 mitochondrial translocation and aggregate formation, which can be recovered by the expression of CHCHD10 WT. How does S59L mediate TDP-43 mitochondrial translocation? Any data to support this?

The physical interaction of TDP-43 and CHCHD10 has been reported in HT22 cells using co-immunoprecipitation with transfected Tomato-HA-tagged TDP-43 and CHCHD10-FLAG [12]. We also repeated and confirmed their result using the same plasmids. Thus, we tested whether S59L mutation in CHCHD10 changes their interaction between TDP-43 and CHCHD10^{S59L}. We found out that CHCHD10^{S59L}-FLAG showed the increased binding capacity to TDP-43-Tomato-HA than CHCHD10^{WT}-FLAG, and this was reconfirmed with a different tagging combination; CHCHD10-HA and TDP-43-FLAG. We presented these results in the revised manuscript in Fig. 4i and Supplementary Fig. 4g-h'.

4. The blocking of TDP-43 mitochondrial localization recovered the S59L phenotype which suggests that mitochondrial aggregates of the TDP-43 are major downstream targets of the S59L toxicity. Any data that there is a functional difference in PM1 rescued S59L phenotype? The OCR data looks interesting but not significant.

PM1 treatment increased ATP production by 51% and mitochondrial length by 48% in S59L-expressing cells compared to control peptides treatment. These are statistically significant. We added the statistical analysis to the revised manuscript (Figure 4h and Supplementary Figure 4f, f' and 9).

5. The S59L or S81L mutants display severe mitochondrial dysfunction and fragmentation. PINK/Parkin is a major stress-responsive pathway (mitophagy) to clear damaged mitochondria from the cellular system. In Fig 5, the authors show that the knockdown of PINK/Parkin is actually preventing fragmentation and mitochondrial dysfunction. How is blocking mitophagy (i.e. the clearance pathway) protective? These data are counter intuitive.

We agree that this looks counter-intuitive. We believe that mitophagy is not always protective, and excessive mitophagy is toxic to cells. CHCHD10^{S59} induces hyperactivation of the PINK1/Parkin pathway in Drosophila and human cells. As far as we know, CHCHD10^{S59L} overexpression is the strongest activator for the PINK1/Parkin pathway in HeLa cells except an uncoupling agent, CCCP. Therefore, the reduction of excessive mitophagy should be beneficial to our models. We presented all supporting data in this manuscript from Drosophila to Patient-derived fibroblasts. Although we did not have evidence showing hyperactive mitophagy in human patients, when we knocked down PINK1 in patient-derived fibroblasts, it also ameliorated mitochondrial fragmentation (Figure 5 g and g'). Indeed, increased mitophagy has been reported in CHCHD10^{S59L} knock-in mice [30]. Complete blocking of mitophagy in the long-term will be harmful to any cell. However, maintaining a balanced mitophagy by reducing excessive mitophagy should be beneficial to cells. Especially, this is important in the PINK1/Parkin-mediated mitophagy because it is a specialized and inducible mitophagy.

Please also see our answer for reviewer 2, general critic #6, and an answer for the next question.

6. It's strange that the mitophagy is mediating the phenotype when the primary insult is protein aggregation and mito dysfunction. Wouldn't this still be occurring, likely even increased, with loss of mitophagic clearance. What happens to the dysfunctional mitos? If this is all mediated by mitophagy how is the S59L mutant of CHCHD10 directly linked to mitophagic signaling? While there is some evidence in the literature for interaction with mitophagic proteins, for publication in Nat Comms it would be nice to have some clear mechanistic data.

Many reports are showing that excessive or maladaptive mitophagy causes cytotoxicity in various disease models [17], [20], [31]. For example, similar to our results, the mutant huntingtin (mhtt) accumulates in mitochondria after binding with VCP in a Huntington's disease (HD) model. This elicits excessive mitophagy, causing neuronal cell death. Blocking mhtt-VCP interaction corrects excessive mitophagy and reduces cell death in HD model [17]. In other cases, reducing excessive or maladaptive mitophagy to a normal level results in ameliorating disease-related phenotypes. Especially in ALS, there are several previous reports about the protective effects of the reduced PINK1/Parkin pathways [14]–[16] and we already discussed it in the discussion section in our manuscript.

We believe that cytotoxicity of *CHCHD10*^{S59L} in our systems is due to excessively increased or malregulated PINK1/Parkin-dependent mitophagy and PINK1-dependent, parkin-independent downstream pathways that we identified.

Yes, as you said, *CHCHD10*^{S59L} may work through their interacting partners. For activation of the PINK1/Parkin pathway, *CHCHD10*^{S59L} might use its interaction with PINK1 processing proteases (interaction data are available in public databases). Although this is a direction we are currently pursuing, it seems to be outside the current scope of this manuscript. In this manuscript, we are presenting from the development of *in vivo* models to the demonstration of possible therapeutic strategies using small chemical molecules and novel peptide inhibitors targeting pathway identified in this study. With the genetically tractable *Drosophila* model, we dissected biochemical and genetic pathways mediating the toxicity of *CHCHD10*^{S59L} and identified PINK1 and Parkin as strong genetic modifiers that can modulate S81L-induced phenotypes. Since the PINK1/Parkin pathway is regarded as a protective system, we are providing plenty of biochemical and cell biological evidence to support our claim that S59L-induced toxicity is generated by the PINK1/Parkin pathway using *Drosophila*, HeLa, SH-SY5Y, and patients-derived fibroblasts. We concluded that the previously well documented PINK1/Parkin-dependent mitophagy, and PINK1-dependent, Parkin-independent pathways are toxicity generating mechanisms of *CHCHD10*^{S59L}. Based on these results, we focused on one downstream toxicity mediator, mitofusin, and PINK1 itself to demonstrate possible therapeutic strategies with a recently developed mitofusin agonist and novel peptide inhibitors for PINK1 that we generated. As a basic scientist studying devastating diseases lacking any treatment, our primary strategy is the identification of genetic pathways that can mitigate disease phenotypes and testing whether it can be a therapeutic strategy using existing chemicals or novel molecules. Therefore, in this manuscript, we are providing genetically dissected pathogenic mechanisms of *CHCHD10*^{S59L}. We believe that the timely sharing of genetically dissected and confirmed impactful pathogenic mechanisms is the best way to expand the field to understand this destructive disease and help patients.

7. What is the molecular link between TDP-43 and PINK1?

We examined TDP-43 insolubility and mitochondrial enrichment after PINK1 down-regulation in *CHCHD10*^{S59L}-expressing HeLa cells. We could not find any differences in the TDP-43 insolubility and mitochondrial enrichment when PINK1 was downregulated by RNAi (Figure 8a and a'). On the contrary, we also investigated whether Pink1 accumulation on mitochondria can be affected by blocking TDP-43 mitochondrial translocation in *CHCHD10*^{S59L}-expressing HeLa cells. We did not observe any differences in the PINK1 accumulation on *CHCHD10*^{S59L}-expressing cells with PM1 treatment. We also tested whether RNAi-mediated knockdown of PINK1 rescues TDP43^{M337V}-dependent eye degeneration in *Drosophila*. Although we acquired only inconclusive data in our system, Sun et al. reported that RNAi-mediated knockdown of PINK1 increased the life span of the TDP-43^{Q331K} *Drosophila* model[15]. Therefore, we concluded that PINK1 and

TDP-43 are two independent toxic pathways in our model systems but it is still possible that TDP-43 induced mitochondrial toxicity can induce the PINK1/Parkin-mediated mitophagy pathway in some situations (Figure 9).

References

- [1] M. Baek, Y.-J. Choe, G. Dorn, J. P. Taylor, and N. C. Kim, “Dominant toxicity of ALS–FTD-associated CHCHD10 S59L is mediated by TDP-43 and PINK1,” 2019.
- [2] H. J. Bellen, M. F. Wangler, and S. Yamamoto, “The fruit fly at the interface of diagnosis and pathogenic mechanisms of rare and common human diseases,” *Hum. Mol. Genet.*, vol. 28, no. R2, pp. R207–R214, 2019.
- [3] J. Iyer *et al.*, “Quantitative assessment of eye phenotypes for functional genetic studies using *Drosophila melanogaster*,” *G3 Genes, Genomes, Genet.*, vol. 6, no. 5, pp. 1427–1437, 2016.
- [4] S. Berg *et al.*, “Ilastik: Interactive Machine Learning for (Bio)Image Analysis,” *Nat. Methods*, vol. 16, no. 12, pp. 1226–1232, 2019.
- [5] J. C. Chang and D. B. Morton, “*Drosophila* lines with mutant and wild type human TDP-43 replacing the endogenous gene reveals phosphorylation and ubiquitination in mutant lines in the absence of viability or lifespan defects,” *PLoS One*, vol. 12, no. 7, pp. 1–24, 2017.
- [6] D. Narendra, A. Tanaka, D. F. Suen, and R. J. Youle, “Parkin is recruited selectively to impaired mitochondria and promotes their autophagy,” *J. Cell Biol.*, vol. 183, no. 5, pp. 795–803, 2008.
- [7] D. P. Narendra *et al.*, “PINK1 is selectively stabilized on impaired mitochondria to activate Parkin,” *PLoS Biol.*, vol. 8, no. 1, 2010.
- [8] C. Vives-Bauza *et al.*, “PINK1-dependent recruitment of Parkin to mitochondria in mitophagy,” *Proc. Natl. Acad. Sci. U. S. A.*, vol. 107, no. 1, pp. 378–383, 2010.
- [9] R. G. Carroll, E. Hollville, and S. J. Martin, “Parkin Sensitizes toward Apoptosis Induced by Mitochondrial Depolarization through Promoting Degradation of Mcl-1,” *Cell Rep.*, vol. 9, no. 4, pp. 1538–1553, 2014.
- [10] M. Lazarou *et al.*, “The ubiquitin kinase PINK1 recruits autophagy receptors to induce mitophagy,” *Nature*, vol. 524, no. 7565, pp. 309–314, 2015.
- [11] S. Bannwarth *et al.*, “A mitochondrial origin for frontotemporal dementia and amyotrophic lateral sclerosis through CHCHD10 involvement,” *Brain*, vol. 137, no. 8, pp. 2329–2345, 2014.
- [12] J. A. A. Woo *et al.*, “Loss of function CHCHD10 mutations in cytoplasmic TDP-43 accumulation and synaptic integrity,” *Nat. Commun.*, vol. 8, pp. 1–15, 2017.
- [13] X. Huang *et al.*, “CHCHD2 accumulates in distressed mitochondria and facilitates

- oligomerization of CHCHD10,” *Hum. Mol. Genet.*, vol. 27, no. 22, pp. 3881–3900, 2018.
- [14] Y. Chen *et al.*, “PINK1 and Parkin are genetic modifiers for FUS-induced neurodegeneration,” *Hum. Mol. Genet.*, vol. 25, no. 23, pp. 5059–5068, 2016.
- [15] X. Sun *et al.*, “Distinct multilevel misregulations of Parkin and PINK1 revealed in cell and animal models of TDP-43 proteinopathy,” *Cell Death Dis.*, vol. 9, no. 10, 2018.
- [16] G. M. Palomo *et al.*, “Parkin is a disease modifier in the mutant SOD 1 mouse model of ALS,” *EMBO Mol. Med.*, vol. 10, no. 10, 2018.
- [17] X. Guo *et al.*, “VCP recruitment to mitochondria causes mitophagy impairment and neurodegeneration in models of Huntington’s disease,” *Nat. Commun.*, vol. 7, 2016.
- [18] W. Li *et al.*, “Nitration of Drp1 provokes mitophagy activation mediating neuronal injury in experimental autoimmune encephalomyelitis,” *Free Radic. Biol. Med.*, vol. 143, no. August, pp. 70–83, 2019.
- [19] T. Miyai *et al.*, “Activation of PINK1-Parkin-Mediated Mitophagy Degrades Mitochondrial Quality Control Proteins in Fuchs Endothelial Corneal Dystrophy,” *Am. J. Pathol.*, vol. 189, no. 10, pp. 2061–2076, 2019.
- [20] K. Mizumura *et al.*, “Mitophagy-dependent necroptosis contributes to the pathogenesis of COPD,” *J. Clin. Invest.*, vol. 124, no. 9, pp. 3987–4003, 2014.
- [21] A. G. Rocha *et al.*, “MFN2 agonists reverse mitochondrial defects in preclinical models of Charcot-Marie-Tooth disease type 2A,” *Science (80-.)*, vol. 360, no. 6386, pp. 336–341, 2018.
- [22] P. I. Tsai *et al.*, “PINK1 Phosphorylates MIC60/Mitofilin to Control Structural Plasticity of Mitochondrial Crista Junctions,” *Mol. Cell*, vol. 69, no. 5, pp. 744-756.e6, 2018.
- [23] M. Renna *et al.*, “Autophagic substrate clearance requires activity of the syntaxin-5 SNARE complex,” *J. Cell Sci.*, vol. 124, no. 3, pp. 469–482, 2011.
- [24] A. Totaro *et al.*, “Cell phenotypic plasticity requires autophagic flux driven by YAP/TAZ mechanotransduction,” *Proc. Natl. Acad. Sci. U. S. A.*, vol. 116, no. 36, pp. 17848–17857, 2019.
- [25] S. Jung *et al.*, “Buffering of cytosolic calcium plays a neuroprotective role by preserving the autophagy-lysosome pathway during MPP⁺-induced neuronal death,” *Cell Death Discov.*, vol. 5, no. 1, 2019.
- [26] A. C. Becker *et al.*, “Influenza A virus induces autophagosomal targeting of ribosomal proteins,” *Mol. Cell. Proteomics*, vol. 17, no. 10, pp. 1909–1921, 2018.
- [27] J. M. Heo, A. Ordureau, J. A. Paulo, J. Rinehart, and J. W. Harper, “The PINK1-PARKIN Mitochondrial Ubiquitylation Pathway Drives a Program of OPTN/NDP52 Recruitment and TBK1 Activation to Promote Mitophagy,” *Mol. Cell*, vol. 60, no. 1, pp. 7–20, 2015.
- [28] J. R. Mann *et al.*, “RNA Binding Antagonizes Neurotoxic Phase Transitions of TDP-43,” *Neuron*, vol. 102, no. 2, pp. 321-338.e8, 2019.

- [29] K. Asakawa, H. Handa, and K. Kawakami, “Optogenetic modulation of TDP-43 oligomerization accelerates ALS-related pathologies in the spinal motor neurons,” *Nat. Commun.*, vol. 11, no. 1, pp. 1–16, 2020.
- [30] E. C. Genin *et al.*, “Mitochondrial defect in muscle precedes neuromuscular junction degeneration and motor neuron death in CHCHD10 S59L/+ mouse,” *Acta Neuropathol.*, vol. 138, no. 1, pp. 123–145, 2019.
- [31] S. H. Su, Y. F. Wu, D. P. Wang, and J. Hai, “Inhibition of excessive autophagy and mitophagy mediates neuroprotective effects of URB597 against chronic cerebral hypoperfusion,” *Cell Death Dis.*, vol. 9, no. 7, 2018.

Reviewers' Comments:

Reviewer #1:

Remarks to the Author:

The authors have addressed most if not all my concerns and made some reasonable arguments for the experiments they prefer to do in the future.

I would like to suggest the authors go through the whole manuscript carefully and correct all typos.

Reviewer #2:

Remarks to the Author:

This revision is quite extensive and has addressed many of the concerns of this reviewer. Some new experiments were added, which corroborate the original conclusions of the paper.

Quantifications are adequately done with a few exceptions. hCHCHD10-S59L expression in drosophila is added, which produces the rough eye phenotype. However, drosophila genetic interaction experiments, which identified PINK1, are still done with S81L. Critiques below.

1. Fig. S1b,c show rough eye phenotype P score that is higher for both CHCHD10-WT and CHCHD10-S59L compared to both of their drosophila counterparts, although a mutation-dependent effect is detected. However, the observation that human CHCHD10-WT produces a P score greater than the drosophila S81L mutant is concerning, suggesting that expression of human CHCHD10 per se in drosophila exerts degenerative effects. The crawling phenotype with expression in motor neurons (Fig. S1h) somewhat mitigates this concern.

2. Although PINK1 RNAi robustly rescues the S81L eye phenotype, PINK1 RNAi does not seem to effectively rescue human S59L-mediated eye phenotype in vivo (Fig. S5b , P score: 55 vs 54). This continues to raise the specter that drosophila S81L and human S59L mutations are not functionally equivalent in vivo.

3. Fig. 4b,d,f graphs show normalization at 100% with exactly equal values for EV controls. This should be corrected to reflect normal variability of each data point. Same for similar graphs in Supplemental Figs.

4. Fig. 6 shows basal levels of PINK1, Parkin recruitment without inducer such as CCCP/FCCP. Are inducible levels of PINK1, Parkin recruitment to mitochondria also hyperactive in S59L expressing cells?

5. Could mitochondrial fragmentation and dysfunction simply and nonspecifically increase the mitochondrial recruitment PINK1/Parkin in S59L expressing cells? The PINK1-dependent and PINK1-independent roles of S59L on mitophagy/toxicity are still difficult to interpret.

6. The authors show that WT CHCHD10 protects against TDP-43 insolubility and mitochondrial dysfunction. Similar findings were presented in a recent paper (FASEB 34:8493), which should be cited. The same paper also showed reduced CHCHD10 in FTLD-TDP & TDP-43 mice and that protective effects of CHCHD10 occur in part through Opa1/mitofilin. Given the investigation of the fusion protein Mfn1/2 and mitofilin as modifiers of S59L in the current study, the similarities, differences, and potential implications should be discussed.

Reviewer #3:

Remarks to the Author:

I think the manuscript is substantially improved in the revised version. Thanks for examining closely all of our previous comments.

Specific responses to each reviewer comment:

Reviewer #1 (Remarks to the Author):

The authors have addressed most if not all my concerns and made some reasonable arguments for the experiments they prefer to do in the future.

I would like to suggest the authors go through the whole manuscript carefully and correct all typos.

We appreciate the reviewer's suggestion. We will ensure no typos or mistakes in the manuscript's final version with professional editing.

Reviewer #2 (Remarks to the Author):

This revision is quite extensive and has addressed many of the concerns of this reviewer. Some new experiments were added, which corroborate the original conclusions of the paper. Quantifications are adequately done with a few exceptions. *hCHCHD10-S59L* expression in *drosophila* is added, which produces the rough eye phenotype. However, *drosophila* genetic interaction experiments, which identified *PINK1*, are still done with *S81L*. Critiques below.

1. Fig. S1b,c show rough eye phenotype P score that is higher for both *CHCHD10-WT* and *CHCHD10-S59L* compared to both of their *drosophila* counterparts, although a mutation-dependent effect is detected. However, the observation that human *CHCHD10-WT* produces a P score greater than the *drosophila* *S81L* mutant is concerning, suggesting that expression of human *CHCHD10* per se in *drosophila* exerts degenerative effects. The crawling phenotype with expression in motor neurons (Fig. S1h) somewhat mitigates this concern.

Figure R1. Processed images with ilastik to detect ommatidia. (A) normal control eye image taken with Leica in 2017, (B) normal control eye image taken with SMZ1500 in 2020. Although both images are normal control eyes, the results are dependent on the quality of images and (A) produced P-score 23.8 and (B) produces P-score 41 by flynotyper. Ommatidia in yellow marked area are not well detected in images taken by SMZ1500, resulting in increased P-score.

As the reviewer already pointed out, human *CHCHD10-WT* did not induce apparent defects when expressed in motor neurons. It was the same in the eyes. As shown in the micrographs in Figure1C-WT, the expression of human *CHCHD10-WT* did not cause any obvious abnormal phenotypes. Yes, the P-score of human *CHCHD10-WT* (Figure1 c, S1c) is higher than *Drosophila* *C2C10HS81L* (Figure1 b, S1b). However, this P-score difference is due to the image analysis method and resolution difference with two different equipment. The two independent experiments' P-scores should not be compared directly. To quantify abnormal eye phenotypes, we measured the ommatidial disorderliness with flynotyper [1] after processed with ilastik [2]. Although this method can eliminate human bias when accessing eye degeneration, eye image quality is critical for image processing with ilastik, a

machine-learning-based image segmentation software. While the images in Figure 1b were taken with Leica M205C equipped with a ring light and a DFC320 digital camera in January 2017, the images in Figure 1c were taken in September 2020 with Nikon SMZ1500 equipped with a ring light and a DMX1200 digital camera. Although the difference in image quality between Figure 1b and 1c may not be noticeable in small micrographs, it is evident in a higher zoom mode in the pdf manuscript. As shown in Figure R1, ilastik detects more ommatidia in a higher quality image taken with Leica and produces a low score compared to an image taken with SMZ1500. Unfortunately, we cannot use Leica currently. However, the images from SMZ1500 are enough to serve our purpose to detect abnormal ommatidia arrangement, although it produces higher P-scores and provides a narrow analysis window. We are preparing a detailed protocol manuscript for this analysis that combines illastik and flynotyper.

To show whether human CHCHD10-WT induces defects more clearly, we collected fresh one day old GMR-GAL4 flies having normal eyes. Then we took images using SMZ1500 with the same setting for the newly collected GMR-GAL4 and the 45-days old human CHCHD10-WT flies stored in a deep freezer (-80°C) for about one year. The P-score of the GMR-GAL4 and human CHCHD10-WT are 46.37 ± 3.8 (n=7) and 46.70 ± 4.7 (n=8), respectively. There is no statistical difference between the two groups. Therefore, we revised our manuscript as follows to avoid unnecessary misunderstanding.

1. Method section:

Eye images for group A (Fig. 1b, supplementary Fig. 1d and f, Fig. 3a, supplementary Fig. 3d and Fig. 5a) were taken by a Leica M205C stereomicroscope equipped with a ring light and a Leica DFC320 digital camera. Eye images for group B (Fig. 1c, Fig. 3f, supplementary Fig. 3a and e, supplementary fig. 5b, supplementary Fig. 7a and b) were taken by Nikon SMZ1500 equipped with a ring light and a Nikon DXM1200 digital camera. Although we have applied the same criteria to analyze all eye images with illastik and flynotyper, the P-scores only can be compared within each experiment due to the difference caused by equipment, setting, and researchers.

2. Results description:

However, while wild-type C2C10H and CHCHD10 expression did not cause any apparent defects, expression of C2C10H^{S81L} and CHCHD10^{S59L} caused mild but mutation-dependent degeneration with depigmentation as the flies aged, regardless of FLAG tagging (Fig. 1b, c, and Supplementary Fig. 1d).

3. Supplementary figure 1 legends:

*(b and c) Degenerative eye quantification of Fig 1b and Fig 1c, respectively. Boxes indicate median, 25th and 75th percentiles. Bars indicate the highest and lowest values (t-test, *p < 0.05, n = 4 for each group). Please note that P-scores should not be directly compared between 1b and 1c (please see the method section). The P-scores of normal GMR-GAL4 for 1b and 1c is 23.12 ± 1.5 (n=5) and 46.37 ± 3.8 (n=7), respectively*

2. Although PINK1 RNAi robustly rescues the S81L eye phenotype, PINK1 RNAi does not seem to effectively rescue human S59L-mediated eye phenotype in vivo (Fig. S5b, P score: 55 vs 54). This continues to raise the specter that drosophila S81L and human S59L mutations are not functionally equivalent in vivo.

The low P-score difference can be partially due to image quality taken by SMZ1500. However, in our experiments, human CHCHD10^{S59L} in the fly system always shows weaker phenotypes than C2C10H^{S81L}. The genetic interaction between *Drosophila* PINK1 and human CHCHD10^{S59L} is also weaker, as shown in the micrographs (Fig. S5b) and P-scores (Fig. S5b'). However, in the micrograph (Fig S5b), human CHCHD10^{S59L} induces depigmentation and degeneration in some areas, and the reduction of those defects is apparent with PINK1 RNAi. It is evident that human CHCHD10^{S59L} consistently acts in the same manner as C2C10H^{S81L} in the *Drosophila* system but to a different extent.

As we previously stated, we usually use human genes in the human system and fly genes in the fly system due to the species difference. Although many human genes can rescue their fly counterparts' knockout phenotypes in the fly system, this does not mean that the human gene rescuing null phenotypes can be 100% exchangeable with the fly counterpart. The human gene may have various levels of functional activity in the fly system. Importantly, CHCHD10 does not have any catalytic activity. The only way how CHCHD10 works may be through protein-protein interaction. Co-evolution of genes may preserve interactions between fly proteins or between human proteins in their own systems. However, through the co-evolution with their interacting partners, the human protein may lose some interactions to the binding partners of fly counterparts or may have weaker interactions compared to their fly counterparts. Considering the species difference, when a human protein shows expected results in the fly system, it definitely supports their same role in the fly system and human system. However, if a human protein does not show expected results in the fly system, this result cannot determine whether the human and fly proteins are functionally conserved. It is actually inconclusive and requires further studies. It should be studied in the human system and the fly system separately whether their functions are conserved for the same cellular pathways in their own systems. This manuscript provides plenty of evidence supporting their roles in the same pathways in the *Drosophila* systems and the human systems using fly genes in the fly system and human genes in the human system. We also provide evidence showing human CHCHD10^{S59L} works and interacts with PINK1 in the same manner as its fly counterpart (though it is weaker).

3. Fig. 4b,d,f graphs show normalization at 100% with exactly equal values for EV controls. This should be corrected to reflect normal variability of each data point. Same for similar graphs in Supplemental Figs.

We re-normalized all data and performed statistical analyses again. All changes have been updated.

4. Fig. 6 shows basal levels of PINK1, Parkin recruitment without inducer such as CCCP/FCCP. Are inducible levels of PINK1, Parkin recruitment to mitochondria also hyperactive in S59L expressing cells?

In Figure 6, we showed that CHCHD10^{S59L}-dependent accumulation of PINK1 and Parkin. A small number of cells showed PINK1 accumulation without S59L. It may be caused by transfection or overexpression stress.

If the reviewer asks whether the level of PINK1 and Parkin accumulation in S59L-expressing cells is comparable to the hyperactive CCCP-induced PINK1 and Parkin accumulation, it is not easy to compare directly. 10 μ M – 20 μ M CCCP treatment rapidly

induces PINK1 and Parkin accumulation within 30 minutes, and more than 80% of cells show PINK1/Parkin accumulation in 6 hours (Supplementary Figure 8). After 24 hours, we observed cell death or cells that do not have mitochondria due to extensive mitophagy. CHCHD10^{S59L} transient over-expression for 24 hours induces PINK1 and Parkin accumulation in 85-90% of transfected cells, but those cells survive for a few more days (we have not examined mitochondrial defects at the later time point). We have not quantified the amount of PINK1/Parkin accumulation or signal strength in cells after treating CCCP or transfecting S59L. However, CCCP-induced PINK1/Parkin accumulation seems more robust than S59L-induced accumulation. Taken all these together, we think that high concentration CCCP is a more potent inducer for PINK1 and Parkin accumulation. However, S59L-induced PINK1/Parkin accumulation is also high enough to cause mitochondrial defects as evidenced by siRNA, peptide treatment, and PINK1^{KO} experiments.

If the reviewer asks whether the inducible levels of PINK1, Parkin recruitment to mitochondria with CCCP is also hyperactive in S59L expressing cells, we do not know the answer. Because S59L expression causes PINK1, Parkin accumulation in 85-90% of cells, to measure CCCP's effect on the PINK1, Parkin accumulation in S59L-expressing cells, we might need to titrate down the expression level of S59L and the concentration of CCCP. We have not tried the experiment. However, when we treated a low dose (5 μ M) of CCCP to S59L-expressing cells, it strongly induced cell death compared to normal HeLa cells within 16 hours. This result also suggests that S59L-induced mitochondrial defects are high enough but slightly less to cause robust cell death, such as high concentration CCCP treatment.

5. Could mitochondrial fragmentation and dysfunction simply and nonspecifically increase the mitochondrial recruitment PINK1/Parkin in S59L expressing cells? The PINK1-dependent and PINK1-independent roles of S59L on mitophagy/toxicity are still difficult to interpret.

The mislocalization and aggregation of CHCHD10^{S59L} might induce mitochondrial functional defects affecting many different proteins and mitochondrial cristae formation without a single specific or essential target. These mitochondrial defects may disrupt mitochondrial membrane potential and recruit PINK1 and Parkin in mitochondria. This case might be regarded as a nonspecific increase of PINK1/Parkin accumulation. To define how PINK1 can be stabilized and accumulated by S59L, we tested two known mechanisms, disruption of mitochondrial membrane potential and mitochondrial unfolded protein response. Currently, our studies are inconclusive, and the data are not included in this manuscript. However, based on our preliminary data [3], we do not think that the nonspecific membrane potential disruption is the mechanism of PINK1 accumulation in S59L-expressing cells (described in our first response letter). We also do not think that the mitochondrial UPR is a mechanism for PINK1/Parkin accumulation in S59L transiently transfected cells [3]. In the meantime, we do not know whether a specific downstream effector of S59L induces PINK1 accumulation. It will be interesting to find a specific effector of S59L directly involved in PINK1 stabilization and accumulation, such as a specific protease.

We found that both PINK1 and Parkin are genetic interactors with mutant C2C10H using our *Drosophila* model. We validated the *Drosophila* results in multiple human cells. Specifically, we used HeLa cells as a primary model to validate our *Drosophila* results because HeLa cells have been widely used to study the PINK1/Parkin pathways. Although other researchers studying CHCHD10 also have used HeLa cells often, HeLa cells do not express Parkin. Thus,

we used an engineered “HeLa cells stably expressing Parkin” to study Parkin’s effect and mitophagy on the defects caused by CHCHD10^{S59L}. With HeLa cells deficient Parkin expression, we showed that other PINK1 substrates such as mitofilin and mitofusin could induce mitochondrial defects without Parkin. This result does not necessarily mean that there are two different separate pathways, Parkin-dependent and Parkin-independent, in human patients. This result simply means that S59L-induced toxicity can be mediated by PINK1 and PINK1’s downstream substrates, including Parkin, mitofilin, and mitofusin. Original HeLa cells and “Hela stably expressing Parkin” models provided an opportunity to further delineate the role of genetic interactors, including mitofilin and mitofusin, on how S59L induces toxicity in HeLa cells without Parkin.

6. The authors show that WT CHCHD10 protects against TDP-43 insolubility and mitochondrial dysfunction. Similar findings were presented in a recent paper (FASEB 34:8493), which should be cited. The same paper also showed reduced CHCHD10 in FTLTDP & TDP-43 mice and that protective effects of CHCHD10 occur in part through Opa1/mitofilin. Given the investigation of the fusion protein Mfn1/2 and mitofilin as modifiers of S59L in the current study, the similarities, differences, and potential implications should be discussed.

We discussed the suggested paper and another paper in the discussion section.

Discussion:

While we were revising our manuscript, two independent groups reported that CHCHD10^{S59L}-mediated OMA1 peptidase activation [4], subsequent degradation of OPA1 resulted in mitochondrial fragmentation and defects [4], [5], and the protective effect of wild-type CHCHD10 against TDP-43 mitochondrial accumulation [5]. Because OMA1 and PINK1 can be activated in the same experimental conditions, it will be interesting to determine the relationship between the OMA1-OPA1 pathway and the PINK1-mediated pathway in the CHCHD10^{S59L}-induced pathogenesis. Although we have not observed significant protective effects of wild-type CHCHD10 against toxic TDP-43^{M337V} in Drosophila, it might be due to the strong overexpression of TDP-43^{M337V}, and it is still worth investigating the protective role of CHCHD10 in ALS-FTD and other degenerative diseases showing mitochondrial defects. Interestingly, in contrast to our findings, two groups also made similar findings in multiple CHCHD2 and CHCHD10 double knockout models [4] or by knocking down CHCHD10 in cell culture [5]. Although we demonstrated that the toxicity of CHCHD10^{S59L} was generated by a gain-of-toxicity mechanism in our model systems, we cannot completely rule out the possibility that other mechanisms also contribute to the CHCHD10^{S59L} pathogenesis. These studies and our results suggest that CHCHD10^{S59L} gain-of-toxicity, partial loss of normal CHCHD10, and dominant-negative like inhibition for CHCHD2 may co-exist or contribute to certain stages of disease pathogenesis.

Reviewer #3 (Remarks to the Author):

I think the manuscript is substantially improved in the revised version. Thanks for examining closely all of our previous comments.

We appreciate the reviewer’s contribution to making our manuscript better.

- [1] J. Iyer *et al.*, "Quantitative assessment of eye phenotypes for functional genetic studies using *Drosophila melanogaster*," *G3 Genes, Genomes, Genet.*, vol. 6, no. 5, pp. 1427–1437, 2016.
- [2] S. Berg *et al.*, "Ilastik: Interactive Machine Learning for (Bio)Image Analysis," *Nat. Methods*, vol. 16, no. 12, pp. 1226–1232, 2019.
- [3] M. Baek, Y.-J. Choe, G. Dorn, J. P. Taylor, and N. C. Kim, "Dominant toxicity of ALS–FTD-associated CHCHD10 S59L is mediated by TDP-43 and PINK1," 2019.
- [4] Y. T. Liu *et al.*, "Loss of CHCHD2 and CHCHD10 activates OMA1 peptidase to disrupt mitochondrial cristae phenocopying patient mutations," *Hum. Mol. Genet.*, vol. 29, no. 9, pp. 1547–1567, 2020.
- [5] T. Liu *et al.*, "CHCHD10-regulated OPA1-mitofilin complex mediates TDP-43-induced mitochondrial phenotypes associated with frontotemporal dementia," *FASEB J.*, no. December 2019, pp. 1–17, 2020.

Reviewers' Comments:

Reviewer #2:

Remarks to the Author:

The authors have addressed the concerns and questions of this reviewer. Thanks for making clarifications and revision.

Specific responses to each reviewer comment:

REVIEWERS' COMMENTS

Reviewer #2 (Remarks to the Author):

The authors have addressed the concerns and questions of this reviewer. Thanks for making clarifications and revision.

We appreciate your contribution to making our manuscript better.